# Weisfeiler Lehman Test on Combinatorial Complexes: Generalized Expressive Power of Topological Neural Networks

## Abstract

Combinatorial complexes have unified set-based (e.g., graphs, hypergraphs) and part-whole (e.g., simplicial, cellular complexes) structures into a common topological framework. Existing topological neural networks and Weisfeiler-Lehman variants remain fragmented, lacking a unified theoretical foundation for topological deep learning. In this work, we introduce the Combinatorial Complex Weisfeiler-Lehman (CCWL) test, an axiomatic-style extension of the WL test to combinatorial complexes. CCWL formalizes topological message passing through four types of neighborhood relation and provides a unified perspective on the expressive power of higher-order variants. We further prove that upper and lower neighborhoods are sufficient among the four adjacent WL tests to reach the expressivity of the full CCWL framework across topological structures of combinatorial complexes. Building on this framework, we also propose the Combinatorial Complex Isomorphism Network (CCIN) and evaluate it on synthetic and real-world benchmarks [1]. Experimental results indicate CCIN outperforms baseline methods and offers a generalized expressive framework for topological deep learning.

## 1. Introduction

Combinatorial complexes (CCs) have recently emerged as a unifying topological formalism that simultaneously encodes set-type relations, such as those in graphs and hypergraphs, and part-whole hierarchical relations, as found in simplicial and cellular complexes (Hajij et al., 2022; 2023). Many scientific and relational datasets naturally exhibit interactions that go beyond pairwise graphs. To capture such higher-order dependencies, hypergraphs (Feng et al., 2024), simplicial complexes (Gurugubelli & Chepuri, 2023; Taha et al., 2025), and cell complexes (Bodnar et al., 2021a; Eitan et al., 2025) have been widely adopted as expressive representations of multi-way, hierarchical, and geometric structures arising in physical simulations, biological networks, and geometric modeling (Papamarkou et al., 2024b; Millán et al., 2025). CCs have become the unified higher-order topological frameworks in topological deep learning.

Prior neural architectures for combinatorial complexes have demonstrated the potential of learning over rich higher-order topological structures. Existing higher-order neural networks are typically designed for specific structures, including hypergraph neural networks (Huang & Yang, 2021), simplicial neural networks, and cell complex networks (Taha et al., 2025; Huang et al., 2024; Battiloro et al., 2024). These models rely on incompatible neighborhood definitions and message-passing mechanisms, reflecting different ways of generalizing graphs through set-based expansions or hierarchical refinements (Millán et al., 2025). To address this, TopoTune (Papillon et al., 2025) studies combinatorial complex neural networks, but simplifies as three neighborhood types and disrupts the higher-order information propagation. However, these models fail to faithfully preserve the topological neighborhood of combinatorial complexes and limits the expressive ability (Hajij et al., 2023; Papillon et al., 2023; Besta et al., 2025). This highlights the need for a principled and expressive framework that preserves the higher-order neighborhoods and topological message passing inherent to combinatorial complexes.

Weisfeiler-Lehman (WL) has established theoretical foundation for characterizing the expressive power of graph neural networks (GNNs) (Weisfeiler & Leman, 1968). For graphs, the 1-WL test characterizes the expressiveness of message-passing GNNs (Maron et al., 2018; Xu et al., 2018; Feng et al., 2022; Puny et al., 2023). In parallel, recent works explore substructure-aware GNNs (Zhang et al., 2024a; 2023b; Thiede et al., 2021; Zhang et al., 2023a; 2024b; Bevilacqua et al., 2022; Bouritsas et al., 2022), studying expressive subgraphs and paths from a structural and permutation perspective. When extending beyond graphs to

---

[1]Anonymous Institution, Anonymous City, Anonymous Region, Anonymous Country. Correspondence to: Anonymous Author <anon.email@domain.com>.

Preliminary work. Under review by the International Conference on Machine Learning (ICML). Do not distribute.

[1]Code available at https://anonymous.4open.science/r/CCIN-26009

higher-order topological structures, including Hypergraph WL for hypergraphs that focus on node-hyperedges (Zhang et al., 2025b; Feng et al., 2024), MPSN employs on boundary and coboundary of simplicial complexes (Bodnar et al., 2021b), and Cellular WL operates on different dimensional cell of cellular complexes (Bodnar et al., 2021a). However, existing WL variants are designed for domain-specific and mutually incomparable. There remains the absence of a common theoretical foundation (Papp & Wattenhofer, 2022; Papamarkou et al., 2024a), which limits the understanding of expressive power on combinatorial complexes.

Combinatorial complexes naturally admit multiple types of neighborhood relations, including upper and lower adjacencies as well as boundary and co-boundary relations (Hajij et al., 2022; Eitan et al., 2025), which together provide a complete description of higher-order interactions. However, all four relations are incorporated into message passing often results in redundant and repetitive information propagation. To reduce this redundancy, existing higher-order neural architectures typically restrict message passing to a subset of neighborhood relations. For instance, simplicial networks such as MPSN focus on boundary and co-boundary interactions (Bodnar et al., 2021a; Eijkelboom et al., 2023; Wu et al., 2023), while cellular networks propagate information along incident and adjacent cell relations (Morris et al., 2023). Similarly, hypergraph WL tests and neural models rely on node-hyperedge relations (Zhang et al., 2025b; Feng et al., 2024), but it ignores the interactions between hyperedges. However, such simplifications inevitably overlook certain higher-order dependencies (Taha et al., 2025; Chen et al., 2026). More broadly, the absence of a Weisfeiler-Lehman refinement defined on combinatorial complexes prevents a unified theoretical framework across higher-order topological domains. A critical challenge is to demonstrate the minimal neighborhood function refinement rule that preserves the upper bound of expressive power.

The benefits of combinatorial complexes generalize set-type and part-whole topological structures. A generalization of Weisfeiler-Lehman to combinatorial complexes fills the gap the unified framework of topological message passing and expressive power on combinatorial complexes. The conceptual framework and theoretical results developed in this paper directly address the open problems and challenges (e.g., research directions 7 and 9) highlighted in the position literature (Papamarkou et al., 2024a;b), including topological message passing generalization, and the expressivity power proofs of topological invariance in TDL models.

**Contributions:** Our main contributions are summarized as:

- **(Axiomatic):** We introduce an unified framework of Combinatorial Complex Weisfeiler-Lehman test (CCWL) and their message passing through the four neighbor function refinement. CCWL enables the prin-

cipled modeling of graph, hypergraph, simplicial complex, and cellular complex structures.

- **(Theoretical)** We formally define the CCWL refinement and prove that the upper and lower adjacency refinements are sufficient for the full CCWL (Theorem 4.8). We extend CCWL results on four structures, analyzing the upper bound (Lemma 4.9) and generalized variants (Theorem 4.11) for higher-order domains.

- **(Empirical)** We evaluate CCWL on both synthetic and real-world datasets spanning graphs, hypergraphs, and higher-order topological structures. CCWL consistently achieves superior performance on graph and complex-level classification tasks, demonstrating its practical advantages and strong generalization across heterogeneous topological structures.

## 2. Related Works

**Topological Neural Networks for Higher-order Graph.** Motivated by the limitations of pairwise graphs, a wide range of higher-order neural networks has been developed to model topological relational structures (Pham et al., 2025; Verma et al., 2024; Chen et al., 2021) in Topological Deep Learning (TDL). Hypergraph neural networks extend GNNs via node-hyperedge incidence relations (Feng et al., 2019; Huang & Yang, 2021; Xie et al., 2025). Simplicial and cellular neural networks exploit boundary, co-boundary, and adjacency operators to propagate information across cells of different dimensions (Ebli et al., 2020; Bodnar et al., 2021a; Eijkelboom et al., 2023; Wu et al., 2023; Liu et al., 2024; Hajij et al., 2020). More recent works incorporate equivalence or attention mechanisms to enhance geometric modeling capacity (Battiloro et al., 2025; Ballester et al., 2024). While these architectures successfully capture structure-specific higher-order interactions, their message-passing mechanisms are limited by being tied to a fixed subset of neighborhood relations (Taha et al., 2025; Eitan et al., 2025). In practice, our formulation is characterized through a novel model like combinatorial complex to simplify the rich neighborhood structure of higher-order domains.

**Expressive Power of Topological Deep Learning.** Weisfeiler-Lehman framework provides a principled foundation for the expressive power of topological deep learning. Recent methods have been proposed on low-order structures. On graphs, the 1-WL test captures the discriminative power of message-passing GNNs, while higher-order variants such as $k$-WL, sparse WL, and loopy WL offer increased expressivity at the cost of higher computational complexity (Weisfeiler & Leman, 1968; Maron et al., 2018; Xu et al., 2018; Morris et al., 2020b; Paolino et al., 2024; Papp & Wattenhofer, 2022; Zhang et al., 2025a). Beyond graphs, a growing body of work studies expressivity through substructure-

aware or equivariant extensions, including subgraph-WL architectures (Zhang et al., 2024a; 2023b; Thiede et al., 2021; Zhang et al., 2023a; 2024b; Bevilacqua et al., 2022; Bouritsas et al., 2022). Several WL extensions have also been proposed for higher-order domains. Hypergraph WL tests refine node and hyperedge representations through bipartite incidence relations (Zhang et al., 2025b; Feng et al., 2024). For simplicial complexes, message-passing simplicial networks induce a WL-type refinement via boundary and co-boundary operators (Bodnar et al., 2021b; Eitan et al., 2025). Similarly, cellular WL defines refinement rules over cells of different dimensions (Bodnar et al., 2021a; Chen et al., 2022; Truong & Chin, 2024; Papillon et al., 2025). However, these WL variants are domain-specific with different neighborhood systems and refinement rules. Their expressive powers are not directly comparable to characterize expressivity across higher-order topological domains (Papamarkou et al., 2024a). Our work operates on combinatorial complexes and preserves the inherent neighbors function to unify message passage on higher-order graph structures.

**Combinatorial Complexes Studies.** Combinatorial complexes emerge to model the general topological relations (Hajij et al., 2022; 2023). Several frameworks leverage CCs to design flexible neural networks, including TopoTune (Papillon et al., 2025), SMCN (Eitan et al., 2025), topological neural networks (Battiloro et al., 2025), CCMamba (Chen et al., 2026) and modular toolkits for learning on generalized topological domains (Telyatnikov et al., 2024; Hajij et al., 2024; Anonymous, 2026). Table 1 shows previous studies mainly focus on specific high-order modeling, and still lack a unified model for the four structures. Our research is dedicated to establishing the framework of expressive power from WL test on combinatorial complexes.

## 3. Preliminaries

Combinatorial complexes is a frontier generalization structure that combines features of cellular complexes and hypergraphs. Here, cellular complexes are a generalization of simple complexes (regular faces, tetrahedrons), and hypergraphs generalize graphs whose element hyperedges can connect any set of pairwise or several nodes.

We define combinatorial complexes as follows.

**Definition 3.1** (**Combinatorial Complex**). *A combinatorial complex (CC) is a triple $(\mathcal{S}, \mathcal{C}, rk)$ consisting of a set $\mathcal{S}$, a subset $\mathcal{C}$ of $P(S)$, and a function $rk : \mathcal{C} \to \mathbb{Z}_{\geq 0}$ such that for all $v \in \mathcal{S}$, $\{v\} \in \mathcal{C}$ and $rk(\{v\}) = 0$. The function $rk$ is order-preserving, if $\sigma, \tau \in \mathcal{C}$ satisfy $\sigma \subseteq \tau$, then $rk(\sigma) \leq rk(\tau)$. If rank of a cell $\sigma \in \mathcal{C}$ is $k$, then we denote it as a $k-$cell. The dimension of $CC$ is defined as the maximal rank among its cell: $dim(\mathcal{CC}) := \max_{\sigma \in \mathcal{CC}} rk(\sigma)$.*

*Table 1.* Overview of expressive power methods and their corresponding structure domains. Supports (✓) or does not support (✗).

| Model | Graph | Hypergraph | Simplex | Cellular |
|---|---|---|---|---|
| GIN (Xu et al., 2018) | ✓ | ✗ | ✗ | ✗ |
| rMPNN (Paolino et al., 2024) | ✓ | ✗ | ✗ | ✗ |
| HomoGNN (Zhang et al., 2024a) | ✓ | ✗ | ✗ | ✗ |
| KGWL (Zhang et al., 2025b) | ✓ | ✗ | ✗ | ✗ |
| ED-HNN (Wang et al., 2023a) | ✗ | ✓ | ✗ | ✗ |
| IMPSN (Eijkelboom et al., 2023) | ✗ | ✓ | ✗ | ✗ |
| RePHINE (Immonen et al., 2023) | ✗ | ✓ | ✗ | ✗ |
| EMPSN (Eijkelboom et al., 2023) | ✗ | ✓ | ✗ | ✗ |
| MPSN (Bodnar et al., 2021b) | ✗ | ✗ | ✗ | ✓ |
| CWN (Bodnar et al., 2021a) | ✗ | ✗ | ✓ | ✗ |
| TopNets (Verma et al., 2024) | ✗ | ✗ | ✗ | ✓ |
| ETNNs (Battiloro et al., 2025) | ✗ | ✗ | ✓ | ✗ |
| SMCN (Eitan et al., 2025) | ✗ | ✗ | ✓ | ✗ |
| CCIN (ours) | ✓ | ✓ | ✓ | ✓ |

This definition is useful for referring to specific structural levels of the complex. i.e., $\mathcal{CC}^{(0)}$ contains only the nodes (0-cells), $\mathcal{CC}^{(1)}$ contains the nodes and edges (0- and 1-cells), $\mathcal{CC}^{(2)}$ contains the full 2-skeleton including nodes, edges, and faces(2-cells). The combinatorial structure of $\mathcal{CC}$ can be compactly described by an incidence relation called the boundary relation, denoted $\sigma \prec \tau$, indicating that cell $\sigma$ lies on the boundary of a higher-dimensional cell $\tau$. The reflexive and transitive closure of this relation induces a partial order over the set of cells as $\sigma \preceq \sigma$, and if $\sigma \prec \tau$ and $\tau \prec \rho$, then $\sigma \prec \rho$. such as the boundary relation $\sigma \prec \tau$, iff $\sigma \prec \tau$ and there is no cell $\delta$ such that $\sigma \prec \delta \prec \tau$.

Now we introduce the four neighborhood incidence relation.

**Definition 3.2** (**Cell Adjacencies**). *For cell complex $\mathcal{CC}$ and a cell $\sigma \in P(S)$, we define four type adjacencies as*

*1. Boundary adjacent $\mathcal{B}(\sigma) = \{\tau | \tau \prec \sigma\}$.*

*2. Co-boundary adjacent $\mathcal{C}(\sigma) = \{\tau | \sigma \prec \tau\}$.*

*3. Lower adjacent $\mathcal{N}_{\downarrow}(\sigma) = \{\tau | \exists \delta, s.t. \ \delta \prec \sigma \ and \ \delta \prec \tau\}$.*

*4. Upper adjacent $\mathcal{N}_{\uparrow}(\sigma) = \{\tau | \exists \delta, s.t. \ \sigma \prec \delta \ and \ \tau \prec \delta\}$.*

Importantly, the message passing mechanism usually considers multi-hop neighborhood relationships in $\mathcal{CC}$. A cell span multiple ranks and propagate from 0-cell to 2-cell, e.g., messages pass from node to face. In this work, we assume that all incidences of cell elements are rank-adjacent, and the complex contains no free faces.

## 4. Combinatorial Complex Weisfeiler Lehman

Based on the neighborhood function, we introduce the Weisfeiler-Lehman test on combinatorial complex $\mathcal{CC}$ in Figure 1. The coloring function $c$ at the $t$-th iteration is $c^{(t)} : \mathcal{C} \to \mathbb{N}$, where the color values represent the labels of topological structure. The multisets $\{\{\cdot\}\}$ generalizes the sets that allow multiple instances of each element.

**Definition 4.1.** *Let $c$ be a coloring of the cells in a complex*

*Figure 1.* Illustration of Combinatorial Complex Weisfeiler-Lehman Test . CCWL test conducts a combinatorial complex, which consists of four nodes $v_1, v_2, v_3, v_4$, five edges $e_{12}, e_{13}, e_{23}, e_{24}, e_{34}$ and two faces $f_{123} = \{e_{12}, e_{13}, e_{23}\}$, $f_{234} = \{e_{23}, e_{24}, e_{34}\}$. At the initialization, 0-cells (nodes) $c_0^{(0)} = \{\bullet\}$, 1-cells (edges) $c_1^{(0)} = \{\bullet\}$, 2-cells (faces) $c_2^{(0)} = \{\bullet\}$. CCWL test stables at iteration $t = 2$.

$X$ with $c_\sigma$ denoting the color assigned to cell $\sigma \in P_X$. Define $\mathcal{B}(\sigma, \tau) := \mathcal{B}(\sigma) \cap \mathcal{B}(\tau)$ and $\mathcal{C}(\sigma, \tau) := \mathcal{C}(\sigma) \cap \mathcal{C}(\tau)$. We define the following multisets of colors for cell $\sigma$:

1. The boundary colors: $c_\mathcal{B}(\sigma) = \{\{c_\tau | \tau \in \mathcal{B}(\sigma)\}\}$.

2. The co-boundary colors: $c_\mathcal{C}(\sigma) = \{\{c_\tau | \tau \in C(\sigma)\}\}$.

3. The lower adjacent colors: $c_{\mathcal{N}_\downarrow}(\sigma) = \{\{(c_\tau, c_\delta) | \tau \in \mathcal{N}_\downarrow(\sigma) \text{ and } \delta \in \mathcal{B}(\sigma, \tau)\}\}$.

4. The upper adjacent colors: $c_{\mathcal{N}_\uparrow}(\sigma) = \{\{(c_\tau, c_\delta) | \tau \in \mathcal{N}_\uparrow(\sigma) \text{ and } \delta \in C(\sigma, \tau)\}\}$.

Building on the four neighborhood coloring multisets, we define the update rule for cell $\sigma \in \mathcal{CC}$ as $c^{(t+1)}(\sigma) = \text{HASH}(c^{(t)}(\sigma), c_\mathcal{B}^{(t)}(\sigma), c_\mathcal{C}^{(t)}(\sigma), c_{\mathcal{N}_\downarrow}^{(t)}(\sigma), c_{\mathcal{N}_\uparrow}^{(t)}(\sigma))$, where HASH is an injective multiset hashing function. The details of coloring process can be found in Appendix A.

To distinguish non-isomorphic combinatorial complexes, we generalize the WL test on the combinatorial complexes, and introduce the iterative color refinement algorithm CCWL.

**Definition 4.2** (CCWL Test). *Let $\mathcal{CC}_1 = (\mathcal{S}_1, \mathcal{C}_1, \text{rk}_1)$ and $\mathcal{CC}_2 = (\mathcal{S}_2, \mathcal{C}_2, \text{rk}_2)$ be two CC, and a coloring function $c$ on cells. Given initial features $\{h_\sigma \mid \sigma \in \mathcal{C}_1 \cup \mathcal{C}_2\}$ and neighborhood orders $k \subseteq \mathbb{Z}^+$ (default: $k = \{1\}$), the CCWL iteration proceeds as follows:*

*1. All the cell of the same rank $\sigma \in \mathcal{CC}_1, \tau \in \mathcal{CC}_2$ are initialized with the same color, defined as $c^{(0)}(\sigma), c^{(0)}(\tau)$.*

*2. Given the color $c^{(t)}$ of cell at iteration $t$, we update the color of next iteration as $c^{(t+1)}$ by injective mapping the multisets of colors belonging to the adjacent cells of $\sigma$ by HASH function: $c^{(t+1)} = \text{HASH}(c^{(t)}, c_\mathcal{B}^{(t)}, c_\mathcal{C}^{(t)}, c_{\mathcal{N}_\downarrow}^{(t)}, c_{\mathcal{N}_\uparrow}^{(t)})$,*

*3. It stops when a stable coloring is reached.*

For example, we provide the rules for the WL test on the combinatorial complex in Figure 1(a), and Figure 1(b)(c) show that the CCWL test stops and reaches a stable state $c^{(1)} = c^{(2)}$ after two iterations. Two combinatorial complexes are considered non-isomorphic if their relabeled colors are different. Otherwise, the test is inconclusive. Then we define the combinatorial complex isomorphism as

**Definition 4.3** (Combinatorial Complex Isomorphism). *Given two combinatorial complexes $\mathcal{CC}_1 = (\mathcal{S}_1, \mathcal{C}_1, rk_1)$ and $\mathcal{CC}_2 = (\mathcal{S}_2, \mathcal{C}_2, rk_2)$, the object is to demonstrate the isomorphism as $\mathcal{CC}_1 \cong \mathcal{CC}_2$ if there exists a bijective function $\mathcal{A} : \mathcal{CC}_1 \to \mathcal{CC}_2$. The mapping $\mathcal{A}$ holds that*

$$(c_i, c_j) \in \mathcal{CC}_1 \Rightarrow (\mathcal{A}(c_i), \mathcal{A}(c_j)) \in \mathcal{CC}_2. \quad (1)$$

However, unlike the isomorphism function in the graph structure, the labeling information process between nodes and the relabeling process of different rank structures (e.g., nodes, edges and faces) in the combinatorial complex make the injective transfer have information redundancy. Beyond individual cell comparisons, we can also relate colorings at different levels. We formalize the refinement relationships between colorings can propagate across complexes.

**Lemma 4.4.** *For two combinatorial complexes $\mathcal{CC}_1, \mathcal{CC}_2$ are isomorphic if and only if for iterations $\forall t \geq 0$, they satisfy $\{\{c^{(t)}(\sigma) : \sigma \in \mathcal{CC}_1\}\} = \{\{c^{(t)}(\tau) : \tau \in \mathcal{CC}_2\}\}$.*

**Lemma 4.5.** *Let $\mathcal{CC}_1, \mathcal{CC}_2$ be any regular CC with $A \subseteq P_{\mathcal{CC}_1}, B \subseteq P_{\mathcal{CC}_2}$. Consider two combinatorial colorings $c, d$ such that $c \sqsubseteq d$. If $\{\{d_\sigma^{\mathcal{CC}_1} | \sigma \in A\}\} \neq \{\{d_\tau^{\mathcal{CC}_2} | \tau \in B\}\}$, then $\{\{c_\sigma^{\mathcal{CC}_1} | \sigma \in A\}\} \neq \{\{c_\tau^{\mathcal{CC}_2} | \tau \in B\}\}$.*

**Remark.** Two coloring functions $c$ and $d$ satisfying the relation $c \sqsubseteq d$ indicate that the coloring function $d$ has a stronger ability to distinguish non-isomorphic structures compared to the coloring function $c$.

This Lemma further yields a corollary, showing that differences in the coloring function emerge at a refined level.

**Corollary 4.6.** *Consider two colorings functions $c, d$ such that $c \sqsubseteq d$. For all combinatorial complexes $CC_1, CC_2$, if $d^{CC_1} \neq d^{CC_2}$, then $c^{CC_1} \neq c^{CC_2}$.*

Recent works have proved the upper bound of the expressive power of GNNs (Xu et al., 2018), and hypergraph isomorphism computation on HGNNs (Feng et al., 2024). Similarly, the expressive power of part-whole relations neural networks are also explored (Bodnar et al., 2021a;b). We observe that the two-stage operations on hypergraph (Zhang et al., 2025b) as $c_e^{(t)} = \text{HASH}(c_e^{(t-1)}, \{\{c_u^{(t-1)}, u \in \mathcal{N}_v(e)\}\})$, $c_v^{(t)} = \text{HASH}(c_v^{(t-1)}, \{\{c_u^{(t-1)}, u \in \mathcal{N}_e(v)\}\})$ are structurally equivalent to upper adjacent colors. Moreover, the WL on cellular update follows $\text{HASH}(c^{(t)}, c_{\mathcal{B}}^{(t)}, c_{\mathcal{N}_\uparrow}^{(t)})$. However, in hypergraphs, the use of boundary proximity information during downpropagation does not take into account lower adjacency. The WL test can be generalized to combinatorial complexes with four adjacencies. In practice, we observe that upper adjacent colorings subsume boundary colorings, i.e., $c_{\mathcal{B}} \sqsubseteq c_{\mathcal{N}_\uparrow}$, and downward adjacent colorings subsume co-boundary colorings, i.e., $c_{\mathcal{C}} \sqsubseteq c_{\mathcal{N}_\downarrow}$. Based on Corollary 4.6, we further derive the neighborhood refinements for the topological message update process.

**Lemma 4.7.** *CCWL with $\text{HASH}(c_\sigma^t, c_{\mathcal{B}}^t(\sigma), c_{\mathcal{N}_\downarrow}^t(\sigma), c_{\mathcal{N}_\uparrow}^t(\sigma))$ is as powerful as CCWL with the generalized update rule $\text{HASH}(c_\sigma^t, c_{\mathcal{B}}^t(\sigma), c_{\mathcal{C}}^t(\sigma), c_{\mathcal{N}_\downarrow}^t(\sigma), c_{\mathcal{N}_\uparrow}^t(\sigma))$.*

**Theorem 4.8.** *CCWL with $\text{HASH}(c_\sigma^{(t)}, c_{\mathcal{N}_\downarrow}^{(t)}(\sigma), c_{\mathcal{N}_\uparrow}^{(t)}(\sigma))$ is as powerful as CCWL with the generalized update rule $\text{HASH}(c^{(t)}(\sigma), c_{\mathcal{B}}^{(t)}(\sigma), c_{\mathcal{C}}^{(t)}(\sigma), c_{\mathcal{N}_\downarrow}^{(t)}(\sigma), c_{\mathcal{N}_\uparrow}^{(t)}(\sigma))$.*

Building on this, we can derive this lemma as

**Lemma 4.9.** *CCWL is at least as powerful as 1-WL in distinguishing of non-isomorphic combinatorial complexes.*

The details of the proof are in Appendix B.1. We establish the CCWL test generalizes and strengthens existing WL variants on graphs, simplicial complexes, and cellular complexes through lifting into the combinatorial complex.

**Definition 4.10** (**Combinatorial Lifting Map**). *A combinatorial lifting map is a function $f : \mathcal{G} \to CC$ be mapping a graph to a regular combinatorial complex such that $\mathcal{G}_1 \cong \mathcal{G}_2 \iff f(\mathcal{G}_1) \cong f(\mathcal{G}_2)$.*

Example (Clique Complex Lifting to CC): Let $f(\mathcal{G})$ be the clique-based combinatorial complex, where each node in $\mathcal{G}$ maps to a 0-cell, each edge maps to a 1-cell, each $(k+1)$-clique maps to a k-dimensional cell with rank $k$. The incidence structure is preserved in the complexes, the $CCWL(f(\mathcal{G})) \succ WL(\mathcal{G})$ in expressive power, i.e. it can distinguish regular graphs with different triangle counts.

**Theorem 4.11.** *Let $\mathcal{G}, \mathcal{H}, \mathcal{S}, \mathcal{C}$ be a graph, hypergraph, simplicial complex and cellular complex, respectively. There exists a mapping from each of these structures into a combinatorial complex $CC$, the CCWL test can simulate the 1-WL, hypergraph WL, simplicial-WL and cellular-WL tests.*

We discussed some non-isomorphic structural pairs as counterexamples that cannot be distinguished by these WL tests, but can be distinguished by the CCWL test in Appendix B.2.

# 5. Combinatorial Complex Neural Networks

We begin with the message passing of Combinatorial Complex Neural Networks (CCNN) (Hajij et al., 2022). Suppose $CC$ be a combinatorial complex, $N_{\mathcal{C}}$ be a collection of four neighborhood functions. The $l$-th layer of CCNN updates the embedding $h_\sigma^l \in \mathbb{R}^{F^l}$ of cell $\sigma$ can be written as

$$h_\sigma^{l+1} = \phi \left( h_\sigma^l, \bigotimes_{\mathcal{N} \in N_{\mathcal{C}}} \underset{\tau \in \mathcal{N}(\sigma)}{\text{AGG}} \left( M_{\mathcal{N}(\sigma)}(h_\sigma^l, h_\tau^l) \right) \right), \quad (2)$$

where $h_\sigma^0 := h_\sigma$ are the initial features, $\bigotimes$ is an intra-neighborhood aggregator, AGG represents the aggregation function, $M_{(.)}$ denotes as messing information of different neighborhood, and the update function $\phi$ are learnable functions. To clarify the higher-order message passing on a $CC$, we introduce the message passing operations of a combinatorial complex with four adjacent types from Definition 3.2. For a cell $\sigma \in \mathcal{C}$, we define four message passing rules as

$$
\begin{aligned}
m_{\mathcal{B}}^{t+1}(\sigma) &= \underset{\tau \in \mathcal{B}(\sigma)}{\text{AGG}} \left( M_{\mathcal{B}} \left( h_\sigma^t, h_\tau^t \right) \right), \\
m_{\mathcal{C}}^{t+1}(\sigma) &= \underset{\tau \in \mathcal{C}(\sigma)}{\text{AGG}} \left( M_{\mathcal{C}} \left( h_\sigma^t, h_\tau^t \right) \right), \\
m_{\mathcal{N}_\downarrow}^{(t+1)}(\sigma) &= \underset{\tau \in \mathcal{N}_\downarrow, \delta \in \mathcal{B}(\sigma, \tau)}{\text{AGG}} \left( M_{\mathcal{N}_\downarrow}(h_\sigma^{(t)}, h_\tau^{(t)}, h_\delta^{(t)}) \right), \\
m_{\mathcal{N}_\uparrow}^{(t+1)}(\sigma) &= \underset{\tau \in \mathcal{N}_\uparrow, \delta \in \mathcal{C}(\sigma, \tau)}{\text{AGG}} \left( M_{\mathcal{N}_\uparrow}(h_\sigma^{(t)}, h_\tau^{(t)}, h_\delta^{(t)}) \right).
\end{aligned}
\quad (3)
$$

Within the Theorem 4.8, we have proved that CCWL without boundary and co-boundary has the same expressive power in distinguishing non-isomorphic combinatorial complexes as CCWL with the complete set of adjacencies. Therefore, the cells in combinatorial complex neural networks receive two types of messages, then update operation takes into account these two types of incoming messages and updates the features of the cells:

$$h_\sigma^{(t+1)} = \text{Update} \left( h_\sigma^{(t)}, m_{\mathcal{N}_\downarrow}^{(t+1)}(\sigma), m_{\mathcal{N}_\uparrow}^{(t+1)}(\sigma) \right). \quad (4)$$

To obtain a global embedding from a $d$-dimension combinatorial complex $CC$ of a CCNN with $L$ layers, the readout function takes as input the sets of features corresponding to all dimensions of the combinatorial complex

$$h_{CC} = \text{READOUT}(\{\{h_\sigma^{(L)}\}\}_{\dim=0}, \ldots, \{\{h_\sigma^{(L)}\}\}_{\dim=d}), \quad (5)$$

where READOUT can be a permutation invariant function, such as sum, mean, or max pooling function, resulting in the process differentiable computational operations.

As before, CCWL has developed the sufficient conditions for the powerful CCNN over multi-rank structures. we further propose the Combinatorial Complex Isomorphism Network (CCIN), which leverages summation-based aggregation and rank-adaptive update functions to achieve high expressiveness on topological neural networks. Similar to the Equation 2, the CCIN layer can be defined as

$$h_\sigma^{(l+1)} = \phi^{(l)}\Big((1 + \epsilon^{(l)})h_\sigma^{(l)} + \sum_{\mathcal{N} \in \mathcal{N}_C} \sum_{\tau \in \mathcal{N}(\sigma)} \psi_{\mathcal{N}}^{(l)}(h_\sigma^{(l)}, h_\tau^{(l)})\Big),$$

(6)

where $\epsilon$ is a learnable parameter, and $\psi_{\mathcal{N}}^{(l)}(\cdot, \cdot)$ represents a learning function for neighborhood massage. Here these function can also employ summation, max and mean aggregation function. Following the Theorem.4.11, we conclude that combinatorial complexes neural networks Update$(h_\sigma^{(t)}, h_{\mathcal{N}_\downarrow}^{(t)}(\sigma), h_{\mathcal{N}_\uparrow}^{(t)}(\sigma))$ exhibit the CCWL test's ability to distinguish non-isomorphic combinatorial complexes through sufficient number of layers and injectivity of local aggregation functions.

**Lemma 5.1.** *CCIN is as powerful as CCWL when we employ an injective neighborhood aggregators and a sufficient number of MLP layers.*

The details and proofs are in Appendix C.1.

## 6. Experimental Analysis

we validate the theoretical and empirical properties of our proposed message passing scheme in synthetic and real-world graph classification tasks. The lack of combinatorial complex benchmarks has been recognized as a challenge in TDL (Papamarkou et al., 2024a;b). To address this, we employ the lifting methods (Telyatnikov et al., 2024; Bodnar et al., 2021a) to infer the higher-order structure by applying cyclic lifting on real-world graph benchmarks in these graph data, which evaluates the ability of CCIN and other benchmark methods to capture topological properties. We provide additional information about experiments in Appendix D.

### 6.1. Synthetic Datasets

**Strongly Regular Graphs.** To study the expressivity of CCIN, we employ strongly regular graphs within the same family as indistinguishable non-isomorphic graphs. For the strongly regular graph families, since any two graphs within the same family cannot be distinguished by the 3-WL test (Bouritsas et al., 2022). We build an untrained model with a complex size of $k = 4$ for each graph. Figure 2 shows the percentage of failure rates. The failure rates indicate that CCIN achieves the superior performance over GIN, MSPN

and 3-WL baseline models on SR families (in Appendix Table D.7, a larger k value achieves better performance). CCIN effectively distinguishes regular graphs, confirming its expressive power on the higher-order structures.

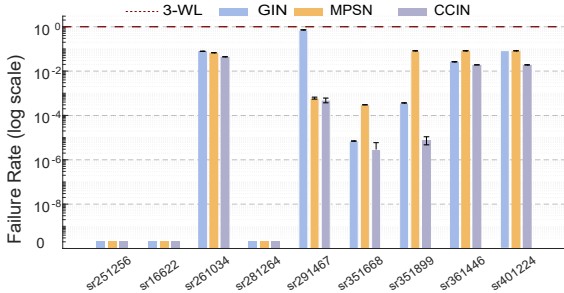

*Figure 2.* Failure rate on SR families(ring num $k = 4$).

### 6.2. Real-World Datasets

**TUDataset.** We also evaluate CCIN with recent baselines on several real-world datasets from the TUDataset(Morris et al., 2020a) consists of bioinformatics and social networks, where graphs are lifted into cell complexes to model higher-order interactions. MUTAG (Kazius et al., 2005), PTC, PROTEINS(Borgwardt et al., 2005; Dobson & Doig, 2003) consist of graphs where nodes represent atoms and edges are chemical bonds. NCI1 and NCI109 (Wale et al., 2008) contain the chemical compounds screened for activity against various cancer cell lines. IMDB-B, IMDB-M, REDDIT-B and REDDIT-M (Yanardag & Vishwanathan, 2015; Hu et al., 2020) are preprocessed movie collaboration and Reddit networks, where graphs represent interactions between actors or users, and the tasks involve graph classification. The details of datasets can be found in Appendix D.1.

Table 2 reports the graph classification performance of CCIN on the TUDataset benchmark dataset, where CCIN achieved excellent results over baseline methods. Compared to traditional message-passing graph neural networks, CCIN demonstrates a clear advantage on most datasets. On MUTAG and IMDB-B, CCIN achieves improvements of 7.82% and 4.09% respectively compared to GIN, and 8.90% gains (83.2 vs. 76.4) compared to the GraphSNN model on NCI1, and 2.69% benefits (76.1 vs. 74.1) compared to PROTEINs. These results indicate that GNNs, which rely on node-level message passing, have limitations in modeling high-order structures. Furthermore, compared to higher-order topological neural networks, CCIN also demonstrates competitive or even superior performance. Compared to CWN, CCIN achieves a 7.11% improvement on MUTAG (96.4 vs. 90.0) and a 6.53% improvement on IMDB-B (78.3 vs. 73.5). Compared to TopNets and TopoTune topological methods, CCIN maintains the notable improvement on the MUTAG and IMDB datasets. These results indicate CCIN achieves

*Table 2.* Graph classification accuracy (Mean±Std) of our CCIN and the baselines on the datasets from TUDataset collection. Best performance is highlighted in bold. N/A means not available.

| | MUTAG | PTC | PROTEINS | NCI1 | NCI109 | IMDB-B | IMDB-M |
|---|---|---|---|---|---|---|---|
| RWK (Gärtner et al., 2003) | 79.2±2.1 | 55.8±0.6 | 59.6±0.3 | N/A | N/A | N/A | N/A |
| GK(k = 3) (Shervashidze et al., 2009) | 81.4±1.9 | 55.4±0.4 | 71.4±1.5 | 62.5±0.3 | 64.9±1.0 | 50.9±3.8 | N/A |
| WL kernel (Shervashidze et al., 2011) | 90.4±5.7 | 59.9±4.3 | 75.0±3.1 | 86.0±1.8 | 73.8±3.9 | N/A | N/A |
| DGCNN (Zhang et al., 2018a), | 85.8±1.8 | 58.6±2.5 | 75.5±0.9 | 74.4±0.5 | N/A | 70.0±0.9 | 47.8±0.9 |
| IGN (Cai & Wang, 2022) | 83.9±13.0 | 58.5±6.9 | 76.6±5.5 | 74.3±2.7 | 72.8±1.5 | 72.0±5.5 | 48.7±3.4 |
| GIN (Xu et al., 2018) | 89.4±5.6 | 64.6±7.0 | 76.2±2.8 | 82.7±1.7 | N/A | 75.1±5.1 | 52.0±2.8 |
| PPGNs (Maron et al., 2018) | 90.6±8.7 | 66.2±6.6 | 77.2±4.7 | 83.2±1.1 | 80.2±1.4 | 73.0±5.8 | 50.5±3.6 |
| Natural GN (de Haan et al., 2020) | 89.4±1.6 | 66.8±1.7 | 71.7±1.0 | 82.4±1.3 | N/A | 74.8±2.0 | 51.3±1.5 |
| GSN (Bouritsas et al., 2022) | 92.2±7.5 | 67.2±7.2 | 75.6±5.0 | 83.0±2.0 | N/A | 73.36 | 51.5 |
| MSPN (Bodnar et al., 2021b) | 88.3±10.7 | 62.6±9.3 | 62.6±9.3 | 80.4±1.2 | 79.1±1.6 | 73.7±4.0 | 52.1±3.9 |
| GTR (Huang et al., 2023b) | 86.6±1.4 | 65.2±4.6 | 75.3±0.8 | N/A | N/A | 73.1±0.8 | 79.4±0.3 |
| CWN (Bodnar et al., 2021a) | 90.0±7.4 | 62.1±9.3 | 73.4±4.4 | 84.7±1.7 | 80.3±1.9 | 73.5±4.5 | 51.0±3.1 |
| GraphSNN (Wijesinghe & Wang, 2022) | 87.3±3.1 | 61.6±2.8 | 74.1±3.2 | 76.4±1.7 | N/A | 74.8±3.5 | N/A |
| RePHINE (Immonen et al., 2023) | 87.4±6.3 | 64.9±3.7 | 72.3±1.9 | 80.9±1.9 | 79.2±1.7 | 69.4±3.8 | N/A |
| WLHN (Nikolentzos et al., 2023) | 86.3±7.4 | 65.1±2.4 | 75.9±1.9 | 79.2±2.1 | N/A | 73.4±3.7 | 49.7±3.6 |
| G3N (Wang et al., 2023b) | 89.9±8.0 | 60.0±4.8 | 75.9±2.8 | 78.6±1.9 | 79.2±1.3 | 71.0±2.2 | 45.2±2.8 |
| HTML (Li et al., 2024) | 88.9±1.8 | 66.9±4.2 | 74.9±0.3 | 78.7±0.7 | 78.8±0.6 | 71.7±0.4 | N/A |
| PathNN (Michel et al., 2023) | 90.2±4.7 | 65.8±2.7 | 75.2±3.9 | 77.5±1.6 | 78.1±2.1 | 72.6±3.3 | 50.8±4.5 |
| TopNets (Verma et al., 2024) | 92.7±1.9 | 65.7±3.6 | 73.8±1.5 | 79.1±1.2 | 78.4±0.7 | 73.1±1.8 | N/A |
| TopoTune (Papillon et al., 2025) | 86.4±6.5 | 67.2±4.8 | 72.5±3.1 | 77.6±1.1 | 77.2±0.2 | 76.3±2.7 | N/A |
| KGWL (Zhang et al., 2025b) | 82.5±5.7 | 66.2±3.5 | 72.8±2.6 | 74.9±3.5 | 78.5±1.4 | 74.4±1.9 | 51.6±3.1 |
| CCIN | **96.4±2.1** | **67.6±10.1** | **76.1±2.5** | 83.2±1.6 | **81.1±2.0** | 78.3±4.5 | **54.7±3.1** |

*Table 3.* Performance on on graph regression and classification tasks. Results are reported over 9 runs with seed 1-9. (Mean±Std)

| Method | REDDIT-B | REDDIT-M | MOLHIV |
|---|---|---|---|
| | (Accuracy) | | (ROC-AUC) |
| WLkernel(Shervashidze et al., 2011) | 81.0±3.1 | 52.5±2.1 | N/A |
| GIN(Xu et al., 2018) | 91.1±1.8 | 56.2±1.8 | 77.07±1.49 |
| RetGK(Zhang et al., 2018b) | 90.8±0.2 | 54.2±0.3 | N/A |
| HGCN(Chami et al., 2019) | 86.3±1.6 | 52.7±2.0 | 75.91±1.48 |
| GSN (Bouritsas et al., 2022) | 91.1±1.8 | 56.2±1.8 | 77.99±1.00 |
| MSPN(Bodnar et al., 2021b) | 92.7±0.9 | 57.0±2.0 | 78.25±0.31 |
| CWN (Bodnar et al., 2021a) | 93.1±1.0 | 48.2±6.6 | 78.58±0.57 |
| G3N(Wang et al., 2023b) | 89.4±2.1 | N/A | 79.00±1.34 |
| MGNN(Kanatsoulis & Ribeiro, 2024) | 92.0±1.8 | 56.1±1.6 | N/A |
| WLHN (Nikolentzos et al., 2023) | 90.7±1.9 | 55.2±1.2 | 78.41±0.31 |
| GraphSNN (Wijesinghe & Wang, 2022) | 92.7±2.0 | **57.5±1.5** | 78.51±1.72 |
| HTML(Li et al., 2024) | 90.7±0.6 | 55.9±0.4 | 78.68±0.61 |
| CCIN | **93.4±1.1** | 57.3±1.7 | **80.45±1.38** |

*Table 4.* Performance on on graph regression and classification tasks. Results are reported over 9 runs with seed 1-9. (Mean±Std)

| Model | PEPTIDES-FUNC | PEPTIDES-STRUCT |
|---|---|---|
| | (AP) ↑ | (MAE) ↓ |
| GIN (Xu et al., 2018) | 0.5498±0.0079 | 0.3547±0.0045 |
| GatedGCN (Dwivedi et al., 2022a) | 0.5498±0.0079 | 0.3547±0.0045 |
| SAN+LapPE(Kreuzer et al., 2021) | 0.6384±0.0121 | 0.2683±0.0043 |
| SAN+EdgeRWSE(Dwivedi et al., 2022a) | 0.6002±0.0048 | 0.2679±0.0015 |
| GraphFP (Luong & Singh, 2023) | 0.6267±0.0073 | 0.3137±0.0019 |
| 2-DRFWL (Zhou et al., 2023) | 0.5953±0.0048 | 0.2594±0.0038 |
| GPS (Rampášek et al., 2022) | 0.6435±0.0041 | 0.2547±0.0005 |
| CWN (Bodnar et al., 2021a) | 0.6237±0.0038 | 0.2537±0.0042 |
| GTR (Huang et al., 2023b) | 0.6351±0.0079 | 0.2568±0.0019 |
| PathNN (Michel et al., 2023) | 0.6384±0.0052 | 0.2540±0.0046 |
| EMPSN (Eijkelboom et al., 2023) | 0.6156±0.0080 | 0.2539±0.0015 |
| Subgraphormer(Bar-Shalom et al., 2024) | 0.6415±0.0052 | 0.2529±0.0020 |
| CCIN | **0.6493±0.0262** | **0.2501±0.0073** |

competitive and unifies higher-order graph message passing architectures, suggesting that its combinatorial interaction modeling effectively balances expressiveness and stability.

**Large-Scale Social and Molecular Benchmarks.** Table 3 presents the performance of CCIN on large-scale social networks and molecular graph benchmarks. CCIN also demonstrates excellent results in multi-classification tasks. On REDDIT-B, CCIN achieves 93.4 accuracy and outperforms all competitors, representing a 2.30% improvement over the classic message-passing model GIN. On the REDDIT-M dataset, CCIN maintains competitive performance compared to the baseline. On the large-scale molecular property prediction benchmark MOLHIV, CCIN achieves a ROC-AUC of 80.45, a 1.87% improvement over CWN. These results indicate that CCIN can effectively capture key information across long structural distances.

**Peptides Benchmarks.** Table 4 summarizes results on Peptides-func and Peptides-struct, which are designed to test long-range interaction (LRI) reasoning. CCIN achieves a competitive average accuracy (AP) of 0.6493 on PEPIDES-FUNC, which notably outperforms GIN (0.5498,+18.09%) and CWN (0.6237, +4.11%), and maintains high competitiveness with advanced baselines GPS and Subgraphormer. These results indicate that CCIN exhibits stable and excellent performance on molecules with large-scale graphs, long information propagation distances, and higher-order structural and topological dependencies.

**ZINC.** We next evaluate CCIN on the large-scale molecular regression benchmark ZINC, including ZINC-small (12K) and ZINC-FULL (250K). These datasets are widely

adopted to assess a model's ability to capture fine-grained chemical structures and long-range dependencies, with performance measured by mean absolute error (MAE), where lower is better. Table 5 shows that CCIN displays performance comparable to classical message-passing models. On ZINC-small without edge features, CCIN surpasses classic message-passing models such as GIN and PNA, which achieves performance comparable to recent TNNs methods as TopoTune and MGNN. When with edge features, CCIN further improves its performance and remains competitive with advanced cellular and higher-order models. On the ZINC-FULL, CCIN achieves the best reported performance among all baselines. This suggests that the proposed model enables more effective propagation of higher-order structural information across deep and wide molecular graphs.

*Table 5.* Mean Absolute Error (Mean±Std) of different methods on ZINC(12K), ZINC-FULL(250K).

| Model | ZINC-small | | ZINC-FULL |
|---|---|---|---|
| | w/o edge feats | w/i edge feats | |
| GIN (Xu et al., 2018) | 0.387±0.015 | 0.252±0.014 | N/A |
| PNA (Corso et al., 2020) | 0.320±0.032 | 0.188±0.004 | N/A |
| HIMP (Fey et al., 2020) | N/A | 0.151±0.006 | 0.036±0.002 |
| GSN (Bouritsas et al., 2022) | 0.139±0.007 | 0.108±0.018 | N/A |
| GSN (Bouritsas et al., 2022) | 0.139±0.007 | 0.115±0.012 | 0.108±0.018 |
| MPSN (Bodnar et al., 2021a) | 0.137±0.008 | 0.094±0.004 | 0.044±0.003 |
| G3N (Wang et al., 2023b) | 0.165±0.018 | 0.128±0.015 | N/A |
| $I^2$-GNN(Huang et al., 2023a) | N/A | 0.095±0.007 | 0.083±0.003 |
| TopoTune (Papillon et al., 2025) | 0.247±0.005 | 0.191±0.003 | 0.103±0.016 |
| MGNN (Kanatsoulis & Ribeiro, 2024) | 0.140±0.004 | 0.110±0.005 | 0.041±0.001 |
| CCIN | **0.125±0.003** | **0.082±0.009** | **0.033±0.004** |

# 7. Ablation Study

**Ablation of Neighbor Functions** To validate the contribution of different neighborhoods, Table 6 ablates different modules and compares the results of CCIN framework. There is some redundancy between different neighborhoods; removing some neighborhoods actually brings benefits on certain datasets. For example, on RDT-B, removing $\mathcal{C}$ (w/o-$\mathcal{C}$) achieves 91.95, improving 2.80% compared to the full version, indicating that some neighborhoods may introduce redundancy under specific data distributions, interfering with the information in the aggregation process. In contrast, CCIN achieves excellent performance using only upper and lower neighborhoods, improving performance by +4.28% and +1.17% on RDT-B and RDT-M, respectively. These results show that upper and lower neighborhoods can effectively reduce neighborhood redundancy while maintaining considerable expressive power in message aggregation, thus bringing stable and better performance.

**Ablation of Maximum Dimension** Table 7 shows the classification performance of CCIN under different maximum dimensions. The maximum dimension represents the highest-order structural level that the model can encode in

the neighborhood aggregation. With the gradual introduction of higher-order structural information, the model shows notable performance improvements on multiple datasets. When input with 1-dim structure improves performance by 10.05%, 10.76%, and 16.83% on the NCI1, NCI109, and RDT-B datasets, respectively, indicating that 1-cell structural information benefits in discriminating structures in molecular graphs and social networks. Further CCIN introduces two-dimensional information can bring relatively mild improvements. These results indicate that the appropriate introduction of higher-order information can achieve superior performance.

*Table 6.* Performance comparison of CCIN on different neighborhood function ablation. (Mean±Std)

| | PROTEINS | NCI1 | NCI109 | RDT-B | RDT-M |
|---|---|---|---|---|---|
| w/o-$\mathcal{B}$ | 75.40±4.94 | 79.46±2.23 | 77.56±2.11 | 91.43±2.43 | 54.26±2.35 |
| w/o-$\mathcal{C}$ | 74.14±3.84 | 79.44±1.87 | 77.91±1.96 | 91.95±2.35 | 54.89±1.57 |
| w/o-$\mathcal{N}_\uparrow$ | 75.41±4.15 | 72.95±3.17 | 71.78±1.35 | 85.53±3.17 | 54.98±3.73 |
| w/o-$\mathcal{N}_\downarrow$ | 72.88±4.21 | 78.37±1.66 | 77.45±1.71 | 91.65±1.13 | 55.46±2.41 |
| CCIN-Full | 74.95±5.43 | 79.56±1.63 | 78.54±1.42 | 89.15±3.61 | 55.49±1.54 |
| CCIN | **76.14±2.53** | **80.69±1.22** | **79.71±2.05** | **93.43±1.15** | **57.30±1.78** |

*Table 7.* Performance of CCIN on input dimension with 9 runs with seed 1-9. (Mean±Std)

| max dim | PROTEINS | NCI1 | NCI109 | RDT-B |
|---|---|---|---|---|
| 0 | 74.77±4.59 | 70.40±3.36 | 70.15±2.77 | 75.50±1.35 |
| 1 | 74.47±5.01 | 80.45±1.59 | 80.91±1.78 | **92.33±0.85** |
| 2 | **75.12±3.06** | **80.70±1.20** | **81.02±2.35** | 89.83±8.27 |

# 8. Conclusion

We proposed CCWL framework and investigated the expressive power on combinatorial complexes through the Weisfeiler-Lehman test refinement in different neighborhood function. Specially, combinatorial complexes unify graph, hypergraph, simplicial complex and cellular complex, we proved that the upper and lower adjacencies are sufficient to achieve the expressive power of full CCWL framework. This theoretical insight not only resolves redundancy in existing architectures but also establishes complete refinement rules for distinguishing non-isomorphic combinatorial complexes. Building on CCWL, we propose combinatorial complex isomorphism network model that instantiates the CCWL principles. Extensive experiments on synthetic and real-world benchmarks exhibit that CCIN achieves superior performance over traditional GNNs and specialized higher-order models across diverse topological domains, validating both its expressivity and generalization capability.

## Impact Statement

This work proposes a general framework for topological deep learning via the Combinatorial Complex Weisfeiler-Lehman test and its neural instantiation. It lays the groundwork for learning algorithms that reason over complex relational and geometric structures. Potential applications include biological modeling, neural simulation, physical dynamics, and social networks. The model's expressiveness may enhance prediction and structural understanding.

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

# A. Weisfeiler Lehman on Combinatorial Complexes

## A.1. Weisfeiler Lehman to Neighborhood Function

Neighborhood functions define the transitive relationships of various neighborhood relations messages on combinatorial complexes. As the Definition 3.1, for combinatorial complex $\mathcal{CC}$ and a cell $\sigma, \mathrm{rk}(\sigma) = k$, Four adjacencies as followed

- Boundary adjacent $\mathcal{B}(\sigma) = \{\tau | \tau \prec \sigma\}$. These are the set of lower-rank connected cells $(\mathrm{rk}(\tau) = k - 1)$ on their boundary., with the neighborhood specified by the boundary matrix $\mathcal{B}_r$, i.e. the vectors connected with edges $\tau$.

- Co-boundary adjacent $\mathcal{C}(\sigma) = \{\tau | \sigma \prec \tau\}$. These are the higher-dimensional cells $(\mathrm{rk}(\tau) = k + 1)$ with on their boundary. specified by the boundary matrix $\mathcal{B}_r^T$, i.e. the co-boundary cells of a node are the edges it is part of.

- Lower adjacent $\mathcal{N}_\downarrow(\sigma) = \{\tau | \exists \delta, s.t. \delta \prec \sigma \text{ and } \delta \prec \tau\}$. These are the cells of the same dimension as that share a lower dimensional cell $(\mathrm{rk}(\tau) = k - 1)$ on their boundary.

- Upper adjacent $\mathcal{N}_\uparrow(\sigma) = \{\tau | \exists \delta, s.t. \sigma \prec \delta \text{ and } \tau \prec \delta\}$. These are the cells of the same dimension as that are on co-boundary of the same higher-dimensional cell $(\mathrm{rk}(\tau) = k + 1)$.

To clarify the visualization process of the four neighborhoods, Figure A.1 (a) shows the WL test for the boundary neighborhood, where the WL test multisets aggregate information from the $(k - 1)$ cells to the $k$-cells; Figure A.1 (b) shows the WL test for the coboundary neighborhood, where the WL test multisets aggregate information from the $k$-cells to the $(k - 1)$-cells; Figure A.1 (c) shows the WL test for the lower adjacency neighborhood, where the WL test multisets of $k$-cell aggregate information from the $k$-cells from the adjacent $(k - 1)$-cells; and Figure A.1 (d) shows the WL test for the upper adjacency neighborhood, where the WL test multisets of $k$-cell aggregate information from the $(k - 1)$ cells from the adjacent $k$-cells. We formally define these as follows:

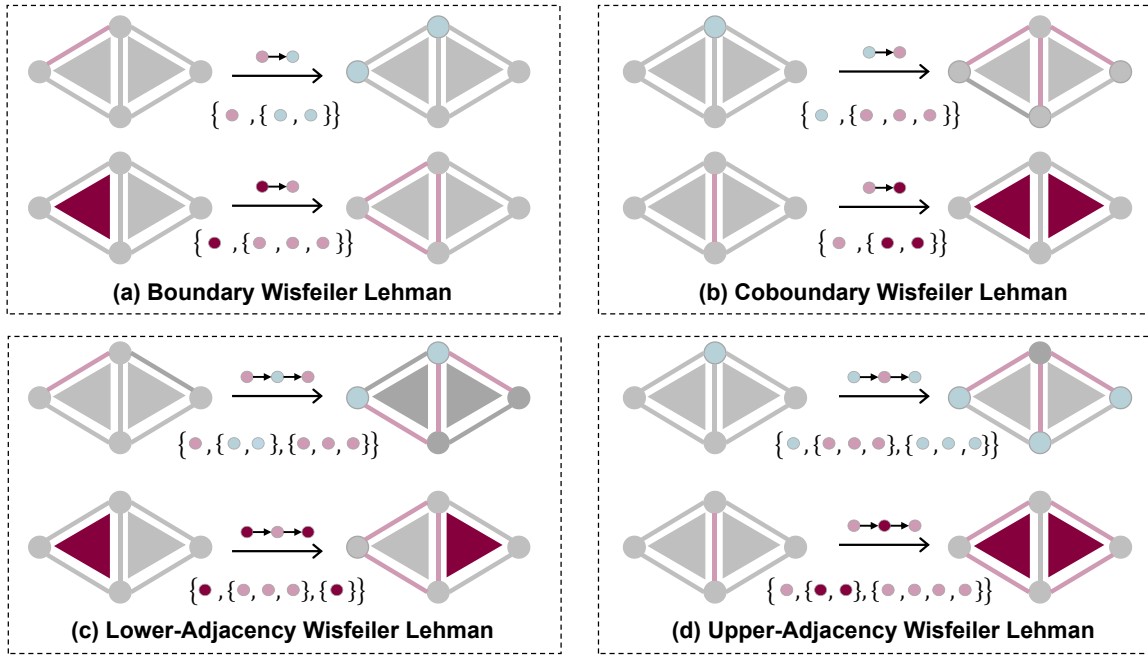

Figure A.1. Illustration of Weisfeiler-Lehman tests on four neighborhood function.

**Definition A.1** (Wisfeiler Lehman on Boundary ). *Given a cell $\sigma \in \mathcal{C}$, the boundary-based WL update is defined as $c_\mathcal{B}^{(t+1)}(\sigma) = \mathrm{HASH}\left(c^{(t)}(\sigma), \{c^{(t)}(\tau) \mid \tau \in \mathcal{B}(\sigma)\}\right)$, e.g., node • to edge •; edge • to face • (see Figure A.1 (a)).*

**Definition A.2** (Wisfeiler Lehman on Co-boundary). *The co-boundary-based WL update is defined as $c_\mathcal{C}^{(t+1)}(\sigma) = \mathrm{HASH}\left(c^{(t)}(\sigma), \{c^{(t)}(\tau) \mid \tau \in \mathcal{C}(\sigma)\}\right)$. e.g., edge • to node •; face • to edge • (see Figure A.1 (b)).*

**Definition A.3** (Wisfeiler Lehman on Lower-Adjacency). *The WL update based on lower adjacent cells is defined as* $c_{\mathcal{N}_\downarrow}^{(t+1)}(\sigma) = \text{HASH}\left(c^{(t)}(\sigma), \left\{(c^{(t)}(\tau), c^{(t)}(\delta))\right\}\right)$. *where* $\tau \in \mathcal{N}_\downarrow(\sigma), \delta \in \mathcal{B}(\sigma, \tau)$, *e.g., edge • to edge • with the bridge node •; face • to face • with the bridge edge • (see Figure A.1 (c)).*

**Definition A.4** (Wisfeiler Lehman on Upper-Adjacency). *The WL update based on upper adjacent cells is defined as* $c_{\mathcal{N}_\uparrow}^{(t+1)}(\sigma) = \text{HASH}\left(c^{(t)}(\sigma), \left\{(c^{(t)}(\tau), c^{(t)}(\delta)) \mid \right\}\right)$. *where* $\tau \in \mathcal{N}_\uparrow(\sigma), \delta \in \mathcal{C}(\sigma, \tau)$, *e.g., node • to node • with the bridge edge •; edge • to edge • with the bridge face • (see Figure A.1 (d)).*

### A.2. Set Type Relations: Graph and Hypergraph

A graph models pairwise relations via nodes and edges, while a hypergraph generalizes this notion by allowing hyperedges to connect multiple nodes; when a hyperedge has size two, it degenerates to a standard graph edge. Hypergraph neural networks typically employ a two-stage message passing scheme: node-to-hyperedge, and hyperedge-to-node, which is analogous to information propagation in graph neural networks. Recent work interprets this mechanism from a $k$-hop neighborhood perspective. For example, (Feng et al., 2022) studied kernel-based message passing, (Xie et al., 2025) proposed KHGNN to capture long-range interactions, and (Paolino et al., 2024) introduced an $r$-loopy Weisfeiler-Leman framework. These approaches aggregate information along multi-hop paths, and under certain conditions, message passing in graphs and hypergraphs can be shown to be equivalent (Chen et al., 2025). (Zhang et al., 2025b) introduced the two-stage update operations in hypergraph neural networks as

$$c_e^{(t)} = \text{HASH}(c_e^{(t-1)}, \{\{c_u^{(t-1)}, u \in \mathcal{N}_v(e)\}\}), \quad c_v^{(t)} = \text{HASH}(c_v^{(t-1)}, \{\{c_u^{(t-1)}, u \in \mathcal{N}_e(v)\}\}), \tag{7}$$

According to definition 3.2, we can conclude that the message passing as source nodes $v$ up to the hyperedges $e$, then back to target nodes $v$, which are structurally equivalent to the upper adjacent $\mathcal{N}_\uparrow$. However, the hypergraph only considers message passing between the source node and the target node during the message passing process, and does not fully consider the message passing process between edges. Inspired by this equivalence, we propose a unified two-stage message passing framework that incorporates four types of neighborhood relations on graph and hypergraph. In case of the scale of hyperedges degrading to 2, the hypergraphs can be generalized into graphs. Therefore, we consider unified concepts from the Hypergraph-Weisfeiler-Leman test. In particular, a node's neighborhood $\mathcal{N}(v)$ can be interpreted through upper-adjacency relations, nodes-to-edge and edge-to-node propagation can be formalized as

$$c_v^{(t+1)} = \text{HASH}\left(c^{(t)}v, \left\{c^{(t)}\tau \mid \tau \in \mathcal{N}(v)\right\}\right) \Rightarrow \quad c_{\mathcal{N}_\uparrow(v)}^{(t+1)} = \text{HASH}\left(c^{(t)}(v), \left\{(c^{(t)}(\tau), c^{(t)}(\delta))\right\}\right), \tag{8}$$

where $\tau$ denotes a hyperedge incident to node $v$, $\tau \in N_\uparrow(v), \delta \in \mathcal{C}(v, \tau)$, and $\delta$ refers to a neighboring node. Similarly, hyperedges-to-hyperedges propagation via hyperedges-to-node and nodes-to-hyperedges steps can be expressed as:

$$c_e^{(t+1)} = \text{HASH}\left(c^{(t)}e, \left\{c^{(t)}\tau \mid \tau \in \mathcal{E}(e)\right\}\right) \Rightarrow \quad c_{\mathcal{N}_\downarrow(e)}^{(t+1)} = \text{HASH}\left(c^{(t)}(e), \left\{(c^{(t)}(\tau), c^{(t)}(\delta))\right\}\right), \tag{9}$$

where $\tau \in N_\downarrow(e), \delta \in \mathcal{B}(e, \tau)$, $\mathcal{E}(e)$ denotes the set of hyperedges adjacent to $e$ via a shared node, which offers a view of information propagation in graphs and hypergraphs. We summarize these neighborhood relations in Table A.1.

*Table A.1.* Neighborhood relations for nodes and hyperedges in a hypergraph.

|  | Boundary | Co-Boundary | Lower Adjacency | Upper Adjacency |
|---|---|---|---|---|
| Nodes $v$ | N/A | Hyperedges $e$ incident to node $v$ | N/A | Nodes sharing hyperedges incident to $v$ |
| Hyperedges $e$ | Nodes $v$ incident to hyperedge $e$ | N/A | Hyperedges sharing nodes incident to $e$ | N/A |

**Case 1**: Let $\mathcal{G}$ be a graph and let $e_{12} = \{v_1, v_2\}$ be a 1-cell (an edge). Specifically, the boundary of $e$ by $\partial(e_{12}) = \{v_1, v_2\}$, and the co-boundary of a node $v_1$ by $\delta(v_1) = \{e' \in \mathcal{H} \mid e' \in \{e_{12}, e_{14}\}\}$, node $v_2$ by $\delta(v_2) = \{e' \in \mathcal{H} \mid e' \in \{e_{12}, e_{24}, e_{23}\}\}$. we consider the boundary and co-boundary as special cases of adjacency-WL, as followed,

(i) For a node $v_1$, the co-boundary update $c_\mathcal{C}^{(t+1)}(v_1) = \text{HASH}\left(c^{(t)}(v_1), \{\{c^{(t)}(e') \mid e' \in \delta(v) = \{e_{12}, e_{14}\}\}\}\right)$ aggregates exactly the colors of 1-cells that are pairwise lower adjacent via the boundary bridge $v_1$.

(ii) For an edge $e_{12} = \{v_1, v_2\}$, the boundary update $c_\mathcal{B}^{(t+1)}(e_{12}) = \text{HASH}\left(c^{(t)}(e_{12}), \{\{c^{(t)}(u) \mid u \in \partial(e_{12}) = \{v_1, v_2\}\}\}\right)$ aggregates exactly the colors of the two 0-cells that are upper adjacent witnessed by the co-boundary $e_{12}$.

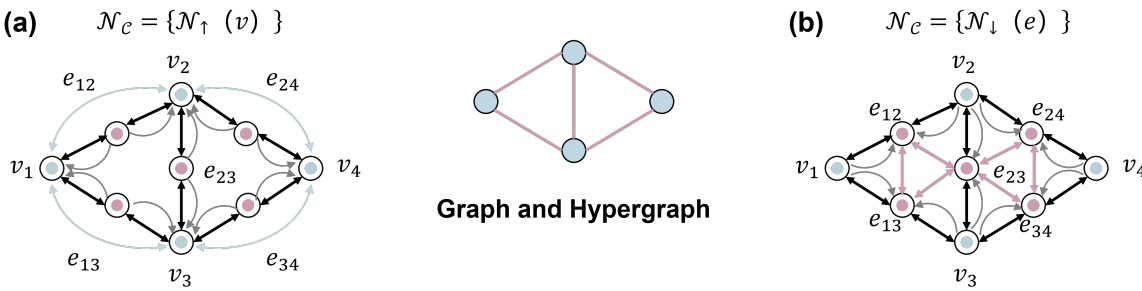

*Figure A.2.* Illustration of message aggregation on graph, hypergraph.

*Proof.* As illustrated in Figure A.2, in both graphs and hypergraphs, message passing from a node to hyperedge and back to its connected node corresponds to propagation within the upper neighborhood, whereas message passing from a hyperedge to a node and back to its connected hyperedge corresponds to propagation within the lower neighborhood.

For instance, the edge $e_{12} = \{v_1, v_2\}$, we can obtain that, (i) By definition of co-boundary, $\delta(v_1)$ is the set of incident edges $e'$ with $v_1 \in \partial(e')$. Hence two edges $e_{12}, e_{14} \in \delta(v_1)$ satisfy $v_1 \in \partial(e_{12}) \cap \partial(e_{14})$, i.e., $e_{12} \in \mathcal{N}_\downarrow(e_{14})$ with boundary bridge $v_1 \in \mathcal{B}(e_{12}, e_{14})$. Therefore the multiset $\{\{c^{(t)}(e') \mid e' \in \delta(v_1)\}\}$ coincides with the multiset of colors of lower-adjacent 1-cells connected through the bridge $v_1$. (ii) For $e_{12} = \{v_1, v_2\}$, $\partial(e_{12}) = \{v_1, v_2\}$ by the definition of boundary. Moreover $v_1$ and $v_2$ are upper adjacent since $\{v_1, v_2\} \subseteq \partial(e_{12})$, and $e \in \mathcal{C}(v_1, v_2)$. Thus the multiset $\{\{c^{(t)}(v_1), c^{(t)}(v_2)\}\}$ is exactly the multiset of node colors that is coupled by upper adjacency witnessed by the $e_{12}$. $\square$

**Case 2**. Let $\mathcal{H} = (V, \mathcal{E})$ be a hypergraph. Consider a hyperedge $e = \{v_1, v_2, \ldots, v_k\} \in \mathcal{E}$ with $k \geq 2$. We view $e$ as a generalized 1-cell whose boundary is the node set $\partial(e) = \{v_1, \ldots, v_k\}$. For any node $v \in V$, its co-boundary is defined as $\delta(v) = \{e' \in \mathcal{E} \mid v \in e'\}$. Under the standard projection (or clique expansion) of a hypergraph $\mathcal{G}' = \text{Proj}(\mathcal{H})$ (Wen & Yu, 2025), each hyperedge $e = \{v_1, \ldots, v_k\}$ is mapped to a set of pairwise connections $\{(v_i, v_j) \mid 1 \leq i < j \leq k\}$, i.e., every pair of nodes in the hyperedge is connected by an edge in the projected graph $\mathcal{G}'$. Therefore, the projected graph $\mathcal{G}'$ reduces all hyperedges to edges of cardinality 2. Consequently, Case 1 applies to $\mathcal{G}'$, where the induced incidence adjacencies align precisely with the upper- and lower-adjacency relations defined earlier. More generally, in the original hypergraph, a hyperedge $e = \{v_1, \ldots, v_k\}$ extends this construction. Its boundary $\partial(e)$ induces upper adjacency among all node pairs contained in $e$, and aggregating over these nodes recovers the boundary-type update. Similarly, the co-boundary of a node $v$ (the set of hyperedges containing $v$), which induces lower adjacency among hyperedges that intersect at $v$.

Therefore, these cases on graph and hypergraph illustrate a broader condition: boundary and co-boundary aggregations can be expressed as same-dimensional upper- and lower-adjacency aggregations, without altering the multisets used in the hash functions. This observation justifies the set-type substructure of combinatorial complex weisfeiler-leman update process.

### A.3. Part-type Relations: Simplicial and Cellular Complex

Simplicial and cellular complexes naturally admit boundary and co-boundary operators, which encode cross-dimensional incidence relations between cells. Accordingly, message passing on such complexes has traditionally been formulated in terms of downward (boundary) and upward (co-boundary) maps. An alternative but closely related viewpoint is to describe these incidence relations via adjacency relations among cells of the same dimension. Two $k$-cells are said to be lower-adjacent if they share a common $(k-1)$-cell, and upper-adjacent if they share a common $(k+1)$-cell. In details, boundary- and co-boundary-based information can be encoded through same-dimensional lower and upper adjacency relations augmented with appropriate bridge cells. From the perspective of Weisfeiler-Leman refinement, prior works (Michel et al., 2023; Bodnar et al., 2021a;b) have shown that, for simplicial and cellular complexes, restricting the update rule to boundary information and upper-adjacency neighborhoods already suffices to match the expressive power of more general WL schemes. Specifically, the refinement rule $\text{HASH}\left(c^{(t)}(\sigma), c_{\mathcal{B}}^{(t)}(\sigma), c_{\mathcal{C}}^{(t)}(\sigma), c_{\mathcal{N}_\downarrow}^{(t)}(\sigma), c_{\mathcal{N}_\uparrow}^{(t)}(\sigma)\right)$ in distinguishing non-isomorphic simplicial or cellular complexes. However, boundary and shared-boundary relations are cross-dimensional. For instance, in a graph viewed as a 1-dimensional complex, the boundary of an edge is a pair of nodes, and in a simplicial complex, the boundary of a face consists of edges. In both cases, the boundary operator maps a $k$-cell to a collection of $(k-1)$-cells. Even shared-boundary relations between two $k$-cells (e.g., two faces sharing an edge) are mediated by a

lower-dimensional bridge cell. In contrast, the Weisfeiler-Leman refinement operates within a fixed dimension, updating and relabeling cells of the same type (nodes with nodes, edges with edges, faces with faces) at each iteration.

Moreover, the lower- and upper-adjacency color multisets heave encoded the cross-dimensional incidence information. From lower adjacent , for each lower-adjacent cell $\tau \in \mathcal{N}_\downarrow(\sigma)$, the bridge cell $\delta \in \mathcal{B}(\sigma, \tau) = \partial(\sigma) \cap \partial(\tau)$ is a shared boundary cell of $\sigma$ and $\tau$. Thus, each element $(c_\tau, c_\delta)$ explicitly contains the color of a boundary cell of $\sigma$. Similarly, from the upper adjacent, for each upper-adjacent cell $\tau \in \mathcal{N}_\uparrow(\sigma)$, the bridge cell $\delta \in \mathcal{C}(\sigma, \tau) = \delta(\sigma) \cap \delta(\tau)$ is a shared co-face of $\sigma$ and $\tau$, and the token $(c_\tau, c_\delta)$ explicitly contains the color of a co-boundary cell of $\sigma$. the boundary-type multiset $\{c(\zeta) \mid \zeta \in \partial(\sigma)\}$ can be functionally recovered from $c_{\mathcal{N}_\downarrow}(\sigma)$ by projecting onto the second component of the tokens, and the co-boundary-type multiset $\{c(\gamma) \mid \gamma \in \delta(\sigma)\}$ can be recovered from $c_{\mathcal{N}_\uparrow}(\sigma)$. Therefore, lower- and upper-adjacency WL updates do not discard boundary or co-boundary information; instead, they encode such cross-dimensional incidence relations within same-dimensional neighborhoods. This observation motivates our design choice: using upper and lower adjacency yields a unified, same-dimensional CCWL scheme that faithfully captures boundary and co-boundary aggregations while avoiding explicit cross-dimensional message passing. As a result, the resulting refinement procedure is conceptually simpler and more readily extensible to hypergraphs and general combinatorial complexes.

**(a)** $\quad \mathcal{N}_\mathcal{C} = \{\mathcal{N}_\uparrow(\sigma)\}$          **(b)** $\quad \mathcal{N}_\mathcal{C} = \{\mathcal{N}_\downarrow(\sigma)\}$

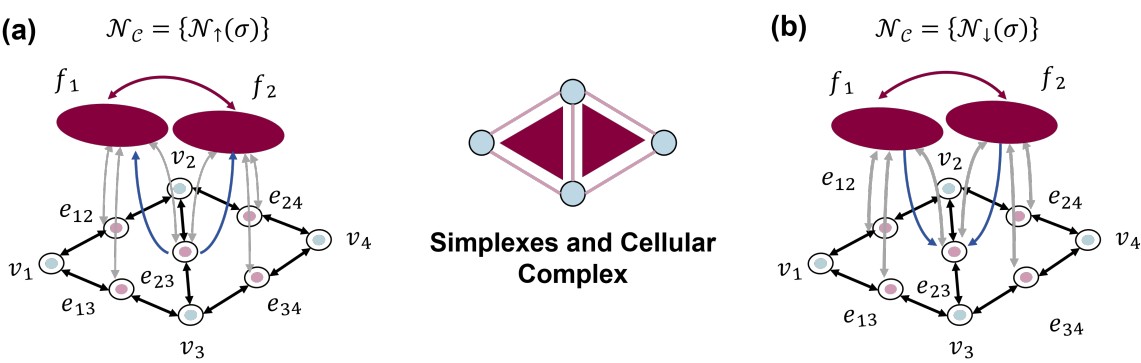

*Figure A.3.* Illustration of message aggregation on simplicial complex, cellular complex.

**Case 3**. Consider two adjacent 2-simplices (faces) $f_1 = \{v_1, v_2, v_3\}$, $f_2 = \{v_2, v_3, v_4\}$, which share the common edge $e_{23} = \{v_2, v_3\}$. The edge sets are $\partial(f_1) = \{e_{12}, e_{23}, e_{13}\}, \partial(f_2) = \{e_{23}, e_{24}, e_{34}\}$, where $e_{12} = \{v_1, v_2\}$, $e_{13} = \{v_1, v_3\}$, $e_{24} = \{v_2, v_4\}$, and $e_{34} = \{v_3, v_4\}$. Its four neighborhood relations are followed as

- Boundary: $\partial(f_1) = \{e_{12}, e_{23}, e_{13}\}, \partial(f_2) = \{e_{23}, e_{24}, e_{34}\}, \partial(e_{ij}) = \{v_i, v_j\}$,

- Co-boundary: $\delta(e_{12}) = \delta(e_{13}) = \{f_1\}, \delta(e_{24}) = \delta(e_{34}) = \{f_2\}, \delta(e_{23}) = \{f_1, f_2\}$.

- Lower adjacency: Two $k$-cells $\sigma, \tau$ are lower-adjacent if they share a common boundary $(k{-}1)$-cell, i.e., $\partial(\sigma) \cap \partial(\tau) \neq \emptyset$. In this example: (1) Nodes (0-cells) have no lower adjacency since they have no boundary. (2) Edges (1-cells) are lower-adjacent if they share a node, e.g., $e_{12} \in \mathcal{N}_\downarrow(e_{23})$ (share $v_2$), $e_{23} \in \mathcal{N}_\downarrow(e_{34})$ (share $v_3$). (3) Faces (2-cells) are lower-adjacent since $\partial(f_1) \cap \partial(f_2) = \{e_{23}\} \neq \emptyset$, hence $f_1 \in \mathcal{N}_\downarrow(f_2)$, with bridge cell $e_{23}$.

- Upper adjacency. Two $k$-cells $\sigma, \tau$ are upper-adjacent if they share a common co-boundary $(k{+}1)$-cell, i.e., $\delta(\sigma) \cap \delta(\tau) \neq \emptyset$. In this example: (1) Nodes (0-cells) are upper-adjacent if they belong to a common edge, e.g., $v_2 \in \mathcal{N}_\uparrow(v_3)$ (common co-face $e_{23}$), $v_1 \in \mathcal{N}_\uparrow(v_2)$ (common co-face $e_{12}$). (2) Edges (1-cells) are upper-adjacent if they belong to a common face, e.g., $e_{12} \in \mathcal{N}_\uparrow(e_{13})$ (co-face $f_1$), $e_{23} \in \mathcal{N}_\uparrow(e_{24})$ (co-face $f_2$). (3) Faces (2-cells) have no upper adjacency since there is no 3-cell.

*Proof.* By definition, boundary $\mathcal{B}(\sigma, \tau) := \partial(\sigma) \cap \partial(\tau)$ and co-boundary $\mathcal{C}(\sigma, \tau) := \delta(\sigma) \cap \delta(\tau)$ relations are cross-dimensional: the operator $\partial$ maps $k$-cells to $(k{-}1)$-cells, while operator $\delta$ maps $k$-cells to $(k{+}1)$-cells. Lower and upper adjacency, in contrast, relate cells of the same dimension and are witnessed by shared boundary or co-boundary cells.

As illustrated in Figure. A.3, we observe that the shared edge $e_{23}$ serves as the bridge cell witnessing both the lower adjacency between $f_1$ and $f_2$ and the co-boundary relations $\delta(e_{23}) = \{f_1, f_2\}$. we conclude that:

(1) The boundary of the shared edge is $\partial(e_{23}) = \{v_2, v_3\}$. Its lower-adjacent edges include, $e_{12}, e_{24}$ (sharing $v_2$) and $e_{13}, e_{34}$ (sharing $v_3$). Here, $\mathcal{B}(e_{23}, e_{12}) = \{v_2\}$, $\mathcal{B}(e_{23}, e_{24}) = \{v_2\}$, $\mathcal{B}(e_{23}, e_{13}) = \{v_3\}$, $\mathcal{B}(e_{23}, e_{34}) = \{v_3\}$, and therefore

$$\bigcup_{\tau \in \mathcal{N}_\downarrow(e_{23})} \mathcal{B}(e_{23}, \tau) = \{v_2, v_3\} = \partial(e_{23}). \tag{10}$$

(2) The co-boundary of the shared edge is $\delta(e_{23}) = \{f_1, f_2\}$, where $e_{23}$ is upper-adjacent to $e_{12}$ and $e_{13}$ via the co-face $f_1$, and to $e_{24}$ and $e_{34}$ via the common co-face $f_2$. $\mathcal{C}(e_{23}, e_{12}) = \{f_1\}$, $\mathcal{C}(e_{23}, e_{13}) = \{f_1\}$, $\mathcal{C}(e_{23}, e_{24}) = \{f_2\}$, $\mathcal{C}(e_{23}, e_{34}) = \{f_2\}$. Thus,

$$\bigcup_{\tau \in \mathcal{N}_\uparrow(e_{23})} \mathcal{C}(e_{23}, \tau) = \{f_1, f_2\} = \delta(e_{23}). \tag{11}$$

(3) For two faces $f_1, f_2$, their adjacency exposes shared boundary information. The faces satisfy $\partial(f_1) \cap \partial(f_2) = \{e_{23}\}$, so $f_1 \sim_\downarrow f_2$ with bridge cell $e_{23}$, i.e., $\mathcal{B}(f_1, f_2) = \{e_{23}\}$. Therefore, lower adjacency between faces captures their shared boundary. Note that in this example $\partial(f_1) = \{e_{12}, e_{23}, e_{13}\}$ contains edges that are not shared with any other 2-cell (namely $e_{12}, e_{13}$), so they cannot be recovered from $\mathcal{B}(f_1, \tau)$ over $\tau \in \mathcal{N}_\downarrow(f_1)$; rather, they are exposed by the cross-dimensional boundary operator itself (e.g., this condition holds in a tetrahedron).

This case shows that boundary and co-boundary are cross-dimensional incidence relations, whereas lower and upper adjacency relate same-dimensional cells and expose incidence information via bridge cells $\mathcal{B}(\sigma, \tau)$ and $\mathcal{C}(\sigma, \tau)$. In particular, for the shared edge $e_{23}$, the equalities

$$\partial(e_{23}) = \bigcup_{\tau \in \mathcal{N}_\downarrow(e_{23})} \mathcal{B}(e_{23}, \tau), \qquad \delta(e_{23}) = \bigcup_{\tau \in \mathcal{N}_\uparrow(e_{23})} \mathcal{C}(e_{23}, \tau) \tag{12}$$

hold exactly, demonstrating how same-dimensional adjacency (augmented with bridge cells) can faithfully encode cross-dimensional neighborhoods used in WL refinement.

$\square$

**Case 4**. Consider the 3-simplex (tetrahedron) $\sigma = \{v_1, v_2, v_3, v_4\}$. Its 2-faces are $f_1 = \{v_1, v_2, v_3\}$, $f_2 = \{v_1, v_2, v_4\}$, $f_3 = \{v_1, v_3, v_4\}$, $f_4 = \{v_2, v_3, v_4\}$. The 1-cells (edges) are $e_{12}, e_{13}, e_{14}, e_{23}, e_{24}, e_{34}$, and the 0-cells are $v_1, v_2, v_3, v_4$.

- Boundary operator satisfies $\partial(\sigma) = \{f_1, f_2, f_3, f_4\}$, $\partial(f_1) = \{e_{12}, e_{13}, e_{23}\}, \partial(f_2) = \{e_{12}, e_{14}, e_{24}\}$, $\partial(f_3) = \{e_{13}, e_{14}, e_{34}\}$, $\partial(f_4) = \{e_{23}, e_{24}, e_{34}\}$, $\partial(e_{ij}) = \{v_i, v_j\}$.

- Co-boundary includes the nodes: $\delta(v_1) = \{e_{12}, e_{13}, e_{14}\}$, $\delta(v_2) = \{e_{12}, e_{23}, e_{24}\}$, $\delta(v_3) = \{e_{13}, e_{23}, e_{34}\}$, $\delta(v_4) = \{e_{14}, e_{24}, e_{34}\}$, and edges $e_{ij}$: $\delta(e_{ij}) = \{f_i, f_j\}$, but when $i = 1, 2, 3, 4$, $\delta(f_i) = \{\sigma\}$ $\delta(\sigma) = \emptyset$.

*Proof.* For same-dimensional cells $\eta, \zeta$, define bridge sets, boundary $\mathcal{B}(\eta, \zeta) := \partial(\eta) \cap \partial(\zeta)$, and co-boundary $\mathcal{C}(\eta, \zeta) := \delta(\eta) \cap \delta(\zeta)$, so that $\zeta \in \mathcal{N}_\downarrow(\eta)$ iff $\mathcal{B}(\eta, \zeta) \neq \emptyset$, and $\zeta \in \mathcal{N}_\uparrow(\eta)$ iff $\mathcal{C}(\eta, \zeta) \neq \emptyset$. Here, we need to prove that for every cell $\eta$ of dimension $k \in \{1, 2\}$, the following equalities hold:

$$\partial(\eta) = \bigcup_{\zeta \in \mathcal{N}_\downarrow(\eta)} \mathcal{B}(\eta, \zeta), \qquad \delta(\eta) = \bigcup_{\zeta \in \mathcal{N}_\uparrow(\eta)} \mathcal{C}(\eta, \zeta). \tag{13}$$

**(1) Edges ($k = 1$).** For an edge $\eta = e_{ij}$, $e_{ij} = \partial f_i \cap \partial f_j$. Similar to Equation 12 in Case 3, $e_{ij}$, we can conclude that

- For the boundary $\partial(e_{ij}) = \{v_i, v_j\}$, each node has degree 3 in a tetrahedron, there exists an edge $\zeta = e_{i\ell} \neq e_{ij}$ sharing $v_i$, hence $\mathcal{B}(e_{ij}, e_{i\ell}) = \{v_i\}$, and similarly an edge $\zeta' = e_{jm} \neq e_{ij}$ sharing $v_j$, hence $\mathcal{B}(e_{ij}, e_{jm}) = \{v_j\}$. Therefore $\partial(e_{ij}) \subseteq \bigcup_{\zeta \in \mathcal{N}_\downarrow(e_{ij})} \mathcal{B}(e_{ij}, \zeta)$, while the reverse inclusion holds trivially because every bridge is a boundary node of $e_{ij}$. This proves the boundary equality for all edges.

- For the co-boundary, each edge $e_{ij}$ lies in exactly two faces; for example $\delta(e_{12}) = \{f_1, f_2\}$. Choose $\zeta$ to be another edge in $f_1$ (e.g. $e_{13}$), then $\mathcal{C}(e_{12}, e_{13}) = \{f_1\}$, and choose $\zeta'$ to be another edge in $f_2$ (e.g. $e_{14}$), then $\mathcal{C}(e_{12}, e_{14}) = \{f_2\}$. Thus both incident faces of $e_{12}$ appear in the union over $\mathcal{C}$-bridges. The same argument applies to any edge, proving the co-boundary equality for all edges.

**(2) Faces ($k = 2$).** For instance, the face $\eta = f_1 = \{v_1, v_2, v_3\}$, we can obtain that

- Its boundary edges are $\partial(f_1) = \{e_{12}, e_{13}, e_{23}\}$. Each boundary edge is shared with exactly one other face:$\mathcal{B}(f_1, f_2) = \{e_{12}\}$, $\mathcal{B}(f_1, f_3) = \{e_{13}\}$, $\mathcal{B}(f_1, f_4) = \{e_{23}\}$. Hence $\bigcup_{\zeta \in \mathcal{N}_\downarrow(f_1)} \mathcal{B}(f_1, \zeta) = \{e_{12}, e_{13}, e_{23}\} = \partial(f_1)$. The same holds for any face by symmetry.

- For the co-boundary, each face has the unique co-face $\sigma$, i.e. $\delta(f_i) = \{\sigma\}$. Moreover, for any distinct faces $f_i \neq f_j$, we have $\mathcal{C}(f_i, f_j) = \{\sigma\}$, hence $f_j \in \mathcal{N}_\uparrow(f_i)$ and $\bigcup_{\zeta \in \mathcal{N}_\uparrow(f_i)} \mathcal{C}(f_i, \zeta) = \{\sigma\} = \delta(f_i)$.

*Table A.2.* Neighborhood relations in a 3-cell complex (here 0,1,2,3-cells represent nodes,edges,faces,tetrahedra, respectively).

| Cell $\sigma$ | Boundary $\partial(\sigma)$ | Co-Boundary $\delta(\sigma)$ | Lower Adjacency $\mathcal{N}_\downarrow(\sigma)$ | Upper Adjacency $\mathcal{N}_\uparrow(\sigma)$ |
|---|---|---|---|---|
| Nodes (0-cells) $v$ | N/A | Edges incident to $v$ | N/A | Nodes sharing an edge with $v$ |
| Edges (1-cells) $e$ | nodes (nodes) of $e$ | Faces incident to $e$ | Edges sharing a node with $e$ | Edges sharing a face with $e$ |
| Faces (2-cells) $f$ | Boundary edges of $f$ | Tetrahedral incident to $f$ | Faces sharing an edge with $f$ | Faces sharing a tetrahedron with $f$ |
| Tetrahedral (3-cells) $t$ | Boundary faces of $t$ | N/A | Tetrahedral sharing a face with $t$ | N/A |

**Remark** Lower and upper adjacency expose exactly the shared incidence structure among same-dimensional cells; they are not intended to enumerate private boundary elements of isolated or top-dimensional cells. We summarize the neighborhood relation in Table A.2. There are two special cases in which the boundary equality in (13) does not apply. First, nodes have no boundary, the boundary equality is vacuous for $k = 0$. Second, although the co-boundary equality holds trivially for the unique top-dimensional cell $\sigma$ since $\delta(\sigma) = \emptyset$, the boundary equality fails in this case. Indeed, because $\sigma$ is the only 3-cell in the complex, $\mathcal{N}_\downarrow(\sigma) = \emptyset$, and consequently $\bigcup_{\zeta \in \mathcal{N}_\downarrow(\sigma)} \mathcal{B}(\sigma, \zeta) = \emptyset \neq \partial(\sigma)$. Therefore, the boundary equality for the top-dimensional cell holds only when additional 3-cells sharing boundary faces with $\sigma$ are present.

- Nodes (0-cells). Nodes have no boundary by definition as $\partial(v) = \emptyset$ and the boundary recovery identity is vacuous for $k = 0$. In contrast, the co-boundary recovery identity holds exactly. Indeed, for a node $v_i$, its co-boundary is $\delta(v_i) = \{e_{ij} \mid j \neq i\}$. For any neighbor $v_j \in \mathcal{N}_\uparrow(v_i)$, the bridge set satisfies $\mathcal{C}(v_i, v_j) = \delta(v_i) \cap \delta(v_j) = \{e_{ij}\}$. Each edge $e_{ij} \in \delta(v_i)$ appears in exactly one bridge set with some upper-adjacent node $v_j$, and we obtain $\delta(v_i) = \bigcup_{v_j \in \mathcal{N}_\uparrow(v_i)} \mathcal{C}(v_i, v_j)$. For 0-cells, the co-boundary can be recovered from upper adjacency together with bridge cells.

- Tetrahedral (3-cell). For the tetrahedron $\sigma$, there is no higher-dimensional cell, and thus $\delta(\sigma) = \emptyset$ , $\mathcal{N}_\uparrow(\sigma) = \emptyset$, so the co-boundary recovery identity holds trivially. However, $\sigma$ is the unique 3-cell in the complex, and consequently $\mathcal{N}_\downarrow(\sigma) = \emptyset$. Although the boundary of $\sigma$ is nonempty, $\partial(\sigma) = \{f_1, f_2, f_3, f_4\}$, none of these boundary faces is shared with another 3-cell. Hence, $\bigcup_{\zeta \in \mathcal{N}_\downarrow(\sigma)} \mathcal{B}(\sigma, \zeta) = \emptyset \neq \partial(\sigma)$. Thus, the boundary recovery identity does not hold for the unique top-dimensional cell unless additional 3-cells are present that share boundary faces with $\sigma$.

$\square$

Existing research has only focused on geometric structural relationships within two dimensions (faces), due to the lack of three-dimensional volumetric datasets. This direction requires further investigation.

**Case 5** (Combinatorial complex). Let $\mathcal{CC} = (\mathcal{S}, \mathcal{C}, \mathrm{rk})$ be a combinatorial complex as in Definition X, and let $\sigma \in \mathcal{C}$ be a $k$-cell, i.e., $\mathrm{rk}(\sigma) = k$. We define the boundary and co-boundary operators purely in terms of rank: $\partial(\sigma) := \{\eta \in \mathcal{C} \mid \eta \subset \sigma, \mathrm{rk}(\eta) = k - 1\}$, $\delta(\sigma) := \{\gamma \in \mathcal{C} \mid \sigma \subset \gamma, \mathrm{rk}(\gamma) = k + 1\}$. For the Same-dimensional adjacencies and bridge sets, such as the two $k$-cells $\sigma, \tau \in \mathcal{C}$, define the bridge sets as $\mathcal{B}(\sigma, \tau) := \partial(\sigma) \cap \partial(\tau)$, $\mathcal{C}(\sigma, \tau) := \delta(\sigma) \cap \delta(\tau)$. The lower- and upper-adjacency neighborhoods of $\sigma$ are $\mathcal{N}_\downarrow(\sigma) := \{\tau \in \mathcal{C} \mid \mathrm{rk}(\tau) = k, \tau \neq \sigma, \mathcal{B}(\sigma, \tau) \neq \emptyset\}$, $\mathcal{N}_\uparrow(\sigma) := \{\tau \in \mathcal{C} \mid \mathrm{rk}(\tau) = k, \tau \neq \sigma, \mathcal{C}(\sigma, \tau) \neq \emptyset\}$.

**Coverage condition** We define that $\sigma$ is boundary-covered if every boundary cell of $\sigma$ is shared with some other $k$-cell: $\forall \eta \in \partial(\sigma), \exists \tau \in \mathcal{C} : \mathrm{rk}(\tau) = k, \tau \neq \sigma, \eta \in \partial(\sigma) \cap \partial(\tau)$. For example, for a face $f_1$, there exists a face $f_2(e.g., \mathrm{rk}(f_1) = \mathrm{rk}(f_2))$ such that $e_{23} \in \partial(f_1) \cap \partial(f_2)$. Similarly, $\sigma$ is co-boundary-covered if every co-boundary cell of $\sigma$ is shared with some other $k$-cell: $\forall \gamma \in \delta(\sigma) \exists \tau \in \mathcal{C} : \mathrm{rk}(\tau) = k, \tau \neq \sigma, \gamma \in \delta(\sigma) \cap \delta(\tau)$.

From the bridge sets in CC, let $\sigma$ be a $k$-cell in a combinatorial complex. If $\sigma$ is boundary-covered, then $\partial(\sigma) = \bigcup_{\tau \in \mathcal{N}_\downarrow(\sigma)} \mathcal{B}(\sigma, \tau)$. If $\sigma$ is co-boundary-covered, then $\delta(\sigma) = \bigcup_{\tau \in \mathcal{N}_\uparrow(\sigma)} \mathcal{C}(\sigma, \tau)$.

*Proof.* We prove the boundary statement ($\subseteq$) Let $\eta \in \partial(\sigma)$. Since $\sigma$ is boundary-covered, there exists a $k$-cell $\tau \neq \sigma$ such that $\eta \in \partial(\sigma) \cap \partial(\tau) = \mathcal{B}(\sigma, \tau)$. Hence $\mathcal{B}(\sigma, \tau) \neq \emptyset$, so by definition $\tau \in \mathcal{N}_\downarrow(\sigma)$, and thus $\eta \in \bigcup_{\tau \in \mathcal{N}_\downarrow(\sigma)} \mathcal{B}(\sigma, \tau)$. ($\supseteq$) For any $\tau \in \mathcal{N}_\downarrow(\sigma)$, we have $\mathcal{B}(\sigma, \tau) = \partial(\sigma) \cap \partial(\tau) \subseteq \partial(\sigma)$, hence the union of such sets is contained in $\partial(\sigma)$. Combining the two inclusions yields $\partial(\sigma) = \bigcup_{\tau \in \mathcal{N}_\downarrow(\sigma)} \mathcal{B}(\sigma, \tau)$. The co-boundary follows the same argument with $\delta$ and $\mathcal{C}(\cdot, \cdot)$. $\square$

## B. Proof Theorem

### B.1. Combinatorial Complex Weisfeiler Lehman

**Definition B.1.** *A coloring is a rule that assigns to every cell $\sigma$ of a combinatorial complex $\mathcal{CC}$ a color chosen from a fixed palette. For a given complex $\mathcal{CC}$ and a cell $\sigma \in \mathcal{CC}$, we write $c_\sigma^{\mathcal{CC}}$ for the color assigned to $\sigma$ by the coloring $c$.*

**Definition B.2.** *Let $\mathcal{CC}_1, \mathcal{CC}_2$ be two combinatorial complexes and $c$ be a coloring function. We denote $\mathcal{CC}_1, \mathcal{CC}_2$ are c-similar, denoted by $c^{\mathcal{CC}_1}, c^{\mathcal{CC}_2}$, if the number of cells in $\mathcal{CC}_1$ colored with a given color equals the number of cells in $\mathcal{CC}_2$ with the same color. Otherwise, we have $c^{\mathcal{CC}_1} \neq c^{\mathcal{CC}_2}$.*

**Definition B.3.** *A combinatorial coloring $c$ refines a combinatorial coloring $d$, denoted by $c \sqsubseteq d$, if for all cell complexes $X$ and $Y$ and all $\sigma \in \mathcal{CC}_1$ and $\tau \in \mathcal{CC}_2$, $c_\sigma^{\mathcal{CC}_1} = c_\tau^{\mathcal{CC}_2}$ implies $d_\sigma^{\mathcal{CC}_1} = d_\tau^{\mathcal{CC}_2}$. Additionally, if $d \sqsubseteq c$, we say the two colorings are equivalent and we represent it by $c \equiv d$.*

**Lemma B.4.** *Let $c^t$ and $d^t$ be the CCWL colorings using full and reduced schemes, respectively. Then for all $t$, we have:*
$$c^t \sqsubseteq d^{t+1}, \quad d^{t+1} \sqsubseteq c^t \quad \Rightarrow \quad c^t \equiv d^{t+1}.$$

*Proof.* Assume that for a fixed $t \geq 0$, the following two refinement relations hold: (1) $c^t \sqsubseteq d^{t+1}$, (2) $d^{t+1} \sqsubseteq c^t$. By the definition of refinement, condition (1) means that for any two combinatorial complexes $\mathcal{CC}_1, \mathcal{CC}_2$ and any cells $\sigma \in \mathcal{CC}$, $\tau \in \mathcal{CC}_2$, $c_\sigma^{\mathcal{CC}_1,t} = c_\tau^{\mathcal{CC}_2,t} \implies d_\sigma^{\mathcal{CC}_1,t+1} = d_\tau^{\mathcal{CC}_2,t+1}$. Similarly, condition (2) implies: $d_\sigma^{\mathcal{CC}_1,t+1} = d_\tau^{\mathcal{CC}_2,t+1} \implies c_\sigma^{\mathcal{CC}_1,t} = c_\tau^{\mathcal{CC}_2,t}$. Combining both implications, we can obtain: $c_\sigma^{\mathcal{CC}_1,t} = c_\tau^{\mathcal{CC}_2,t} \iff d_\sigma^{\mathcal{CC}_1,t+1} = d_\tau^{\mathcal{CC}_2,t+1}$. This means that the colorings $c^t$ and $d^{t+1}$ induce the same partition on the set of all cells across all combinatorial complexes. By Definition 3, we conclude that $c^t \equiv d^{t+1}$. Since this holds under the given assumptions for arbitrary $t$, the lemma is proved.

$\square$

*Proof of Lemma 4.4.* Suppose $\mathcal{CC}_1$ and $\mathcal{CC}_2$ are isomorphic, there is a bijection $\varphi : \mathcal{C}_1 \to \mathcal{C}_2$ that respects both cell ranks and all the neighborhood relations CCWL uses: $\mathcal{B}, \mathcal{C}, \mathcal{N}_\downarrow$, and $\mathcal{N}_\uparrow$. We prove corresponding cells always get the same color. From the Definition 4.2 at $t = 0$, initialization of cells are the same, and $\varphi$ preserves it, so $\sigma$ and $\varphi(\sigma)$ start identically colored. We suppose that it holds at step $t$. To compute $c^{(t+1)}(\sigma)$, CCWL hashes a tuple of $c^{(t)}(\sigma)$ and the color multisets from its four neighborhoods. But $\varphi$ maps those neighborhoods exactly to those of $\varphi(\sigma)$, and by the induction hypothesis, all neighbors already share colors. In the case of the HASH is injective function and two input tuples are identified, then final colorings are the same as $c^{(t+1)}(\sigma) = c^{(t+1)}(\varphi(\sigma))$. The argument repeats cleanly for every $t$. And since $\varphi$ is a bijection function, the color multisets over the two complexes remain identical at every iteration. $\square$

*Proof for Lemma 4.5.* Assume that $c \sqsubseteq d$, i.e., for any combinatorial complexes $\mathcal{CC}_1(\mathcal{S}_1, \mathcal{C}_1, rk_1), \mathcal{CC}_2(\mathcal{S}_2, \mathcal{C}_2, rk_2)$ and cells $\sigma \in \mathcal{CC}_1, \tau \in \mathcal{CC}_2$, $c_\sigma^{\mathcal{CC}} = c_\tau^{\mathcal{CC}_2} \implies d_\sigma^{\mathcal{CC}} = d_\tau^{\mathcal{CC}_2}$. Equivalently, the contrapositive holds: $d_\sigma^{\mathcal{CC}} \neq d_\tau^{\mathcal{CC}_2} \implies c_\sigma^{\mathcal{CC}} \neq c_\tau^{\mathcal{CC}_2}$. Suppose for contradiction that the multisets of $c$-colors are $\{\!\{c_\sigma^{\mathcal{CC}_1} \mid \sigma \in P(\mathcal{S}_1)\}\!\} = \{\!\{c_\tau^{\mathcal{CC}_2} \mid \tau \in P(\mathcal{S}_2)\}\!\}$. This means there exists a bijection $\phi : P(\mathcal{S}_1) \to P(\mathcal{S}_2)$ such that for all $\sigma \in P(\mathcal{S}_\infty)$, $c_\sigma^{\mathcal{CC}_1} = c_{\phi(\sigma)}^{\mathcal{CC}_2}$. Since $c \sqsubseteq d$, we have: $c_\sigma^{\mathcal{CC}_1} = c_{\phi(\sigma)}^{\mathcal{CC}_2} \implies d_\sigma^{\mathcal{CC}_1} = d_{\phi(\sigma)}^{\mathcal{CC}_2}$. Therefore, for all $\sigma \in A$, $d_\sigma^{\mathcal{CC}_1} = d_{\phi(\sigma)}^{\mathcal{CC}_2}$, which implies that the multisets of $d$-colors are also equal: $\{\!\{d_\sigma^{\mathcal{CC}_1} \mid \sigma \in A\}\!\} = \{\!\{d_\tau^{\mathcal{CC}_2} \mid \tau \in B\}\!\}$. But this contradicts the assumption that $\{\!\{d_\sigma^{\mathcal{CC}_1} \mid \sigma \in A\}\!\} \neq \{\!\{d_\tau^{\mathcal{CC}_2} \mid \tau \in B\}\!\}$. Hence, our supposition must be false, and we conclude: $\{\!\{c_\sigma^{\mathcal{CC}_1} \mid \sigma \in A\}\!\} \neq \{\!\{c_\tau^{\mathcal{CC}_2} \mid \tau \in B\}\!\}$. $\square$

*Proof for Corollary 4.6.* Assume that $c \sqsubseteq d$, i.e., for any combinatorial complexes $\mathcal{CC}_1(\mathcal{S}_1, \mathcal{C}_1, \mathrm{rk}_1), \mathcal{CC}_2(\mathcal{S}_2, \mathcal{C}_2, \mathrm{rk}_2)$ and cells $\sigma \in \mathcal{CC}_1, \tau \in \mathcal{CC}_2$, $c_\sigma^{\mathcal{CC}_1} = c_\tau^{\mathcal{CC}_2} \implies d_\sigma^{\mathcal{CC}_1} = d_\tau^{\mathcal{CC}_2}$. Equivalently, the contrapositive holds: $d_\sigma^{\mathcal{CC}_1} \neq d_\tau^{\mathcal{CC}_2} \implies c_\sigma^{\mathcal{CC}_1} \neq c_\tau^{\mathcal{CC}_2}$. Now, suppose for contradiction that $c^{\mathcal{CC}_1} = c^{\mathcal{CC}_2}$, meaning the multisets of $c$-colors over the entire cell sets are equal: $\{\!\{c_\sigma^{\mathcal{CC}_1} \mid \sigma \in \mathcal{C}_1\}\!\} = \{\!\{c_\tau^{\mathcal{CC}_2} \mid \tau \in \mathcal{C}_2\}\!\}$. This implies there exists a bijection $\phi : \mathcal{C}_1 \to \mathcal{C}_2$ such that for all $\sigma \in \mathcal{C}_1$, $c_\sigma^{\mathcal{CC}_1} = c_{\phi(\sigma)}^{\mathcal{CC}_2}$. Since $c \sqsubseteq d$, we have: $c_\sigma^{\mathcal{CC}_1} = c_{\phi(\sigma)}^{\mathcal{CC}_2} \implies d_\sigma^{\mathcal{CC}_1} = d_{\phi(\sigma)}^{\mathcal{CC}_2}$. Therefore, for all $\sigma \in \mathcal{C}_1$, $d_\sigma^{\mathcal{CC}_1} = d_{\phi(\sigma)}^{\mathcal{CC}_2}$, which implies that the multisets of $d$-colors are also equal: $\{\!\{d_\sigma^{\mathcal{CC}_1} \mid \sigma \in \mathcal{C}_1\}\!\} = \{\!\{d_\tau^{\mathcal{CC}_2} \mid \tau \in \mathcal{C}_2\}\!\}$, i.e., $d^{\mathcal{CC}_1} = d^{\mathcal{CC}_2}$. But this contradicts the assumption that $d^{\mathcal{CC}_1} \neq d^{\mathcal{CC}_2}$. Hence, the supposition must be false, then conclude: $c^{\mathcal{CC}_1} \neq c^{\mathcal{CC}_2}$.

□

**Remark** The regularity conditions stated in Lemma 4.7 are sufficient for common combinatorial structures, but not necessary. For graphs, hypergraphs, simplicial complexes, and cell complexes, only information transfer with rank differences of 0 or 1 is handled, and the pattern of associations between ranks is more strictly restricted: upper and lower adjacencies already encode all boundary and co-boundary relationships. Theorem 4.8 provides the generalized update rule to demonstrate the expressive power of the two schemes is the same, i.e., they distinguish the same set of non-isomorphic $\mathcal{CC}$.

***Proof for Lemma** 4.7.* Consider two combinatorial complexes $\mathcal{CC}_1 = (\mathcal{S}_1, \mathcal{C}_1, \mathrm{rk}_1)$ and $\mathcal{CC}_2 = (\mathcal{S}_2, \mathcal{C}_2, \mathrm{rk}_2)$. Let $a^t$ denote the colorings of the generalized rules $\mathrm{HASH}(c_\sigma^t, c_\mathcal{B}^t(\sigma), c_\mathcal{C}^t(\sigma), c_{\mathcal{N}_\downarrow}^t(\sigma), c_{\mathcal{N}_{,\uparrow}}^t(\sigma))$ and $b^t$ denote the colorings restricted rules $\mathrm{HASH}(c_\sigma^t, c_\mathcal{B}^t(\sigma), c_{\mathcal{N}_\downarrow}^t(\sigma), c_{\mathcal{N}_\uparrow}^t(\sigma))$ at iteration $t$. Since the generalized rule incorporates the additional co-boundary colors $c_\mathcal{C}^t(\sigma)$, it is straightforward that $a^t \sqsubseteq b^t$ (Definition B.3). By induction, if $b^t \sqsubseteq a^t$ holds, then $a^t \equiv b^t$.

The base step holds trivially in the case of all cells are initialized with identical colors. Assume $\sigma \in \mathcal{CC}_1$ and $\tau \in \mathcal{CC}_2$ are cells of the same rank such that $b_\sigma^{t+1} = b_\tau^{t+1}$. Then the arguments of the hash function must coincide. Suppose $b_\sigma^t = b_\tau^t$, $b_\downarrow^t(\sigma) = b_\downarrow^t(\tau)$, $b_\uparrow^t(\sigma) = b_\uparrow^t(\tau)$, $b_\mathcal{B}^t(\sigma) = b_\mathcal{B}^t(\tau)$ hold. To demonstrate that these equalities imply $b_\mathcal{C}^t(\sigma) = b_\mathcal{C}^t(\tau)$ of the generalized rules. By the definition of the upper adjacencies $\mathcal{N}_\uparrow(\sigma)$ $(c_{\mathcal{N}_\uparrow}(\sigma) = \{\{(c_\tau, c_\delta)|\tau \in \mathcal{N}_\uparrow(\sigma) \text{ and } \delta \in C(\sigma, \tau)\}\}.)$, it holds $b_\uparrow^t(\sigma) = b_\uparrow^t(\tau)$ that ensures that

$$\{\{b_{\delta_\sigma}^t \mid (\cdot, b_{\delta_\sigma}^t) \in b_\uparrow^t(\sigma)\}\} = \{\{b_{\delta_\tau}^t \mid (\cdot, b_{\delta_\tau}^t) \in b_\uparrow^t(\tau)\}\}. \tag{14}$$

Each coface $\delta_\sigma \in \mathcal{N}_\uparrow(\sigma)$ (and analogously for $\tau$) contributes exactly $\mathrm{rk}(\sigma) + 1$ tuples to its upward multisets. After removing repeated entries corresponding to these cofaces, we obtain

$$\{\{b_{\delta_\sigma}^t \mid \delta_\sigma \in \mathcal{N}_\uparrow(\sigma)\}\} = \{\{b_{\delta_\tau}^t \mid \delta_\tau \in \mathcal{N}_\uparrow(\tau)\}\}, \tag{15}$$

which directly implies $b_\mathcal{C}^t(\sigma) = b_\mathcal{C}^t(\tau)$. By the induction hypothesis, we further have $a_\sigma^t = a_\tau^t$, $a_\downarrow^t(\sigma) = a_\downarrow^t(\tau)$, $a_\uparrow^t(\sigma) = a_\uparrow^t(\tau)$, $a_\mathcal{B}^t(\sigma) = a_\mathcal{B}^t(\tau)$, and $a_\mathcal{C}^t(\sigma) = a_\mathcal{C}^t(\tau)$. Therefore, $a_\sigma^{t+1} = a_\tau^{t+1}$, concluding the proof.

□

***Proof of Theorem** 4.8.* Let $a^t$ denote the coloring obtained from the full generalized rule as the mapping function $\mathrm{HASH}\big(c^{(t)}(\sigma), c_\mathcal{B}^{(t)}(\sigma), c_\mathcal{C}^{(t)}(\sigma), c_{\mathcal{N}_\downarrow}^{(t)}(\sigma), c_{\mathcal{N}_\uparrow}^{(t)}(\sigma)\big)$, and $b^t$ denote the coloring from the restricted rule as $\mathrm{HASH}(c_\sigma^{(t)}, c_{\mathcal{N}_\downarrow}^{(t)}(\sigma), c_{\mathcal{N}_\uparrow}^{(t)}(\sigma))$. Since $a^t$ includes more structural information, we have $a^t \sqsubseteq b^t$. By induction, we further prove that $b^{(t+1)} \sqsubseteq a^t$, which implies $a^t \equiv b^t$. we initial colors are determined by $c^{(0)}(\sigma) = \mathrm{HASH}_0(\mathrm{rk}(\sigma), |\mathcal{B}(\sigma)|, |\mathcal{C}(\sigma)|)$. Since $|\mathcal{B}(\sigma)|$ and $|\mathcal{C}(\sigma)|$ are determined by the neighborhood degrees, $a^0 \equiv b^0$. Suppose $b_\sigma^{t+2} = b_\tau^{t+2}$ for any two cells $\sigma$ and $\tau$ from combinatorial complexes $\mathcal{CC}_1, \mathcal{CC}_2$, respectively. By definition of the update rule, we obtain that $b_\sigma^{(t+1)} = b_\tau^{(t+1)}$, $b_{\mathcal{N}_\uparrow}^{(t+1)}(\sigma) = b_{\mathcal{N}_\uparrow}^{(t+1)}(\tau)$, $b_{\mathcal{N}_\downarrow}^{(t+1)}(\sigma) = b_{\mathcal{N}_\downarrow}^{(t+1)}(\tau)$. The next step aims to clarify that the boundary $\mathcal{B}$ and co-boundary $\mathcal{C}$ also hold as $b_\mathcal{B}^{(t+1)}(\sigma) = b_\mathcal{B}^{(t+1)}(\tau)$ and $b_\mathcal{C}^{(t+1)}(\sigma) = b_\mathcal{C}^{(t+1)}(\tau)$. Given $b_{\mathcal{N}_\uparrow}^{(t+1)}(\sigma) = b_{\mathcal{N}_\uparrow}^{(t+1)}(\tau)$, by definition

$$\left\{\left\{b_{\delta_\sigma}^{(t+1)}|(\cdot, b_{\delta_\sigma}^{(t+1)}) \in b_{\mathcal{N}_\uparrow}^{(t+1)}(\sigma)\right\}\right\} = \left\{\left\{b_{\delta_\tau}^{(t+1)}|(\cdot, b_{\delta_\tau}^{(t+1)}) \in b_{\mathcal{N}_\uparrow}^{(t+1)}(\tau)\right\}\right\}, \tag{16}$$

Note that the upper adjacent cells $\mathcal{N}_\uparrow(\sigma) = \{\tau|\exists\delta, s.t.\sigma \prec \delta \text{ and } \tau \prec \delta\}$. we know that the co-boundary adjacent cells $\mathcal{C}(\sigma) = \{\delta|\sigma \prec \delta\}$. From the definition that upper adjacent colors $\sigma$ is $c_{\mathcal{N}_\uparrow}(\sigma) = \{\{(c_\tau, c_\delta)|\tau \in \mathcal{N}_\uparrow(\sigma) \text{ and } \delta \in C(\sigma, \tau)\}\}$, co-boundary adjacent $c_\mathcal{C}(\sigma) = \{\{c_\tau|\tau \in \mathcal{C}(\sigma)\}\}$,

$$\left\{\left\{(b_{\delta_\sigma}^{(t+1)}, b_{\gamma_\sigma}^{(t+1)})|\delta \in \mathcal{N}_\uparrow(\sigma), \gamma \in \mathcal{C}(\sigma, \delta)\right\}\right\} = \left\{\left\{(b_{\delta_\tau}^{(t+1)}, b_{\gamma_\tau}^{(t+1)})|\delta \in \mathcal{N}_\uparrow(\sigma), \gamma \in \mathcal{C}(\sigma, \delta)\right\}\right\}, \tag{17}$$

This multiset includes all upper adjacent cells $\delta$ that contain $\sigma$ in their boundary, along with colors of their co-boundary neighbors $\gamma$. According to the iteration of WL test, cells with different boundary sizes have different colors. Therefore, we can partition these two multi-sets by the size of the cell boundaries, while preserving the equality between these sub-multisets. Since $\sigma$ is a boundary cell of $\delta$, the tuple $(\delta, \sigma)$ appears with multiplicity $|\mathcal{B}(\delta)| - 1$. For each $n \in \mathbb{N}$, we rewrite

$$\left\{\left\{b_{\delta_\sigma}^{(t+1)}|(\cdot, b_{\delta_\sigma}^{(t+1)}) \in b_{\mathcal{N}_\uparrow}^{(t+1)}(\sigma) \text{ and } |\mathcal{B}(\delta_\sigma)| = n\right\}\right\} = \left\{\left\{b_{\delta_\tau}^{(t+1)}|(\cdot, b_{\delta_\tau}^{(t+1)}) \in b_{\mathcal{N}_\uparrow}^{(t+1)}(\tau) \text{ and } |\mathcal{B}(\delta_\tau)| = n\right\}\right\}, \tag{18}$$

Then for cell $\gamma \in \mathcal{C}(\gamma)$, $\gamma$ exchanges messages with all the other boundary cells of $\gamma_\sigma, \gamma_\tau$. Thus, the value of $|\mathcal{B}(\delta)|$ is encoded in $b_\delta^{(t+1)}$, can partition the multiset by boundary size. Each co-boundary cell $\delta$ with $|\mathcal{B}(\delta)| = n$ sends $n - 1$ repeated messages. Removing multiplicities accordingly reconstructs the true multiset:

$$\left\{\!\!\left\{ b_\delta^{(t+1)} \mid \delta \in \mathcal{C}(\sigma) \right\}\!\!\right\} = \left\{\!\!\left\{ b_\delta^{(t+1)} \mid \delta \in \mathcal{C}(\tau) \right\}\!\!\right\}. \tag{19}$$

we infer that the multisets of colors of co-boundary cells $\mathcal{C}(\sigma)$ and $\mathcal{C}(\tau)$ are identical. That is, $b_\mathcal{C}^{(t+1)}(\sigma) = b_\mathcal{C}^{(t+1)}(\tau)$. If $b_\sigma^{(t+2)} = b_\tau^{(t+2)}$, we conclude $a_\sigma^{(t+1)} = a_\tau^{(t+1)}$, hence $b^{(t+1)} \sqsubseteq a^t$ by the induction hypothesis that $a^t$ already refines all colorings. Thus $b^{(t+1)} \sqsubseteq a^{(t+1)}$, completing the inductive step. Therefore, $a^t \equiv b^t$ for all $t$, and both schemes are equally expressive. Given $b_{\mathcal{N}_\downarrow}^{(t+1)}(\sigma) = b_{\mathcal{N}_\downarrow}^{(t+1)}(\tau)$, by definition

$$\left\{\!\!\left\{ \left\{ b_{\delta_\sigma}^{(t+1)} | (\cdot, b_{\delta_\sigma}^{(t+1)}) \in b_{\mathcal{N}_\downarrow}^{(t+1)}(\sigma) \right\} \right\}\!\!\right\} = \left\{\!\!\left\{ \left\{ b_{\delta_\tau}^{(t+1)} | (\cdot, b_{\delta_\tau}^{(t+1)}) \in b_{\mathcal{N}_\downarrow}^{(t+1)}(\tau) \right\} \right\}\!\!\right\}, \tag{20}$$

The lower adjacent cells $\mathcal{N}_\downarrow(\sigma) = \{\tau | \exists \delta, s.t. \delta \prec \sigma \text{ and } \delta \prec \tau\}$. we know that the boundary adjacent cell $\mathcal{B}(\sigma) = \{\theta \prec \sigma\}$. From the definition, the lower adjacent colors of $\sigma$ are: $c_{\mathcal{N}_\downarrow}(\sigma) = \left\{\!\!\left\{ (c_\theta^{(t)}, c_\omega^{(t)}) \mid \theta \in \mathcal{B}(\sigma), \omega \in \mathcal{C}(\theta) \setminus \{\sigma\} \right\}\!\!\right\}$, and the boundary colors of $\sigma$ are: $c_\mathcal{B}^{(t)}(\sigma) = \left\{\!\!\left\{ c_\theta^{(t)} \mid \theta \in \mathcal{B}(\sigma) \right\}\!\!\right\}$. This neighborhood representation includes all boundary cells $\theta$ of $\sigma$ and their other cofaces $\omega$. Since each such pair $(\theta, \omega)$ occurs $|\mathcal{C}(\theta)| - 1$ times, we can group the elements by the value of $|\mathcal{C}(\theta)|$ to remove neighborhood redundancy, as follows

$$\left\{\!\!\left\{ (b_{\theta_\sigma}^{(t+1)}, b_{\omega_\sigma}^{(t+1)}) \mid \theta \in \mathcal{B}(\sigma), \omega \in \mathcal{C}(\theta) \setminus \{\sigma\} \right\}\!\!\right\} = \left\{\!\!\left\{ (b_{\theta_\tau}^{(t+1)}, b_{\omega_\tau}^{(t+1)}) \mid \theta \in \mathcal{B}(\tau), \omega \in \mathcal{C}(\theta) \setminus \{\tau\} \right\}\!\!\right\}, \tag{21}$$

This multiset includes all boundary cells $\theta$ that belong to $\sigma$, together with their co-boundary neighbors $\omega$. The cells with different co-boundary sizes $|\mathcal{C}(\theta)|$ must have different colors. Therefore, we can partition these multisets by the boundary size of $\theta$, preserving the equality of the sub-multisets. Since $\theta$ is a face of $\sigma$, and each $\theta$ appears $|\mathcal{C}(\theta)| - 1$ times in the neighborhood, we group by boundary degree $n$ and obtain:

$$\left\{\!\!\left\{ b_{\theta_\sigma}^{(t+1)} \mid (\cdot, b_{\theta_\sigma}^{(t+1)}) \in b_{\mathcal{N}_\downarrow}^{(t+1)}(\sigma), |\mathcal{C}(\theta_\sigma)| = n \right\}\!\!\right\} = \left\{\!\!\left\{ b_{\theta_\tau}^{(t+1)} \mid (\cdot, b_{\theta_\tau}^{(t+1)}) \in b_{\mathcal{N}_\downarrow}^{(t+1)}(\tau), |\mathcal{C}(\theta_\tau)| = n \right\}\!\!\right\}. \tag{22}$$

Thus, we can recover the true multiplicity-free multiset of boundary cell colors by grouping and deduplicating. That is:

$$\left\{\!\!\left\{ b_\theta^{(t+1)} \mid \theta \in \mathcal{B}(\sigma) \right\}\!\!\right\} = \left\{\!\!\left\{ b_\theta^{(t+1)} \mid \theta \in \mathcal{B}(\tau) \right\}\!\!\right\}, \tag{23}$$

which means: $b_\mathcal{B}^{(t+1)}(\sigma) = b_\mathcal{B}^{(t+1)}(\tau)$. Therefore, we can reconstruct $c_\mathcal{B}^{(t)}(\sigma)$ from $c_{\mathcal{N}_\downarrow}^{(t)}(\sigma)$, and the same holds for $\tau$. If $b_\sigma^{(t+2)} = b_\tau^{(t+2)}$, then the input sets to the full rule at iteration $t + 1$ are equal. Hence: $a_\sigma^{(t+1)} = a_\tau^{(t+1)}$, and we conclude that:$b^{(t+1)} \sqsubseteq a^{(t+1)}$, completing the inductive step. Therefore, by induction on $t$, we have $a^t \equiv b^t$ for all $t$, and both schemes are equally expressive. $\qquad\square$

**Proof for Lemma** 4.9. Let $\mathcal{CC}_1$ and $\mathcal{CC}_2$ be two combinatorial complexes. Let $c$ denote the stable coloring produced by a WL-style test (e.g., a restricted version operating only on 0-cells or using fewer aggregation rules), and let $d$ denote the stable coloring produced by CCWL. Since CCWL aggregates over more structural information. Specifically, it considers both lower neighbourhoods $\mathcal{N}_\downarrow$, upper neighbourhoods $\mathcal{N}_\uparrow$, boundary and coface relations, its coloring $d$ distinguishes more cells than $c$. Therefore, the partition induced by $c$ is coarser than that induced by $d$, i.e., $c \sqsubseteq d$. Suppose that the WL test distinguishes $\mathcal{CC}_1$ and $\mathcal{CC}_2$. That is, $c^{\mathcal{CC}_1} \neq c^{\mathcal{CC}_2}$, which means the multiset of colors differs between the two complexes. By Corollary 4.6, since $c \sqsubseteq d$, we have: If $c^{\mathcal{CC}_1} \neq c^{\mathcal{CC}_2}$, then $d^{\mathcal{CC}_1} \neq d^{\mathcal{CC}_2}$. Therefore, CCWL also distinguishes $\mathcal{CC}_1$ and $\mathcal{CC}_2$. This holds for any pair distinguishable by the WL test. Hence, CCWL is at least as powerful as the WL test in distinguishing non-isomorphic combinatorial complexes.

$\square$

**Remark** Co-boundary relations are encoded in upper neighborhood. Let $CC$ be a combinatorial complex with a rank-aware coloring function $c^{(t)} : \mathcal{C} \to \mathcal{A}$ at iteration $t$. For any cell $\sigma \in \mathcal{C}$, denote by $\mathcal{C}(\sigma) = \{\delta \in \mathcal{C} : \delta \succ \sigma\}$ and $\mathcal{N}_\uparrow(\sigma) = \{\tau \in \mathcal{C} : \exists \delta \in \mathcal{C}(\sigma) \text{ with } \tau \prec \delta\}$. The coboundary color multiset of $\sigma$ is defined as $c_{\mathcal{C}}^{(t)}(\sigma) = \{\!\!\{ c^{(t)}(\delta) : \delta \in \mathcal{C}(\sigma) \}\!\!\}$, and the upper neighborhood color multiset as $c_{\mathcal{N}_\uparrow}^{(t)}(\sigma) = \{\!\!\{ (c^{(t)}(\tau), c^{(t)}(\delta)) : \tau \in \mathcal{N}_\uparrow(\sigma), \delta \in \mathcal{C}(\sigma), \tau \prec \delta \}\!\!\}$. Then the upper neighborhood multiset refines the coboundary multiset in the sense of multiset projection:$c_{\mathcal{C}}^{(t)}(\sigma) \sqsubseteq c_{\mathcal{N}_\uparrow}^{(t)}(\sigma)$, there exists a surjective projection $\pi : c_{\mathcal{N}_\uparrow}^{(t)}(\sigma) \to c_{\mathcal{C}}^{(t)}(\sigma)$ defined by $\pi(c_\tau, c_\delta) = c_\delta$ such that $\pi\left(c_{\mathcal{N}_\uparrow}^{(t)}(\sigma)\right) = c_{\mathcal{C}}^{(t)}(\sigma)$. For every $\delta \in \mathcal{C}(\sigma)$, there exists at least one $\tau \in \delta \setminus \{\sigma\}$ satisfying $\tau \prec \delta$ and $\tau \in \mathcal{N}_\uparrow(\sigma)$ by the local incidence property of the complex. Hence $(c^{(t)}(\tau), c^{(t)}(\delta)) \in c_{\mathcal{N}_\uparrow}^{(t)}(\sigma)$ and $\pi((c_\tau, c_\delta)) = c_\delta$. Therefore $\pi$ is surjective, implying that $c_{\mathcal{C}}^{(t)}(\sigma)$ can be recovered from $c_{\mathcal{N}_\uparrow}^{(t)}(\sigma)$ by projection, which establishes the refinement relation.

**Remark** Boundary relations are encoded in the lower neighborhood. Let $CC$ be a combinatorial complex with a rank-aware coloring $c^{(t)} : \mathcal{C} \to \mathcal{A}$ at iteration $t$. For $\sigma \in \mathcal{C}$ set $\mathcal{B}(\sigma) = \{\delta \in \mathcal{C} : \delta \prec \sigma\}$. Define the boundary color multiset $c_{\mathcal{B}}^{(t)}(\sigma) = \{\!\!\{ c^{(t)}(\delta) : \delta \in \mathcal{B}(\sigma) \}\!\!\}$. Define the extended lower neighborhood color multiset by $c_{\widetilde{\mathcal{N}}_{\cdot,\downarrow}}^{(t)}(\sigma) = \{\!\!\{ (c^{(t)}(\tau), c^{(t)}(\delta)) : \delta \in \mathcal{B}(\sigma), \tau \in \mathcal{C}, \delta \prec \tau \}\!\!\}$, i.e., pairs of a co-incident same-rank cell $\tau$ and a boundary face $\delta$, allowing $\tau = \sigma$. Then $c_{\widetilde{\mathcal{N}}_{\cdot,\downarrow}}^{(t)}(\sigma)$ refines $c_{\mathcal{B}}^{(t)}(\sigma)$ in the multiset-projection sense: $c_{\mathcal{B}}^{(t)}(\sigma) \sqsubseteq c_{\widetilde{\mathcal{N}}_{\cdot,\downarrow}}^{(t)}(\sigma)$, namely, the projection $\pi(c_\tau, c_\delta) = c_\delta$ is surjective and satisfies $\pi\left(c_{\widetilde{\mathcal{N}}_{\cdot,\downarrow}}^{(t)}(\sigma)\right) = c_{\mathcal{B}}^{(t)}(\sigma)$. For every $\delta \in \mathcal{B}(\sigma)$ we have $\delta \prec \sigma$, hence $(c^{(t)}(\sigma), c^{(t)}(\delta)) \in c_{\widetilde{\mathcal{N}}_{\cdot,\downarrow}}^{(t)}(\sigma)$. Thus the projection $\pi(c_\tau, c_\delta) = c_\delta$ hits $c^{(t)}(\delta)$ for each boundary face $\delta$.

## B.2. Lifting Mapping

A combinatorial complex lifting map is a function $f : G \to CC$ from the space of graphs $G$ to the space of regular combinatorial complexes $CC$ with the property that two graphs $G_1, G_2$ are isomorphic iff the combinatorial complexes $f(G_1), f(G_2)$ are isomorphic. To prove the Theorem 4.11, we provide the propositions as follows:

**Proposition B.5.** *Let $f : \mathcal{G} \to CC$ be a lifting function from graph to combinatorial complexes. Then there exists a lifting $f$ such that CCWL over $f(\mathcal{G})$ is strictly more powerful than the classical Weisfeiler-Lehman (1-WL) test on graphs.*

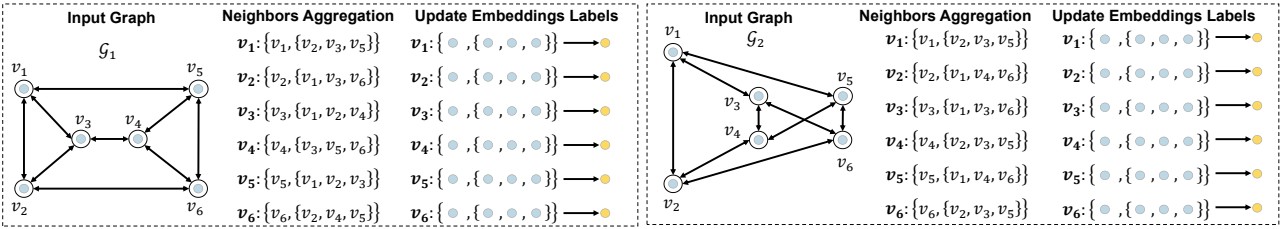

*Figure B.1.* Two non-isomorphic graphs, 1-WL and hypergraph-WL cannot distinguish them.

*Proof.* Let $G = (V, E)$ be a finite simple graph. We define a lifting $f : \mathcal{G} \to CC$ by mapping $G$ to a combinatorial complex $X = f(G)$ as follows. Each node $v \in V$ is represented by a 0-cell $v^0$. Each edge $e = (u, v) \in E$ corresponds to a 1-cell $e^1$ with boundary $\partial e^1 = \{u^0, v^0\}$. For every edge $e \in E$, we attach a 2-cell $P_e^2$ (a parity gadget) whose boundary forms a cycle of 1-cells containing $e^1$ and a set of auxiliary 1-cells introduced solely for this construction. The internal structure of $P_e^2$ is chosen from two non-isomorphic configurations, selected according to a global parity bit in the style of the Cai-Fürer-Immerman (CFI) construction. While the parity is fixed uniformly across all edges within a single graph, distinct graphs may adopt different parity assignments. This lifting is faithful: for any pair of graphs $G_1$ and $G_2$, we have $G_1 \cong G_2$ if and only if $f(G_1) \cong f(G_2)$.

To compare the expressive power of CCWL on $\{f(G)\}$ and the classical 1-WL test on $\mathcal{G}$, observe that CCWL subsumes 1-WL. Specifically, the color update for a 0-cell $v^0$ in $X = f(G)$ aggregates over the multiset of colors of incident 1-cells $\{e^1 \mid v^0 \in \partial e^1\}$, each of which corresponds bijectively to an edge $(v, u) \in E$. Since the adjacent 0-cells of these 1-cells are the neighbors of $v$ in $G$, the resulting aggregation encodes the same neighborhood multiset that governs the 1-WL

refinement at $v$. Consequently, the sequence of color partitions produced by 1-WL on $G$ is recoverable from the restriction of CCWL's coloring to the 0-skeleton of $f(G)$. Thus, CCWL is at least as expressive as 1-WL. To establish strict separation, consider a pair of non-isomorphic Cai-Fürer-Immerman (CFI) graphs $(G_1, G_2)$ that are indistinguishable by 1-WL; their color refinement sequences and the resulting color multisets that coincide at every iteration. We instantiate the lifting $f$ by assigning one parity configuration to all gadgets $\{P_e^2\}$ in $f(G_1)$, and the opposite (non-isomorphic) configuration to those in $f(G_2)$, as prescribed by the CFI construction. Although $G_1$ and $G_2$ are 1-WL-equivalent, the resulting complexes $f(G_1)$ and $f(G_2)$ differ in the combinatorial structure of their 2-cells and associated coboundaries. For certain 1-cells $e^1$, the multiset of cofaces i.e., the 2-cells incident to $e^1$, which differs between the two lifted complexes due to the global parity constraint. Because CCWL incorporates coboundary information during refinement, the affected 1-cells acquire distinct colors, and this distinction may further influence neighboring 0-cells. Then the final color multisets over $f(G_1)$ and $f(G_2)$ diverge. Hence, while 1-WL cannot distinguish $G_1$ from $G_2$, CCWL distinguishes $f(G_1)$ from $f(G_2)$. This demonstrates the existence of a lifting $f$ for which CCWL on $f(\mathcal{G})$ is strictly more expressive than 1-WL on $\mathcal{G}$.

$\square$

**Proposition B.6.** *There exists a lifting $f : \mathcal{H} \to \mathcal{CC}$ from hypergraphs to combinatorial complexes such that CCWL applied to $f(\mathcal{H})$ is strictly more expressive than the hypergraph Weisfeiler-Lehman (HWL) test.*

**Case 5** (1-WL $\prec$ CCWL) Figure B.1 shows the 1-WL and Hypergraph WL tests fail to distinguish the two non-isomorphic regular graphs: their iterative color refinement procedures, which operating solely on nodes and (hyper)edges yields identical stable colorings. In contrast, when these graphs are lifted to combinatorial complexes $\mathcal{CC}_1$ and $\mathcal{CC}_2$ via the lifting mapping function, the CCWL test successfully distinguishes them. Specifically, Figure B.2 displays the variant of CCWL that aggregates over lower and upper neighbors, i.e., $\mathrm{HASH}\big(c^{(t)}(\sigma),\, c_{\mathcal{N}_\downarrow}^{(t)}(\sigma),\, c_{\mathcal{N}_\uparrow}^{(t)}(\sigma)\big)$, which is already sufficient to separate $\mathcal{CC}_1$ and $\mathcal{CC}_2$ . This demonstrates that even a restricted form of CCWL, which omits explicit boundary ( $\mathcal{B}$ ) and coboundary ( $\mathcal{C}$ ) aggregation, attains the full distinguishing power of the general update rule $\mathrm{HASH}\big(c^{(t)}(\sigma),\, c_{\mathcal{B}}^{(t)}(\sigma),\, c_{\mathcal{C}}^{(t)}(\sigma),\, c_{\mathcal{N}_\downarrow}^{(t)}(\sigma),\, c_{\mathcal{N}_\uparrow}^{(t)}(\sigma)\big)$, on this class of instances. The success of the lifting-based approach thus highlights its ability to resolve structural ambiguities that are inherently invisible to classical WL tests operating on flat graph or hypergraph representations.

**Proposition B.7.** *There exists a lifting $f : \mathcal{S} \to \mathcal{CC}$ from simplicial complexes to combinatorial complexes such that CCWL applied to $f(\mathcal{S})$ is strictly more expressive than the Simplicial Weisfeiler-Lehman (SWL) test.*

**Case 5** (SWL $\prec$ CCWL) Figure B.3 example (a) considers two 1-dimensional simplicial complexes $S_1$ and $S_2$. The 1-skeleton of $S_1$ is the cycle graph $C_8$, while $S_2$ consists of two disjoint 4-cycles, i.e., $C_4 \sqcup C_4$. Both complexes have eight nodes and eight edges, and without any 2-simplices. For every 1-simplex, the co-boundary (upper) neighborhood is empty. SWL updates colors by aggregating information from boundary and co-boundary neighborhoods. In the case of no 2-simplices are present here, SWL refinement coincides with the standard 1-WL refinement. we find that both $S_1$ and $S_2$ are 2-regular, each node is incident to exactly two edges, and each edge is bounded by two distinct nodes. At the initialization, all nodes in both complexes are the same multisets of neighboring colors, two identical colors coming from their incident edges. Similarly, every edge observes the same colors. The process stabilizes with one node color and one edge color for both $S_1$ and $S_2$. Hence, the stable colorings produced by SWL are the same on the $S_1$ and $S_2$, and SWL cannot distinguish them:$\mathrm{SWL}(S_1) \equiv \mathrm{SWL}(S_2)$. CCWL introduces higher-dimensional structures (faces), and the refinement process divides nodes or edges into multiple color categories, enabling the differentiation between two structures in Figure B.4.

**Case 5** (SWL $\prec$ CCWL) Figure B.3 example (b) also considers two 1-dimensional simplicial complexes $S_1$ and $S_2$. $S_1$ repents $C_6 \cup C_6$: two 6-cycles glued along an edge($e_{16}$), and $S_2$ denotes $C_5 \cup C_5$: two 5-cycles connected by a bridge edge($e_{16}$). Specifically, nodes aggregate colors from their incident edges, while edges aggregate colors from their two endpoint nodes, since no 2-simplices are present, the upper neighborhoods of all edges are empty. When restricted to 1-dimensional simplicial complexes, SWL test degenerates to the classical 1-dimensional Weisfeiler-Lehman (1-WL) color refinement on graphs. With a uniform initialization, all nodes and all edges share the same color, the first refinement round reveals only degree information. In both graphs considered here, exactly two nodes have degree three and the remaining eight nodes have degree two, leading to an identical partition of nodes into degree-3 and degree-2 color classes. In subsequent iterations, each degree-3 node has an identical local neighborhood structure, consisting of two degree-2 neighbors and one degree-3 neighbor, and the neighborhoods of all degree-2 nodes are likewise locally indistinguishable as color information propagates along the cycle structures. The process stabilizes after the second iteration, producing identical node and edge color distributions for both graphs, and the SWL test fails to distinguish two simplicial complexes. If we lift two simplicial complexes to combinatorial complexes $f(S_1), f(S_2)$, as Figure B.5 shows CCWL can distinguish them.

*Figure B.2.* Combinatorial complex Weisfeiler-Lehman can distinguish the lifting combinatorial complexes $f(G_1), f(G_2)$.

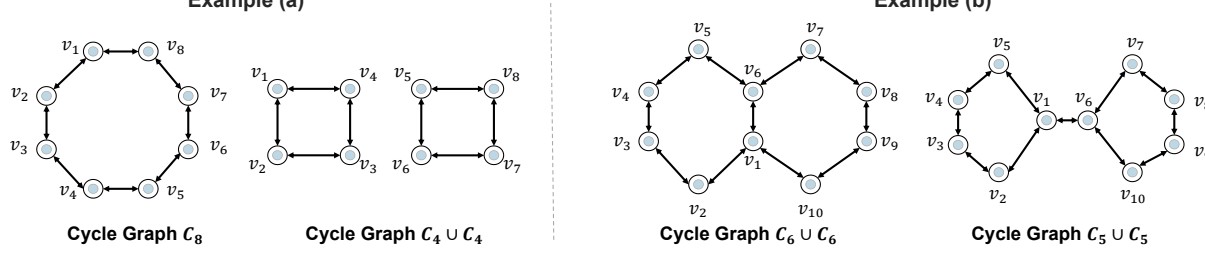

*Figure B.3.* Regular simplicial complexes are non-isomorphic (Simplicial Weisfeiler-Lehman cannot distinguish them). Example (a) $C_8$ : the cycle graph on 8 nodes, $C_4 \cup C_4$: the disjoint union of two 4-cycle graphs. Example (b) $C_6 \cup C_6$: two 6-cycles glued along an edge($e_{16}$), $C_5 \cup C_5$: two 5-cycles connected by a bridge edge($e_{16}$). Note: The boundary of a hollow quadrilateral is a 1-dimensional simple complex, while a solid quadrilateral (as a 2-dimensional region) cannot be directly treated as a simplex; it can be transformed into a 2-dimensional simple complex through triangulation.

**(a) Combinatorial Complex Weisfeiler–Lehman Test on Lifting Simplicial Complexes** $f(\mathcal{S}_1)$

$\boldsymbol{v_1}: \{v_1, \{v_2, v_8\}, \{e_{12}, e_{18}\}\}$ $\boldsymbol{v_2}: \{v_2, \{v_1, v_3\}, \{e_{12}, e_{23}\}\}$ $\boldsymbol{v_3}: \{v_3, \{v_2, v_4\}, \{e_{23}, e_{34}\}\}$ $\boldsymbol{v_4}: \{v_4, \{v_3, v_5\}, \{e_{34}, e_{45}\}\}$

$\boldsymbol{v_5}: \{v_5, \{v_4, v_6\}, \{e_{45}, e_{56}\}\}$ $\boldsymbol{v_6}: \{v_6, \{v_5, v_7\}, \{e_{56}, e_{67}\}\}$ $\boldsymbol{v_7}: \{v_7, \{v_6, v_8\}, \{e_{67}, e_{68}\}\}$ $\boldsymbol{v_8}: \{v_8, \{v_1, v_7\}, \{e_{18}, e_{78}\}\}$

$\boldsymbol{e_{12}}: \{e_{12}, \{v_1, v_2\}, \{e_{18}, e_{23}\}, \{f_1\}\}$ $\boldsymbol{e_{23}}: \{e_{23}, \{v_2, v_3\}, \{e_{12}, e_{34}\}, \{f_1\}\}$ $\boldsymbol{e_{34}}: \{e_{34}, \{v_3, v_4\}, \{e_{23}, e_{45}\}, \{f_1\}\}$

$\boldsymbol{e_{45}}: \{e_{45}, \{v_4, v_5\}, \{e_{34}, e_{56}\}, \{f_1\}\}$ $\boldsymbol{e_{56}}: \{e_{56}, \{v_5, v_6\}, \{e_{45}, e_{67}\}, \{f_1\}\}$ $\boldsymbol{e_{67}}: \{e_{67}, \{v_6, v_7\}, \{e_{56}, e_{78}\}, \{f_1\}\}$

$\boldsymbol{e_{78}}: \{e_{78}, \{v_7, v_8\}, \{e_{67}, e_{18}\}, \{f_1\}\}$ $\boldsymbol{e_{18}}: \{e_{18}, \{v_., v_8\}, \{e_{12}, e_{78}\}, \{f_1\}\}$

$\boldsymbol{f_1}: \{f_1, \{e_{12}, e_{23}, e_{34}, e_{45}, e_{56}, e_{67}, e_{78}, e_{18}\}\}$

**(b) Combinatorial Complex Weisfeiler–Lehman Test on Lifting Simplicial Complexes** $f(\mathcal{S}_2)$

$\boldsymbol{v_1}: \{v_1, \{v_2, v_4\}, \{e_{12}, e_{14}\}\}$ $\boldsymbol{v_2}: \{v_2, \{v_1, v_3\}, \{e_{12}, e_{23}\}\}$ $\boldsymbol{v_3}: \{v_3, \{v_2, v_4\}, \{e_{23}, e_{34}\}\}$ $\boldsymbol{v_4}: \{v_4, \{v_3, v_1\}, \{e_{34}, e_{14}\}\}$

$\boldsymbol{v_5}: \{v_5, \{v_4, v_6\}, \{e_{45}, e_{56}\}\}$ $\boldsymbol{v_6}: \{v_6, \{v_5, v_7\}, \{e_{56}, e_{67}\}\}$ $\boldsymbol{v_7}: \{v_7, \{v_6, v_8\}, \{e_{67}, e_{68}\}\}$ $\boldsymbol{v_8}: \{v_8, \{v_5, v_7\}, \{e_{58}, e_{78}\}\}$

$\boldsymbol{e_{12}}: \{e_{12}, \{v_1, v_2\}, \{e_{14}, e_{23}\}, \{f_1\}\}$ $\boldsymbol{e_{23}}: \{e_{23}, \{v_2, v_3\}, \{e_{12}, e_{34}\}, \{f_1\}\}$ $\boldsymbol{e_{34}}: \{e_{34}, \{v_3, v_4\}, \{e_{23}, e_{14}\}, \{f_1\}\}$

$\boldsymbol{e_{14}}: \{e_{14}, \{v_4, v_1\}, \{e_{12}, e_{34}\}, \{f_1\}\}$ $\boldsymbol{e_{56}}: \{e_{56}, \{v_5, v_6\}, \{e_{45}, e_{67}\}, \{f_2\}\}$ $\boldsymbol{e_{67}}: \{e_{67}, \{v_6, v_7\}, \{e_{56}, e_{78}\}, \{f_2\}\}$

$\boldsymbol{e_{78}}: \{e_{78}, \{v_7, v_8\}, \{e_{67}, e_{58}\}, \{f_2\}\}$ $\boldsymbol{e_{58}}: \{e_{58}, \{v_5, v_8\}, \{e_{56}, e_{78}\}, \{f_2\}\}$

$\boldsymbol{f_1}: \{f_1, \{e_{12}, e_{23}, e_{34}, e_{14}\}\}$ $\boldsymbol{f_2}: \{f_2, \{e_{56}, e_{67}, e_{78}, e_{58}\}\}$

**(c) Update Label of Combinatorial Complex Weisfeiler–Lehman Test on Lifting Simplicial Complexes**

$f(\mathcal{S}_1)$
$\boldsymbol{v_1, v_2, v_3, v_4, v_5, v_6, v_7, v_8}: \{\circ, \{\circ, \circ\}, \{\circ, \circ\}\}$
$\boldsymbol{e_{12}, e_{23}, e_{34}, e_{14}, e_{56}, e_{67}, e_{78}, e_{58}}: \{\circ, \{\circ, \circ\}, \{\circ, \circ\}, \{\bullet\}\}$
$\boldsymbol{f_1}: \{\bullet, \{\circ, \circ, \circ, \circ, \circ, \circ, \circ, \circ\}\}$

$f(\mathcal{S}_2)$
$\boldsymbol{v_1, v_2, v_3, v_4, v_5, v_6, v_7, v_8}: \{\circ, \{\circ, \circ\}, \{\circ, \circ\}\}$
$\boldsymbol{e_{12}, e_{23}, e_{34}, e_{14}, e_{56}, e_{67}, e_{78}, e_{58}}: \{\circ, \{\circ, \circ\}, \{\circ, \circ\}, \{\bullet\}\}$
$\boldsymbol{f_1, f_2}: \{\bullet, \{\circ, \circ, \circ, \circ\}\}$

$CCWL(f(\mathcal{S}_1)) \neq CCWL(f(\mathcal{S}_2))$

**Lifting simple complexes** $f(\mathcal{S}_1), f(\mathcal{S}_2)$
**are non-isomorphic**

*Figure B.4.* Two simplicial complexes ($C_8$ : the cycle graph on 8 nodes, $C_4 \cup C_4$: the disjoint union of two 4-cycle graphs.) are non-isomorphic. Simplicial Weisfeiler-Lehman cannot distinguish them, but Combinatorial complex Weisfeiler-Lehman can distinguish the lifting combinatorial complexes $f(S_1), f(S_2)$.

**Proposition B.8.** *There exists a lifting* $f : \mathcal{C} \to \mathcal{CC}$ *from cellular complexes to combinatorial complexes such that CCWL applied to* $f(\mathcal{C})$ *is strictly more expressive than the Cellular Weisfeiler-Lehman (CWL) test.*

*Proof.* Similar to the proof of Proposition B.5, we can also prove Proposition B.6, Proposition B.7 and Proposition B.8. $\square$

**Case 5** (CWL $\prec$ CCWL) Cellular WL operates on a quotient representation of a cellular complex, in which lower-dimensional cells and the types of incidence relations are collapsed into an untyped adjacency graph, boundary multiplicities and rank-specific interactions are irreversibly lost prior to refinement. There exist two 2-dimensional combinatorial complexes $\mathcal{CC}_1$ and $\mathcal{CC}_2$. they share the same nodes set as $V = \{v_1, v_2, v_3, v_4, v_5, v_6\}$. we consider the common edges set as $e_{12}, e_{13}, e_{23}, e_{14}, e_{15}, e_{45}, e_{26}, e_{36}, e_{46}, e_{56}$, here each node with four connections, and each edge appears in two 2-cells. Given two combinatorial complexes have 4 triangular 2-cells as

- $\mathcal{CC}_1 = \{f_1 = \{v_1, v_2, v_3\}, f_2 = \{v_1, v_4, v_5\}, f_3 = \{v_2, v_3, v_6\}, f_4 = \{v_4, v_5, v_6\}\}$ , here $f_1 \cap f_2 = \{v_1\}$, $f_1 \cap f_3 = \{v_2, v_3\}$, $f_1 \cap f_2 = \emptyset$, $f_2 \cap f_3 = \emptyset$, $f_2 \cap f_4 = \{v_4, v_5\}$, $f_3 \cap f_4 = \{v_6\}$,

- $\mathcal{CC}_1 = \{g_1 = \{v_1, v_2, v_3\}, g_2 = \{v_1, v_4, v_5\}, g_3 = \{v_2, v_4, v_6\}, g_4 = \{v_3, v_5, v_6\}\}$, here $g_1 \cap g_2 = \{v_1\}$, $g_1 \cap g_3 = \{v_2\}$, $g_1 \cap g_4 = \{v_3\}$, $g_2 \cap g_3 = \{v_4\}$, $g_2 \cap g_4 = \{v_5\}$, $g_3 \cap g_4 = \{v_6\}$,

According to the CWL test, we suppose the initial colors: all 0-cells $c_0^{(0)} = \{\bullet\}$, all 1-cells $c_1^{(0)} = \{\bullet\}$ and $c_2^{(0)} = \{\bullet\}$. CWL updates colors with the rules 1-cells aggregate upper neighbors (incident 2-cells), and 2-cells aggregate lower neighbors (boundary 1-cells). Because of every edge appears in exactly two 2-cells in $\mathcal{CC}_1$ and $\mathcal{CC}_2$, all edges receive the same updated color as $c_1^{(1)} = \text{HASH}(\bullet, \{\bullet, \bullet\})$ and each 2-cell aggregates three boundary edges, $c_2^{(1)} = \text{HASH}(\bullet, \{\bullet, \bullet, \bullet\})$. Followed with CWL update rules, all 2-cells get identical colors $c_2^{(t)}(\mathcal{CC}_1) = c_2^{(t)}(\mathcal{CC}_2)$ at any iteration $t$, and fails to distinguish them.

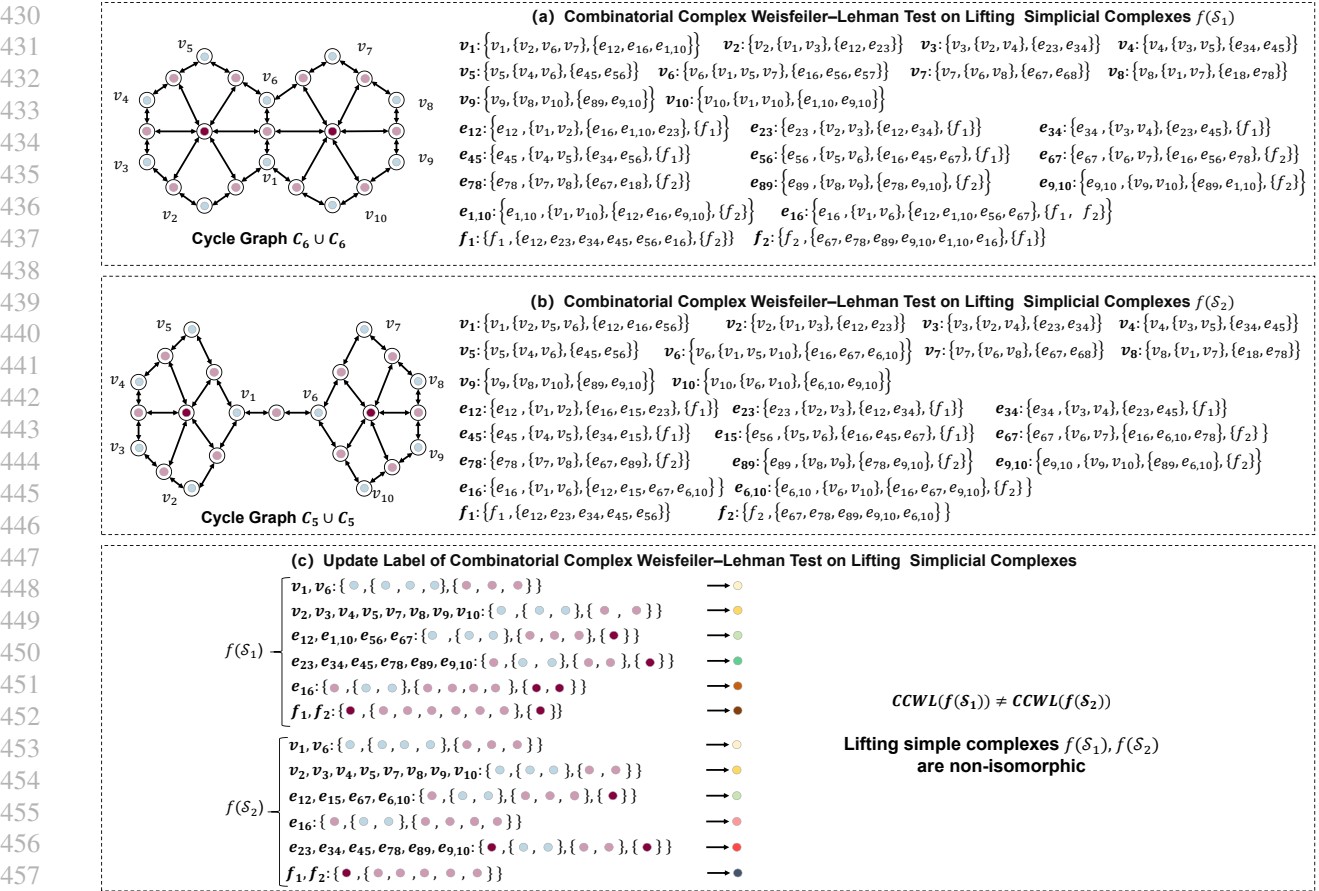

*Figure B.5.* Two simplicial complexes are non-isomorphic ($C_6 \cup C_6$: two 6-cycles glued along an edge($e_{16}$), $C_5 \cup C_5$: two 5-cycles connected by a bridge edge($e_{16}$)). Simplicial Weisfeiler-Lehman cannot distinguish them, but Combinatorial complex Weisfeiler-Lehman can distinguish the lifting combinatorial complexes $f(S_1), f(S_2)$.

In contrast, CCWL operates directly on the full incidence structure of the combinatorial complex, allowing higher-dimensional cells to exchange messages based on arbitrary non-empty intersections, regardless of the underlying skeleton. we also suppose the initial colors: all 0-cells $c_0^{(0)} = \{\circ\}$, all 1-cells $c_1^{(0)} = \{\bullet\}$ and $c_2^{(0)} = \{\bullet\}$. At iteration $t = 1$, each 2-cell $\sigma$ updates its color by aggregating information from all other 2-cells $\tau \neq \sigma(\sigma \cap \tau \neq \emptyset)$. Importantly, CCWL preserves the size of the intersection $\sigma \cap \tau$ as part of the message as

$$c_2^{(1)}(\sigma) = \text{HASH}\big(\bullet, \{(|\sigma \cap \tau|, \bullet)|\tau \in \mathcal{CC}, \tau \neq \sigma, \sigma \neq \tau \neq \emptyset\}\big) \tag{24}$$

In the combinatorial complexes $\mathcal{CC}_1$, the 2-cell $f_1$ intersects $f_2$ with node $v_1$, $f_2$ with edges $v_2, v_3$, $f_4$ with $\emptyset$, then its neighbor multiset is $c_2^{(1)}(f_1) = \{\{\bullet, \{\bullet, \circ\}, \{\bullet, \circ\}, \{\bullet\}\}\}$, but the combinatorial complexes $\mathcal{CC}_2$, the 2-cell $f_1$ intersects $f_2$ with node $v_1$, $f_2$ with edges $v_2$, $f_4$ with $v_3$, then its neighbor multiset is $c_2^{(1)}(g_1) = \{\{\bullet, \{\bullet, \circ\}, \{\bullet, \circ\}, \{\bullet, \circ\}\}\}$. From the multisets, we obtain $c_2^{(t)}(\mathcal{CC}_1) \neq c_2^{(t)}(\mathcal{CC}_2)$. Hence, CCWL can distinguish $\mathcal{CC}_1$ and $\mathcal{CC}_2$ at the first iteration.

***Proof for Theorem 4.11.*** Let $\mathcal{G}, \mathcal{H}, \mathcal{S}$, and $\mathcal{C}$ denote the classes of graphs, hypergraphs, simplicial complexes, and cellular complexes, respectively. We prove this theorem by combining the Propositions B.5, Propositions B.6, Propositions B.7 and Propositions B.8. There exist faithful liftings from each of these structures into combinatorial complexes $\mathcal{CC}$ such that the Combinatorial Complex Weisfeiler Lehman test simulates the corresponding classical WL tests, namely, 1-WL, Hypergraph WL, Simplicial WL and Cellular WL, which is strictly more expressive in each case.

For each discrete structure, we define a faithful lifting map $f$ into the category of combinatorial complexes $\mathcal{CC}$: For graphs $\mathcal{G}$, Proposition B.5 constructs $f_G : \mathcal{G} \to \mathcal{CC}$ preserving incidence via 0- and 1-cells, showing that CCWL on $f_G(\mathcal{G})$ simulates

1-WL by matching node neighbourhood aggregation through 0-cell updates. For hypergraphs $\mathcal{H}$, Proposition B.6 constructs $f_H : \mathcal{H} \to \mathcal{CC}$ preserving incidence via 0- and 1-cells, showing that CCWL on $f_H(\mathcal{H})$ simulates 1-WL by matching node neighbourhood aggregation through 0-cell updates. For simplicial complexes $\mathcal{K}$, Proposition B.7 gives $f_K : \mathcal{K} \to \mathcal{CC}$ that preserves simplex-cell correspondence and face incidences. The CCWL update rules on $k$-cells directly simulate the lower and upper coface aggregation of SWL. Proposition B.8. For cellular complexes $\mathcal{C}$, the construction in the previous proposition defines $f_C : \mathcal{C} \to \mathcal{CC}$ preserving all boundary maps. Since CWL refines cells based on boundary and coboundary multisets, and CCWL aggregates over exactly these incidence relations, the simulation follows immediately. For hypergraphs $\mathcal{H}$, a similar lifting $f_H : \mathcal{H} \to \mathcal{CC}$ can be defined: map each node to a 0-cell, each hyperedge of size $k + 1$ to a $k$-cell with boundary consisting of the corresponding 0-cells. Standard results show that this embedding allows CCWL to simulate Hypergraph WL (which refines nodes based on hyperedge memberships and vice versa) via appropriate aggregation over 0-cells and their incident higher-cells. In all cases, the lifting $f$ is faithful ($A \cong B \iff f(A) \cong f(B)$), and the local aggregation rules of CCWL generalize those of the respective WL variants when restricted to the base cell types.

Each classical WL test is limited by its local update rule. The 1-WL test cannot distinguish non-isomorphic regular graphs with identical local neighborhoods—most famously, Cai-Fürer-Immerman (CFI) graph pairs. Similarly, Hypergraph WL fails on high-uniformity hypergraphs where symmetries persist across all local refinements. For simplicial and cellular complexes, SWL and CWL are blind to global topological invariants: they treat as equivalent any two structures that share the same local incidence patterns but differ in global parity or higher-order connectivity, as demonstrated by CFI-style constructions lifted to these domains.

The liftings described above circumvent these limitations by embedding each structure into a combinatorial complex enriched with CFI-style gadgets. Specifically, we attach $(k+1)$-cells—such as twisted or suspended configurations—whose internal combinatorial structure encodes a global invariant like total parity. Crucially, these gadgets do not modify the original object's local neighborhood structure; hence, classical WL tests remain oblivious to them. In the lifted complex, however, the coface multisets of certain $k$-cells differ between non-isomorphic instances. Since CCWL aggregates over both boundary and coboundary relations, it detects these discrepancies and assigns distinct colors during refinement. Consequently, for each domain there exists a pair $(A, B)$ such that $A \ncong B$, the corresponding classical WL test fails to distinguish $A$ and $B$, yet CCWL distinguishes $f(A)$ and $f(B)$. This establishes that, under appropriate liftings, CCWL is strictly more expressive than 1-WL, Hypergraph WL, SWL, and CWL. $\square$

**Corollary B.9.** *The Combinatorial Complex Weisfeiler-Lehman (CCWL) test is strictly more expressive than Graph-WL(1-WL), Hypergraph-WL(HWL), Simplicial WL (SWL), and Cellular WL (CWL). That is,*

$$\text{1-WL} \prec \text{CCWL}, \quad \text{HWL} \prec \text{CCWL}, \quad \text{SWL} \prec \text{CCWL}, \quad \text{CWL} \prec \text{CCWL}.$$

*Consequently, there exist pairs of non-isomorphic structures indistinguishable by each classical test but distinguished by CCWL under an appropriate faithful lifting into combinatorial complexes.*

*Proof.* Building on Proposition B.5, Proposition B.6, Proposition B.7 and Proposition B.8, this Corollary holds. $\square$

## C. Building Powerful Combinatorial Complex Neural Networks.

### C.1. Bridge between CCWL test and CCNNs

**Building Powerful Combinatorial Complex Isomorphic Networks.** The expressive capacity of Combinatorial Complex Neural Networks (CCNNs) can be characterized through a generalized Weisfeiler-Lehman (WL) refinement process defined over combinatorial complexes. Let $\mathcal{N} = \{\mathcal{B}, \mathcal{C}, \mathcal{N}_\uparrow, \mathcal{N}_\downarrow\}$ be a set of neighborhood functions defined on $\mathcal{CC}$. Then we propose Combinatorial Complex Isomorphic Networks(CCIN) At each iteration, the color of a cell $\sigma$ is updated based on its current state and the colors of its neighbors across all neighborhood types:

$$c^{(t+1)}(\sigma) = \text{HASH}(c^{(t)}(\sigma), \bigcup_{\mathcal{N}} \{(\mathcal{N}, \{\!\!\{c^{(t)}(\tau) : \tau \in \mathcal{N}(\sigma)\}\!\!\})\}), \tag{25}$$

where $\{\!\!\{\cdot\}\!\!\}$ denotes a multiset, and $\bigcup_{\mathcal{N}}$ represents a labeled union across distinct neighborhood types. This iterative refinement continues until color stabilization, as defined in Condition 4.9. Two combinatorial complexes $\mathcal{CC}_1$ and $\mathcal{CC}_2$ are considered non-isomorphic if there exists an iteration $t$ such that

$$\{\!\!\{c^{(t)}(\sigma) : \sigma \in \mathcal{CC}_1\}\!\!\} \neq \{\!\!\{c^{(t)}(\sigma) : \sigma \in \mathcal{CC}_2\}\!\!\}. \tag{26}$$

The continuous formulation of CCNNs can be viewed as a differentiable fuction $\mathcal{T}$ of this discrete refinement scheme. Specifically, we replace the injective hash operation with learnable, and permutation-invariant transformations, as follows:

$$h^{(t+1)}(\sigma) = \phi\left(h^{(t)}(\sigma), \bigoplus_{\mathcal{N}} \{\!\{h^{(t)}(\tau) : \tau \in \mathcal{N}(\sigma)\}\!\})\right), \tag{27}$$

where $\bigoplus$ and $\bigotimes$ denote permutation-invariant aggregation and fusion operators, respectively, and $\phi$, $\psi_{\mathcal{N}}$ are learnable, injective mappings ensuring the expressive equivalence to the underlying CCWL refinement. Therefore, Eq. (27) establishes a one-to-one correspondence between the discrete color refinement of CCWL and the continuous feature propagation in CCNNs, implying that the network attains at least the same discriminative power as the CCWL test. Based on the theoretical validity of the CCWL test, we can establish a neural network to learn the feature transfer of combinatorial complexes from the coloring problem in the WL test. This relabeling process is learned through a mapping function, where the input label of the cell is $c_\sigma^{(t)}$, and the output feature is $h_\sigma^{(t)}$. The following bridge is established between the CCWL test and CCIN:

**Lemma C.1.** *Suppose an injective function $\mathcal{T} : \mathcal{CC}_1 \to \mathcal{CC}_2$, and let the labels $c_\sigma^{(t)}$, and features $h_\sigma^{(t)}$ of node, edge and face in combinatorial complex $\mathcal{CC}$ at iteration t, respectively. If the conditions hold for $k \leq t$, $\left\{\!\!\left\{ c_{\mathcal{CC}_1,\sigma_1}^{(k)} \right\}\!\!\right\} = \left\{\!\!\left\{ c_{\mathcal{CC}_2,\sigma_2}^{(k)} \right\}\!\!\right\}$, and $k \leq t-1$, $h_{\mathcal{CC}_1,\sigma_1}^{(k)} = h_{\mathcal{CC}_2,\sigma_2}^{(k)}$ holds, it follows that $h_{\mathcal{CC}_1,\sigma_1}^{(t)} = h_{\mathcal{CC}_2,\sigma_2}^{(t)}$ at iterations $k$.*

*Proof.* At iteration $k = 0$, all the initial colors are consistent, the condition holds.

By induction, it follows that for all $k \leq t-1$, $\left\{\!\!\left\{ c_{\mathcal{CC}_1,\sigma_1}^{(k)} \right\}\!\!\right\} = \left\{\!\!\left\{ c_{\mathcal{CC}_2,\sigma_2}^{(k)} \right\}\!\!\right\}$, which implies $h_{\mathcal{CC}_1,\sigma_1}^{(k)} = h_{\mathcal{CC}_2,\sigma_2}^{(k)}$. At iteration $t$, it also holds that $\left\{\!\!\left\{ c_{\mathcal{CC}_1,\sigma_1}^{(t)} \right\}\!\!\right\} = \left\{\!\!\left\{ c_{\mathcal{CC}_2,\sigma_2}^{(t)} \right\}\!\!\right\}$, we can conclude:

$$\left\{\!\left\{ \left( c_{\mathcal{CC}_1,\sigma_1}^{(t)}, \bigcup_{\mathcal{N}} \{ (\mathcal{N}, \{\!\{c_{\mathcal{CC}_1,\tau}^{(t)} : \tau \in \mathcal{N}(\sigma_1)\}\!\}) \} \right) \right\}\!\right\} = \left\{\!\left\{ \left( c_{\mathcal{CC}_2,\sigma_2}^{(t)}, \bigcup_{\mathcal{N}} \{ (\mathcal{N}, \{\!\{c_{\mathcal{CC}_2,\tau}^{(t)} : \tau \in \mathcal{N}(\sigma_2)\}\!\}) \} \right) \right\}\!\right\}, \tag{28}$$

By the inductive hypothesis and mapping function $\mathcal{T}$ is injective, we obtain:

$$\left\{\!\left\{ \left( h_{\mathcal{CC}_1,\sigma_1}^{(t)}, \bigoplus_{\mathcal{N}} \{\!\{h^{(t)}(\tau) : \tau \in \mathcal{N}(\sigma_1)\}\!\} \right) \right\}\!\right\} = \left\{\!\left\{ \left( h_{\mathcal{CC}_2,\sigma_2}^{(t)}, \bigoplus_{\mathcal{N}} \{\!\{h^{(t)}(\tau) : \tau \in \mathcal{N}(\sigma_2)\}\!\} \right) \right\}\!\right\}, \tag{29}$$

resulting in the CCIN function's aggregation function generates the same output $h_{\mathcal{CC}_1,\sigma_1}^{(t)} = h_{\mathcal{CC}_2,\sigma_2}^{(t)}$. $\square$

**Proposition C.2.** *Given two non-isomorphic combinatorial complexes $\mathcal{CC}_1$ and $\mathcal{CC}_2$, if CCIN layer $\mathcal{T}$ can distinguish them by $\mathcal{T}(\mathcal{CC}_1) \neq \mathcal{T}(\mathcal{CC}_2)$, then 1-CCWL test also decides $\mathcal{CC}_1 \neq \mathcal{CC}_2$.*

*Proof.* Suppose there exists $k \geq 0$ such that after $k$ iterations, we have $\mathcal{T}(CC_1) \neq \mathcal{T}(CC_2)$ , Due to the features $h_{CC} = \mathcal{T}(CC)$, $h_{CC_1}^{(k)} \neq h_{CC_2}^{(k)}$, but the CCWL test cannot distinguish non-isomorphic $CC_1$ and $CC_2$, i.e., $c_{CC_1}^{(k)} = c_{CC_2}^{(k)}$.

We now prove this Proposition by induction. At the initiation $k = 0$ , then $c_{CC_1}^{(0)} = c_{CC_2}^{(0)}$ implies the initial features $h_{CC_1}^{(0)} = h_{CC_2}^{(0)}$. This means that both the WL test and $\mathcal{T}$ start with the same labels and features, which contradicts our assumption. For $k \geq 0$ , then $c_{CC_1}^{(k-1)} = c_{CC_2}^{(k-1)}$ and $h_{CC_1}^{(k-1)} = h_{CC_2}^{(k-1)}$ . Given that $c_{H_1}^{(k)} = c_{H_2}^{(k)}$ , by Lemma C.1, we obtain $c_{CC_1,\sigma_1}^{(k)} = c_{CC_2,\sigma_2}^{(k)}$, $h_{CC_1,\sigma_1}^{(k)} = h_{CC_2,\sigma_2}^{(k)}$, which implies there exists a mapping function $\mathcal{R}$ holds that $\mathcal{R}(c_{CC,\sigma}^{(k)}) \to h_{CC,\sigma}^{(k)}$. Because $c_{CC_1,\sigma_1}^{(k)} = c_{CC_2,\sigma_2}^{(k)}$, we can obtain the condition

$$\{\!\{c_{CC_1,\sigma_1}^{(k)}\}\!\}_{\sigma_1 \in \mathcal{CC}_1} = \{\!\{c_{CC_2,\sigma_2}^{(k)}\}\!\}_{\sigma_2 \in \mathcal{CC}_2}, \quad \text{s.t.} \quad \{\!\{h_{CC_1,\sigma_1}^{(k)}\}\!\}_{\sigma_1 \in \mathcal{CC}_1} = \{\!\{h_{CC_2,\sigma_2}^{(k)}\}\!\}_{\sigma_2 \in \mathcal{CC}_2}, \tag{30}$$

With the mapping function $\mathcal{R}$, we can conclude that

$$\{\!\{h_{CC_1,\sigma_1}^{(k)}\}\!\}_{\sigma_1 \in \mathcal{CC}_1} = \{\!\{\mathcal{R}(c_{CC_1,\sigma_1}^{(k)})\}\!\}_{\sigma_1 \in \mathcal{CC}_1} = \{\!\{\mathcal{R}(c_{CC_2,\sigma_2}^{(k)})\}\!\}_{\sigma_2 \in \mathcal{CC}_2} = \{\!\{h_{CC_2,\sigma_2}^{(k)}\}\!\}_{\sigma_2 \in \mathcal{CC}_2}. \tag{31}$$

However, this conclusion contradicts the assumption $\mathcal{T}(CC_1) \neq \mathcal{T}(CC_2)$, the proposition has been proved. $\square$

**Proof of Theorem 5.1.** Suppose there exists a injective mapping $\mathcal{R}^{(k)}$ such that $\mathcal{R}^{(k)}(c_{\mathcal{CC},\sigma}^{(k)}) \to h_{\mathcal{CC},\sigma}^{(k)}$. For $k = 0$, $\mathcal{R}^{(0)}$ is the identity mapping in the case of $c_{\mathcal{CC},\sigma}^{(0)}$ and $h_{\mathcal{CC},\sigma}^{(0)}$ are the same. For $k \geq 0$, assuming the injective mapping $\mathcal{R}^{(k)}$ exists, we show that it holds at iterations $k+1$ based on Equation 27, $h_{\mathcal{CC},\sigma}^{(t+1)} = \phi\left(h^{(t)}(\sigma), \bigoplus_{\mathcal{N}} \{\!\{h^{(t)}(\tau) : \tau \in \mathcal{N}(\sigma)\}\!\}\right)$,

Since the composition of injective functions $\phi$ and $\bigoplus_{\mathcal{N}}$ are the injective function, we can rewrite this as,

$$
\begin{aligned}
h_{\mathcal{CC},\sigma}^{(t+1)} &= \varphi\left(h_\sigma^{(t)}, \{\!\{h_\tau^{(t)} : \tau \in \mathcal{N}(\sigma)\}\!\}_{\mathcal{N}(\sigma)\in\mathcal{N}}\right) = \varphi\left(\mathcal{R}^{(t)}(c_\sigma^{(t)}), \{\!\{\mathcal{R}^{(t)}(c_\tau^{(t)}) : \tau \in \mathcal{N}(\sigma)\}\!\}_{\mathcal{N}(\sigma)\in\mathcal{N}}\right) \\
&= \mathcal{R}^{(t)}\left(\{\!\{(c_\sigma^{(t)}, \{\{c_\tau^{(t)}\}\}\}_{\tau\in\mathcal{N}(\sigma)}\}\!\}\right) = \mathcal{R}^{(k+1)}\left(c_\sigma^{(t+1)}\right)
\end{aligned}
\tag{32}
$$

where $\varphi$ is the injective function due to it can be induced with $\phi$ and $\bigoplus_{\mathcal{N}}$, and $\mathcal{R}^{(k+1)}$ is also the injective function. By induction, there also exists a injective mapping $\mathcal{R}^{(k)}$ such that $\mathcal{R}^{(k)}\left(c_{\mathcal{CC},\sigma}^{(k)}\right) \to h_{\mathcal{CC},\sigma}^{(k)}$. If at iterations $k$, $c_{\mathcal{CC}_1}^{(k)} \neq c_{\mathcal{CC}_2}^{(k)}$, then $\left\{\!\left\{c_{\mathcal{CC}_1,\sigma_1}^{(k)}\right\}\!\right\}_{\sigma_1\in\mathcal{CC}_1} \neq \left\{\!\left\{c_{\mathcal{CC}_2,\sigma_2}^{(k)}\right\}\!\right\}_{\sigma_2\in\mathcal{CC}_2}$. By the injectivity assumption, we can derive $h_{\mathcal{CC}_1}^{(k)} \neq h_{\mathcal{CC}_2}^{(k)}$, then combine with Proposition C.2. CCIN is as powerful as CCWL when we employ an injective neighborhood aggregators.

$\square$

## C.2. Generalization of Combinatorial Complexes Message Passing

To clarify the higher-order message passing on a $CC$, we discuss the combinatorial complexes message passing in this section. For a cell $\sigma$, we instantiate the above general rule into four specific message computations, corresponding to the neighborhood types, the embeddings from the four types are integrated as

$$
\mathbf{h}_\sigma^{(t+1)} = \text{Update}\left(\mathbf{h}_\sigma^{(t)}, m_{\mathcal{B}}^{(t+1)}, m_{\mathcal{C}}^{(t+1)}, m_{\mathcal{N}_\downarrow}^{(t+1)}, m_{\mathcal{N}_\uparrow}^{(t+1)}\right),
\tag{33}
$$

where Update is a learnable fusion function, e.g., concatenation followed by a linear transformation (Multilayer Perceptron layer). In details, boundary message aggregates information from lower-dimensional constituents (e.g., edges of a face). The co-boundary message propagates from higher-dimensional cells enclosing $\sigma$. The lower adjacency message models lateral interactions through shared boundaries. The upper adjacency message captures relations through shared co-boundaries.

We introduce the interpretation of isomorphic networks on four neighborhood functions. Let $\mathcal{CC}$ be a combinatorial complex, we consider the following general message-passing update rule for a cell $\sigma \in \mathcal{CC}$ of rank $k = \text{rk}(\sigma)$ at layer $l+1$:

$$
h_\sigma^{(l+1)} = \phi_k^{(l)}\left((1 + \epsilon_k^{(l)})h_\sigma^{(l)} + \sum_{\mathcal{N}\in\mathcal{N}_\mathcal{C}} \gamma_{\mathcal{N},k}^{(l)} \sum_{\tau\in\mathcal{N}(\sigma)} \psi_{\mathcal{N},k}^{(l)}(h_\sigma^{(l)}, h_\tau^{(l)})\right),
\tag{34}
$$

where $h_\sigma^0 = h_\sigma$ represent the initial features, $h_\sigma^{l+1} \in \mathbb{R}^{F^{l+1}}$, $N_\mathcal{C}$ is a collection of neighborhood functions. The functions $\psi_{\mathcal{N},rk(\cdot)} : \mathbb{R}^{F^l} \to \mathbb{R}^{F^{l+1}}$ and the update function $\phi$ are learnable functions, which are typically homogeneous across all neighborhoods and ranks. and $\psi_{\mathcal{N},k}(h_\sigma, h_\tau) = \eta_{\mathcal{N},\tau} \cdot (W_{\mathcal{N},k} \cdot [h_\sigma \| h_\tau])$. According to the Theorem 4.8, we further simplify these four neighborhood relations to upper and lower neighborhood functions, and obtain the following.

Let $\mathcal{C}$ be a combinatorial complex equipped with a collection of neighborhood functions $\mathcal{N}_\mathcal{C}$. We consider the following general message-passing update rule for a cell $\sigma \in \mathcal{C}$ of rank $k = \text{rk}(\sigma)$ at layer $l+1$:

$$
h_v^{(l+1)} = \phi^{(l)}\left((1 + \epsilon^{(l)})h_\sigma^{(l)} + \gamma_{\mathcal{N}_\downarrow}^{(l)} \sum_{\tau\in\mathcal{N}_\downarrow(\sigma)} \psi_{\mathcal{N}_\downarrow}^{(l)}(h_\sigma^{(l)}, h_\tau^{(l)}) + \gamma_{\mathcal{N}_\uparrow}^{(l)} \sum_{\tau\in\mathcal{N}_\uparrow(\sigma)} \psi_{\mathcal{N}_\uparrow}^{(l)}(h_\sigma^{(l)}, h_\tau^{(l)})\right),
\tag{35}
$$

where $\epsilon^{(l)} \in \mathbb{R}$ and $\gamma_{\mathcal{N}(\cdot)} \in \mathbb{R}$ are learnable scalars, $\psi_{\mathcal{N}(\cdot)}$ is a neighborhood- and rank-specific message MLP, $\psi_{\mathcal{N},k}(h_\sigma, h_\tau) = \text{MLP}_{\mathcal{N},k}\left([h_\sigma \| h_\tau]\right)$, $\text{MLP}_{\mathcal{N},k}(x) = W_{\mathcal{N},k}^{(2)} \text{ReLU}\left(W_{\mathcal{N},k}^{(1)}x\right)$, and $\phi_{\text{rk}(\sigma)}$ is a shared rank-wise update. The update uses Summation, Max and Mean aggregation function, guaranteeing permutation invariance within each neighborhood and preserving asymmetries across ranks and relation types. By refining the neighborhood function $\mathcal{N}_\mathcal{C}$ appropriately, the ability of CCIN can generalize on graph, hypergraph, simplxical and cellular, which imply the same separation guarantees for these variants. Moreover, rank-aware parameter shared with aggregation function $\psi$ and $\phi$ enables modeling of multi-rank interactions beyond standard GIN.

**Corollary C.3.** *CCIN layers are cell permutation equivariant.*

# D. Experiments

## D.1. Datasets Details

**Strongly Regular Graphs** We benchmark it on a synthetic dataset consisting of nine families of strongly regular (SR) graphs (Bodnar et al., 2021a;b). Strongly regular graphs represent "hard" instances of graph isomorphism, as their pairs cannot be distinguished by the 3-WL test. Two graphs are considered isomorphic if the Euclidean distance between their representations is below a fixed threshold $\epsilon$. Each graph is lifted to a d-dimensional complex whose size is $(d + 1)$ the size of the largest clique in the family to which it belongs. Details of SR families are illustrated in Table D.1.

*Table D.1.* Details of Strongly Regular Families

| Familty | (16,6,2,2) | (25,12,5,6) | (26,10,3,4) | (28,12,6,4) | (29,14,6,7) | (35,16,6,8) | (35,18,9,9) | (36,14,4,6) | (40,12,2,4) |
|---|---|---|---|---|---|---|---|---|---|
| Graphs Number | 2 | 15 | 10 | 4 | 41 | 3854 | 227 | 180 | 28 |

**TUDataset** TUDataset(Morris et al., 2020a) consists of bioinformatics and social networks. MUTAG (Kazius et al., 2005), PTC, PROTEINS(Borgwardt et al., 2005; Dobson & Doig, 2003) consist of graphs where nodes represent atoms and edges are chemical bonds. They are labeled according to biochemical properties. NCI1 and NCI109 (Wale et al., 2008) derived from the National Cancer Institute, these datasets contain chemical compounds screened for activity against various cancer cell lines. IMDB-B, IMDB-M, RDT-B and RDT-M are preprocessed movie collaboration and Reddit networks (Yanardag & Vishwanathan, 2015; Hu et al., 2020), where graphs represent interactions between actors or users, and the tasks involve classifying the network type or community structure.

**Peptides-func and Peptides-struct** SATPdb (Dwivedi et al., 2022b; Singh et al., 2016) are derived from 15,535 peptides with a total of 2.3 million nodes. Both datasets use the same set of graphs but differ in their prediction tasks. These graphs are constructed in such a way that requires long-range interactions (LRI) reasoning to achieve strong performance in a given task. In concrete terms, they are larger graphs: on average 150.94 nodes per graph, and on average 56.99 graph diameter.

*Table D.2.* Overview of the graph learning datasets.

| Dataset | Graphs | Avg. nodes | Avg. edges | Prediction task | Class | Metric |
|---|---|---|---|---|---|---|
| MUTAG | 188 | 17.93 | 19.79 | Classification | 2 | Accuracy |
| PTC | 344 | 25.56 | 25.96 | Classification | 2 | Accuracy |
| NCI1 | 4110 | 29.87 | 32.30 | Classification | 2 | Accuracy |
| NCI109 | 4127 | 29.68 | 32.13 | Classification | 2 | Accuracy |
| PROTEINS | 1113 | 39.06 | 72.82 | Classification | 2 | Accuracy |
| IMDB-B | 1000 | 19.77 | 96.53 | Classification | 2 | Accuracy |
| IMDB-M | 1 500 | 13.0 | 65.9 | Classification | 3 | Accuracy |
| RDT-B | 2000 | 429.6 | 497.8 | Classification | 2 | Accuracy |
| RDT-M | 5000 | 508.5 | | Classification | 5 | Accuracy |
| ZINK-Small | 12,000 | 23.2 | 24.9 | Regression | 12 | Mean Absolute Error |
| ZINC-FULL | 249,456 | 23.2 | 49.8 | Regression | 12 | Mean Absolute Error |
| OGBG-MOLHIV | 41,127 | 25.5 | 27.5 | Binary Classification | 2 | AUROC |
| OGBG-MOLBACE | 1513 | 34.1 | 36.9 | Binary Classification | 2 | AUROC |
| OGBG-MOLBBP | 2,039 | 24.1 | 51.9 | Binary Classification | 2 | AUROC |
| OGBG-MOLESOL | 19,717 | 13.3 | 13.7 | Regression | | Root Mean Square Error |
| ogbg-moltox21 | 7,831 | 18.6 | 19.3 | Regression | 12 | Root Mean Square Error |
| OGBG-MOLTOXCAST | 8,576 | 18.8 | 19.3 | Regression | 617 | Root Mean Square Error |
| OGBG-MOLPCBA | 437,929 | 26.0 | 28.1 | Regression | 128 | Root Mean Square Error |
| OGBG-PPA | 158,100 | 243.4 | 2,266.1 | Regression | 1 | Root Mean Square Error |
| PEPTIDES-FUNC | 15,535 | 150.9 | 307.3 | Multi-task Classification | 10 | Root Mean Square Error |
| PEPTIDES-STRUCT | 15,535 | 150.9 | 307.3 | Multi-task Regression | 11 | Avg. Precision |

**Open Graph Benchmark (OGB)** The Open Graph Benchmark (OGB) is a collection of large-scale and realistic graph datasets (Wu et al., 2018; Hu et al., 2020) designed to facilitate reproducible and rigorous evaluation of graph machine learning methods. Among its offerings, the molecular property prediction suite, which commonly refers to as the OGB-LSC and OGB-Mol datasets, which comprises twelve standardized benchmarks: MOLHIV, MOLTOX21, MOLTOXCAST, MOLBACE, MOLBBBP, MOLCLINTOX, MOLSIDER, MOLESOL, MOLFREESOLV, and MOLLIPO. These datasets are derived from real-world chemical and biomedical applications, where each graph represents a molecule (with atoms as nodes and bonds as edges), and the task is to predict properties such as toxicity (MOLTOX21, MOLTOXCAST,

MOLCLINTOX), bioactivity (MOLHIV, MOLBACE, MOLBBBP), side effects (MOLSIDER), or physical solubility characteristics (MOLESOL, MOLFREESOLV, MOLLIPO). All datasets follow standardized train/validation/test splits based on molecular scaffolds, ensuring meaningful generalization assessment and enabling fair comparison across methods.

*Table D.3.* Hyper-parameter configurations on benchmark datasets.

| Dataset | Batch size | Layers Num | Embed Dim | lr | Epochs | Dropout | Readout | Max dim | Max Ring Size |
|---|---|---|---|---|---|---|---|---|---|
| MUTAG | 32 | 4 | 64 | 0.001 | 300 | 0.0 | mean / sum / sum | 2 | 6 |
| PTC | 32 | 4 | 64 | 0.001 | 300 | 0.0 | mean / sum / sum | 2 | 6 |
| NCI1 | 32 | 4 | 64 | 0.001 | 300 | 0.0 | mean / sum / sum | 2 | 6 |
| NCI109 | 32 | 4 | 64 | 0.001 | 300 | 0.0 | mean / sum / sum | 2 | 6 |
| PROTEINS | 32 | 4 | 64 | 0.001 | 300 | 0.0 | mean / sum / sum | 2 | 6 |
| IMDB-B | 32 | 4 | 64 | 0.001 | 300 | 0.0 | mean / sum / sum | 2 | 6 |
| IMDB-M | 32 | 4 | 64 | 0.001 | 300 | 0.0 | mean / sum / sum | 2 | 6 |
| RDT-B | 32 | 4 | 64 | 0.001 | 300 | 0.0 | mean / sum / sum | 2 | 6 |
| RDT-M | 32 | 4 | 64 | 0.001 | 300 | 0.0 | mean / sum / sum | 2 | 6 |
| ZINK-Small | 128 | 3 | 48 | 0.003 | 1000 | 0.0 | mean / sum / sum | 2 | 18 |
| ZINC-FULL | 128 | 3 | 128 | 0.003 | 1000 | 0.1 | mean / sum / sum | 2 | 18 |
| OGBG-MOLHIV | 128 | 2 | 48 | 0.0001 | 200 | 0.5 | mean / sum / sum | 2 | 6 |
| PEPTIDES-FUNC | 32 | 3 | 128 | 0.003 | 1000 | 0.15 | sum / sum / sum | 2 | 8 |
| PEPTIDES-STRUCT | 32 | 3 | 128 | 0.003 | 1000 | 0.15 | sum / sum / sum | 2 | 8 |

## D.2. Experiments Setup

All experiments are conducted on a Linux system equipped with two NVIDIA RTX A6000 GPUs (48 GB memory each). For all tasks, we adopt the experimental parameters and model configurations in Table D.3. Other hyperparameters set follow the settings established in the following baselines: GIN (Xu et al., 2018)[1], CWN (Bodnar et al., 2021a)[2], PathGNN (Michel et al., 2023)[3], and SMCN (Ebli et al., 2020)[4]. The hidden dimension size and the number of model parameters are listed as follows. All models are trained using the Adam optimizer. The hidden dimension size and the total number of model parameters are reported accordingly. All models are trained using the Adam optimizer with a StepLR learning rate scheduler, which reduces the learning rate by a factor of 0.5 every 50 epochs.

## D.3. Experimental Results

**Open Graph Benchmark** Table D.4 reports the performance of CCIN on the OGB molecular property prediction benchmark, covering both molecular classification and regression tasks. Overall, CCIN achieves best or near-best results on most datasets, demonstrating stable and consistent performance advantages. On the molecular classification tasks MOLBACE and MOLBBBP, CCIN achieves ROC-AUC of 80.47 and 69.81, respectively, outperforming baseline methods such as GIN and CWN, indicating its stronger discriminative ability in predicting biological activity and molecular permeability. For molecular regression and multi-task prediction tasks, CCIN achieves the lowest RMSE on datasets such as MOLESOL, MOLLIPO, MOLTOX21, and MOLTOXCAST, and maintains competitive performance on MOLCLINTOX and MOLSIDER. These results suggest that CCIN is effective at leveraging structural information in molecular graphs and achieves robust performance across both classification and regression tasks.

*Table D.4.* Classification tasks are evaluated using ROC-AUC, while regression and multi-task datasets are evaluated using RMSE.

| | ROC-AUC% (↑) | | RMSE(↓) | | | | | | |
|---|---|---|---|---|---|---|---|---|---|
| | MOLBACE | MOLBBBP | MOLCLINTOX | MOLESOL | MOLFREESOLV | MOLLIPO | MOLSIDER | MOLTOX21 | MOLTOXCAST |
| GIN(Xu et al., 2018) | 76.73±1.53 | 67.08±1.35 | 0.2627±0.016 | 1.5265±0.173 | **2.1612±0.16** | 1.8118±0.252 | 0.4935±0.011 | 0.3364±0.014 | 0.4337±0.013 |
| HGIN (Zhang et al., 2025b) | 76.68±3.36 | 67.69±1.00 | 0.2594±0.002 | 1.7783±0.012 | 3.7672±0.15 | 1.0416±0.008 | 0.4740±0.002 | 0.3348±0.033 | 0.4019±0.002 |
| MSPN(Bodnar et al., 2021b) | 78.64±1.43 | 66.35±2.78 | **0.2458±0.004** | 1.6371±0.046 | 2.3683±0.07 | 1.1458±0.094 | 0.4829±0.035 | 0.3524±0.035 | 0.3905±0.014 |
| CWN (Bodnar et al., 2021a) | 77.05±2.26 | 65.46±1.00 | 0.2557±0.032 | 1.5870±0.157 | 3.8344±0.11 | 1.0602±0.008 | 0.4702±0.072 | 0.3349±0.002 | 0.3879±0.008 |
| CCIN | **80.47±1.89** | **69.81±1.46** | 0.2514±0.078 | **1.4051±0.293** | 3.7678±0.28 | **1.0332±0.083** | 0.4601±0.069 | **0.3186±.0257** | **0.3599±0.012** |

---

[1] https://github.com/weihua916/powerful-gnns

[2] https://github.com/twitter-research/cwn

[3] https://github.com/gasmichel/PathNNs_expressive

[4] https://github.com/yoavgelberg/SMCN

**Four Neighborhoods** Table D.5 further quantitatively validates the impact of different neighborhood designs on model performance. It can be observed that CCIN, using only upper and lower neighborhoods, significantly outperforms the full model on multiple datasets, especially on large-scale molecular graphs and social network graphs. For example, on the NCI1 and NCI109 datasets, CCIN improves performance by approximately 1.13% (80.69 vs. 79.56) and 1.17% (79.71 vs. 78.54) compared to Full, respectively; the improvement is even more significant on the social network datasets RDT-B and RDT-M, reaching 4.28% (93.43 vs. 89.15) and 1.81% (57.30 vs. 55.49), respectively. This result demonstrates that the multi-scale structural information provided by upper and lower neighborhoods is sufficient to effectively characterize the core discriminative features of a graph, while introducing additional finer-grained neighborhood modules may lead to information redundancy or noise accumulation on some datasets. Meanwhile, ablation experiments show that removing $\mathcal{N}\uparrow$ causes the most significant performance degradation, for example, a decrease of 6.61% and 6.76% on NCI1 and NCI109, respectively, validating the crucial role of upper-layer neighborhoods in capturing global structural information. Conversely, removing $\mathcal{B}$ or $\mathcal{N}\downarrow$ actually improves performance on some datasets (e.g., on MUTAG, w/o-$\mathcal{B}$ improves performance by approximately 1.5% compared to Full), indicating that certain neighborhood modules may introduce mismatched information under specific graph structures.These results demonstrate that the upper and lower neighborhoods constitute the core of topological neural network relationship modeling. While other neighborhood modules can supplement information in specific scenarios, they may also introduce unnecessary complexity, thereby affecting the model's generalization performance on different datasets.

*Table D.5.* Predictive performance on graph classification.(Mean±Std)

| | MUTAG | PTC | PROTEINS | NCI1 | NCI109 | IMDB-B | IMDB-M | RDT-B | RDT-M |
|---|---|---|---|---|---|---|---|---|---|
| w/o-$\mathcal{B}$ | 90.62±7.16 | 62.64±8.87 | 75.40±4.94 | 79.46±2.23 | 77.56±2.11 | 74.73±4.73 | 51.93±3.73 | 91.43±2.43 | 54.26±2.35 |
| w/o-$\mathcal{C}$ | 90.55±8.32 | 62.35±8.48 | 74.14±3.84 | 79.44±1.87 | 77.91±1.96 | 74.32±4.51 | 52.29±4.66 | 91.95±2.35 | 54.89±1.57 |
| w/o-$\mathcal{N}\uparrow$ | 87.77±9.22 | 62.05±6.49 | 75.41±4.15 | 72.95±3.17 | 71.78±1.35 | 73.93±2.74 | 51.98±3.62 | 85.50±3.17 | 54.98±3.73 |
| w/o-$\mathcal{N}\downarrow$ | 89.75±7.1 | 61.17±8.42 | 72.88±4.21 | 78.37±1.66 | 77.45±1.71 | 74.41±4.93 | 52.79±3.61 | 91.65±1.13 | 55.46±2.41 |
| CCIN-Full | 89.14±7.72 | 61.47±8.37 | 74.95±5.43 | 79.56±1.63 | 78.54±1.42 | 75.70±4.29 | 52.00±3.47 | 89.15±3.61 | 55.49±1.54 |
| CCIN | 96.30±5.24 | 62.65±10.12 | 76.14±2.53 | 80.69±1.22 | 79.71±2.05 | 74.34±4.58 | 52.71±3.17 | 93.43±1.15 | 57.30±1.78 |

**Ring Size** The maximum ring size controls the order of the same-dimensional combinatorial expansion induced by the bridging cell. Figure D.1 shows when the ring size increases, the edges number of face are involved with more information.

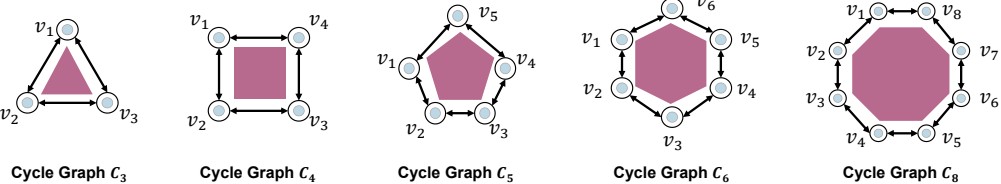

Cycle Graph $C_3$     Cycle Graph $C_4$     Cycle Graph $C_5$     Cycle Graph $C_6$     Cycle Graph $C_8$

*Figure D.1.* Regular simplicial complexes are non-isomorphic (Simplicial Weisfeiler-Lehman cannot distinguish them).

*Table D.6.* Comparison of maximum ring performance on CCIN model. Maximum ring size is set 4 to 14 ($\geq$ 3).

| Ring Size | MUTAG | PTC | PROTEINS | NCI1 | NCI109 |
|---|---|---|---|---|---|
| 4 | 96.30±5.24 | 62.75±6.04 | 76.58±6.02 | 79.40±1.85 | 79.94±3.88 |
| 5 | 94.44±7.86 | 60.78±5.00 | 74.17±0.85 | 80.45±1.44 | 80.26±0.64 |
| 6 | 92.59±6.93 | 61.76±2.40 | 74.17±2.58 | 81.02±1.24 | 80.42±2.12 |
| 7 | 94.44±7.86 | 62.75±5.00 | 74.17±4.49 | 82.24±0.69 | 80.26±2.00 |
| 8 | 96.30±5.24 | 61.76±2.40 | 75.38±1.85 | 81.75±0.72 | 80.34±1.21 |
| 9 | 96.29±5.23 | 61.76±4.81 | 73.87±1.47 | 82.40±0.57 | 81.63±0.45 |
| 10 | 96.32±5.23 | 61.82±2.78 | 74.77±4.41 | 82.18±2.32 | 81.20±2.10 |
| 11 | 96.29±5.23 | 59.80±11.34 | 74.77±5.56 | 82.56±1.69 | 80.09±1.29 |
| 12 | 94.45±7.86 | 59.80±6.04 | 73.87±6.74 | 81.26±0.52 | 81.23±0.81 |
| 13 | 96.29±5.23 | 60.78±3.67 | 74.17±5.57 | 82.72±0.79 | 80.51±0.46 |
| 14 | 96.29±2.61 | 61.76±2.41 | 75.07±4.73 | 82.48±2.15 | 79.93±1.19 |

To evaluate the impact of the maximum ring size hyperparameter, we progressively increased the ring size from 4 to 14 and reported classification performance on multiple biomolecular graph datasets. Table D.6 shows that moderately increasing the ring size generally helps improve the model's ability to represent local closed structures (rings/loops). For example, on

NCI1, with the ring size increasing from 4 to 7 improved the accuracy from 79.40 to 82.24 (an improvement of 3.58%), and then gradually improved the accuracy at 9, 11, and 13, indicating that a longer range of closed structure information can supplement the local structural cues of the molecular graph; NCI109 also achieved the highest accuracy of 81.63 at Ring size=9, an improvement of 2.11% compared to 79.94 at Ring size=4. In contrast, the optimal value for the small molecule dataset MUTAG appears at smaller or medium ring lengths (e.g., approximately 96.29 for Ring size = 4, 8, 10, and 14), but drops to 92.59 at Ring size = 6. This indicates that when the dataset size is small or the statistical instability of higher-order closed structures is high, excessively increasing the ring length may introduce redundant and noisy structures. We also calculated the failure rate at different Ring sizes on the Strongly Regular Graphs benchmark. Table D.7 shows that when the Ring Num is small, some baseline models still exhibit non-zero failure rates on certain SRG instances; however, as the Ring Num increases, the failure rate of CCIN approaches 0 on most SRG instances, demonstrating that more comprehensive closure combination information helps enhance structure discriminative ability. The ring length controls the "closed information coverage" of the same-dimensional combination expansion. Appropriately increasing the Ring size characterizes the local ring structure and improve the distinguishing ability, but an excessively large ring length may lead to statistical instability and redundant closed patterns on some datasets.

*Table D.7.* Failure rates (%) on Strongly Regular Graphs (Rings: 4,5,6,7,8). (Mean±Std).

| | Model | sr251256 | sr16622 | sr261034 | sr281264 | sr291467 | sr351668 | sr351899 | sr361446 | sr401224 |
|---|---|---|---|---|---|---|---|---|---|---|
| | GIN | 0.00±0.00 | 0.00±0.00 | 7.83e-02±0.0 | 0.00±0.00 | 7.15e-03±0.0 | 7.03e-06±3.52e-07 | 3.64e-04±0.0 | 2.61e-02±0.0 | 5.71e-02±0.0 |
| | HGIN | 0.00±0.00 | 0.00±0.00 | 4.44e-02±0.0 | 0.00±0.00 | 0.00±0.00 | 3.02e-06±2.1e-07 | 7.79e-06±3.1e-06 | 1.82e-02±6.1e-06 | 1.82e-02±0.0 |
| 4 | MPSN | 0.00±0.00 | 0.00±0.00 | 6.67e-02±0.0 | 0.00±0.00 | 6.15e-04±6.1e-05 | 3.02e-04±2.7e-06 | 0.00±0.00 | 8.16e-02±0.0 | 8.16e-02±0.0 |
| | CIN | 0.00±0.00 | 0.00±0.00 | 4.44e-02±0.0 | 0.00±0.00 | 0.00±0.00 | 3.52e-06±2.09e-07 | 7.79e-06±3.1e-06 | 1.92e-02±7.0e-05 | 1.92e-02±0.0 |
| | CCIN | 0.00±0.00 | 0.00±0.00 | 4.44e-02±0.0 | 0.00±0.00 | 4.88e-04±1.2e-04 | 4.88e-06±3.02e-06 | 7.79e-06±3.1e-06 | 1.91e-02±8.1e-05 | 1.85e-02±0.0 |
| | GIN | 0.00±0.00 | 0.00±0.00 | 6.67e-02±0.0 | 0.00±0.00 | 6.10e-04±0.0 | 6.43e-03±0.0 | 4.52e-06±0.0 | 0.00±0.00 | 0.0 |
| | HGIN | 0.00±0.00 | 0.00±0.00 | 4.44e-02±0.0 | 0.00±0.00 | 0.00±0.00 | 0.00±0.00 | 0.00±0.00 | 1.85e-02±7.0e-05 | 1.85e-02±0.0 |
| 5 | MPSN | 0.00±0.00 | 0.00±0.00 | 6.67e-02±0.0 | 0.00±0.00 | 6.1e-04±6.1e-05 | 3.02e-04±2.7e-06 | 0.00±0.00 | 8.16e-02±0.0 | 8.16e-02±0.0 |
| | CIN | 0.00±0.00 | 0.00±0.00 | 4.44e-02±0.0 | 0.00±0.00 | 0.00±0.00 | 3.5e-06±2.1e-07 | 7.79e-06±3.1e-06 | 1.92e-02±7.0e-05 | 1.85e-02±0.0 |
| | CCIN | 0.00±0.00 | 0.00±0.00 | 0.00±0.00 | 0.00±0.00 | 0.00±0.00 | 0.00±0.00 | 0.00±0.00 | 0.00±0.00 | 5.3e-03±0.0 |
| | GIN | 0.00±0.00 | 0.00±0.00 | 0.00±0.00 | 0.00±0.00 | 0.00±0.00 | 0.00±0.00 | 0.00±0.00 | 0.00±0.00 | 0.00±0.00 |
| | HGIN | 0.00±0.00 | 0.00±0.00 | 6.67e-02±0.0 | 0.00±0.00 | 6.1e-04±0.0 | 0.00±0.00 | 0.00±0.00 | 8.16e-02±0.0 | 8.16e-02±0.0 |
| 6 | MPSN | 0.00±0.00 | 0.00±0.00 | 0.00±0.00 | 0.00±0.00 | 0.00±0.00 | 0.00±0.00 | 0.00±0.00 | 0.00±0.00 | 0.00±0.00 |
| | CIN | 0.00±0.00 | 0.00±0.00 | 6.67e-02±0.0 | 0.00±0.00 | 6.1e-04±6.1e-05 | 3.02e-04±2.7e-06 | 0.00±0.00 | 8.16e-02±0.0 | 8.16e-02±0.0 |
| | CCIN | 0.00±0.00 | 0.00±0.00 | 0.00±0.00 | 0.00±0.00 | 0.00±0.00 | 0.00±0.00 | 0.00±0.00 | 0.00±0.00 | 0.00±0.00 |
| | GIN | 0.00±0.00 | 0.00±0.00 | 0.00±0.00 | 0.00±0.00 | 6.1e-04±0.0 | 0.00±0.00 | 0.00±0.00 | 0.00±0.00 | 0.00±0.00 |
| | HGIN | 0.00±0.00 | 0.00±0.00 | 0.00±0.00 | 0.00±0.00 | 0.00±0.00 | 0.00±0.00 | 0.00±0.00 | 0.00±0.00 | 0.00±0.00 |
| 7 | MPSN | 0.00±0.00 | 0.00±0.00 | 6.67e-02±0.0 | 0.00±0.00 | 6.1e-04±6.1e-05 | 3.1e-04±2.7e-06 | 0.00±0.00 | 8.16e-02±0.0 | 8.16e-02±0.0 |
| | CIN | 0.00±0.00 | 0.00±0.00 | 0.00±0.00 | 0.00±0.00 | 0.00±0.00 | 0.00±0.00 | 0.00±0.00 | 0.00±0.00 | 0.00±0.00 |
| | CCIN | 0.00±0.00 | 0.00±0.00 | 0.00±0.00 | 0.00±0.00 | 0.00±0.00 | 0.00±0.00 | 0.00±0.00 | 0.00±0.00 | 0.00±0.00 |
| | GIN | 0.00±0.00 | 0.00±0.00 | 0.00±0.00 | 0.00±0.00 | 2.3e-06±0.0 | 0.00±0.00 | 0.00±0.00 | 0.00±0.00 | 0.00±0.00 |
| | HGIN | 0.00±0.00 | 0.00±0.00 | 0.00±0.00 | 0.00±0.00 | 0.00±0.00 | 0.00±0.00 | 0.00±0.00 | 0.00±0.00 | 0.00±0.00 |
| 8 | MPSN | 0.00±0.00 | 0.00±0.00 | 1.67e-06±0.0 | 0.00±0.00 | 0.00±0.00 | 0.00±0.00 | 0.00±0.00 | 5.71e-03±0.0 | 0.00±0.00 |
| | CIN | 0.00±0.00 | 0.00±0.00 | 0.00±0.00 | 0.00±0.00 | 0.00±0.00 | 0.00±0.00 | 0.00±0.00 | 0.00±0.00 | 0.00±0.00 |
| | CCIN | 0.00±0.00 | 0.00±0.00 | 0.00±0.00 | 0.00±0.00 | 0.00±0.00 | 0.00±0.00 | 0.00±0.00 | 0.00±0.00 | 0.00±0.00 |

**Max Dim** Table D.8 shows the performance variation of CCIN under different maximum dimensionality (max dim) input features, indicating that high-order structural information has a notable impact on on model performance.

*Table D.8.* Performance of CCIN model with different maximum dimension features

| max dim | MUTAG | PTC | PROTEINS | NCI1 | NCI109 | IMDB-B | IMDB-M | RDT-B |
|---|---|---|---|---|---|---|---|---|
| 0 | 75.93±6.93 | 59.80±6.04 | 74.77±4.59 | 70.40±3.36 | 70.15±2.77 | 75.67±2.62 | 52.44±1.91 | 75.50±1.78 |
| 1 | 96.30±5.24 | 60.78±2.77 | 74.47±5.01 | 80.45±1.59 | 80.91±1.78 | 76.67±3.09 | 54.22±4.91 | 92.33±0.85 |
| 2 | 94.44±7.86 | 64.71±2.40 | 75.12±3.06 | 80.70±1.20 | 81.02±2.35 | 78.00±5.72 | 53.33±7.1 | 89.83±8.27 |

When the maximum dimension is 0 (containing only 0-cell node information), the model's performance is limited on multiple datasets. For instance, the classification accuracy on MUTAG, NCI1, and RDT-B is 75.93, 70.40, and 75.50, respectively. After introducing 1-dimensional structural information by incorporating 1-cell (edge-level) features, the model performance improves substantially. Specifically, the accuracy on MUTAG increases from 75.93 to 96.30, corresponding to a relative improvement of approximately 26.8%; on NCI1, it increases from 70.40 to 80.45 (a 14.3% relative improvement); and on RDT-B, it rises from 75.50 to 92.33, achieving a 22.3% relative improvement. Building upon this, further incorporating

2-dimensional structural information by adding 2-cell features leads to continued performance gains on some datasets. For example, on PTC, the accuracy improves from 60.78 to 64.71 yielding a relative improvement of 6.5%. On NCI1 and NCI109, smaller relative improvements of 0.3% and 0.1% are observed, suggesting that higher-order topological structures can further complement complex relational patterns. However, on datasets such as MUTAG and RDT-B, the performance exhibits slight fluctuations after introducing 2-dimensional information, implying that in scenarios with fewer higher-order structures, excessively high-dimensional representations may introduce redundant information.

**Layer Number**   To analyze the impact of network depth on model performance, we increased the number of layers in CCIN from 1 to 8 and evaluated it on the molecular graph dataset and the Strongly Regular Graphs benchmark. Table D.9 shows that in molecular graph classification tasks, the model is generally stable with varying layer counts, and optimal performance typically occurs in shallower structures (1-3 layers). For example, on MUTAG, a 1-layer model achieves a peak accuracy of 98.15%, while performance significantly decreases with 7 and 8 layers, indicating that excessively deep networks may cause oversmoothing problem on small datasets. For large-scale datasets such as NCI1 and NCI109, the performance difference between different layer counts is smaller, with 2-3 layers achieving comparable results to deeper networks. On the SR family benchmarks (Table D.10), network depth has a more significant impact on structural discriminative ability. As the number of layers increases from 1 to 3-4, the failure rate of the model on most SRG instances drops rapidly and approaches 0, indicating that moderately increasing the depth helps to aggregate higher-order structural information, when the number of layers increases further, the performance improvement tends to stable.

*Table D.9.* Classification accuracy of CCIN model with different layers. Layer number is set 1 to 8.

| layer num | MUTAG | PTC | PROTEINS | NCI1 | NCI109 |
|---|---|---|---|---|---|
| 1 | 98.15±2.62 | 62.75±5.00 | 75.98±3.32 | 81.43±1.65 | 79.85±2.09 |
| 2 | 94.44±7.86 | 63.73±5.00 | 76.28±4.90 | 81.83±1.79 | 79.61±0.52 |
| 3 | 96.30±2.62 | 61.76±4.80 | 74.47±3.32 | 81.02±0.79 | 80.58±1.05 |
| 4 | 92.59±6.93 | 61.76±2.40 | 74.17±2.58 | 81.02±1.24 | 80.42±2.12 |
| 5 | 94.44±4.54 | 63.73±3.67 | 75.98±2.36 | 81.02±0.40 | 80.74±1.60 |
| 6 | 94.44±4.54 | 59.80±6.04 | 73.87±3.68 | 80.94±0.64 | 80.26±1.69 |
| 7 | 83.33±9.07 | 60.78±3.67 | 75.98±2.12 | 81.83±0.61 | 81.31±1.24 |
| 8 | 72.22±9.07 | 63.72±1.39 | 73.27±2.97 | 81.51±1.82 | 80.17±1.49 |

*Table D.10.* Failure rates (%) on Strongly Regular Graphs (Rings: 4,5,6,7). (Mean±Std).

| Model | sr251256 | sr16622 | sr261034 | sr281264 | sr291467 | sr351668 | sr351899 | sr361446 | sr401224 |
|---|---|---|---|---|---|---|---|---|---|
| 1 | 9.52±0.00 | 0.00±0.00 | 6.67±0.0 | 0.00±0.00 | 67.81±2.07 | 3.11±0.13 | 2.51±0.17 | 8.09±0.02 | 8.20±0.0 |
| 2 | 5.71±0.00 | 0.00±0.00 | 6.22±0.17 | 0.00±0.00 | 4.12±0.21 | 0.08±2.79e-05 | 0.12±9.28e-05 | 5.62±0.14 | 7.83±0.12 |
| 3 | 0.00±0.00 | 0.00±0.00 | 4.44±0.00 | 0.00±0.00 | 4.87e-4±1.19e-04 | 4.90e-06±2.21e-06 | 7.79e-06±3.12e-06 | 1.94±8.08e-05 | 1.85±0.00 |
| 4 | 0.00±0.00 | 0.00±0.00 | 4.44±0.00 | 0.00±0.00 | 0.00±0.00 | 0.00±0.00 | 0.00±0.00 | 1.12±0.09 | 1.74±0.01 |
| 5 | 0.00±0.00 | 0.00±0.00 | 3.56±0.00 | 0.00±0.00 | 0.00±0.00 | 0.00±0.00 | 0.00±0.00 | 0.98±0.09 | 1.27±0.05 |
| 6 | 0.00±0.00 | 0.00±0.00 | 2.67±0.17 | 0.00±0.00 | 0.00±0.00 | 0.00±0.00 | 0.00±0.00 | 0.92±7.39e-05 | 1.01±0.02 |
| 7 | 0.00±0.00 | 0.00±0.00 | 2.22±0.00 | 0.00±0.00 | 0.00±0.00 | 0.00±0.00 | 0.00±0.00 | 0.81±1.15e-04 | 0.89±0.02 |
| 8 | 0.00±0.00 | 0.00±0.00 | 2.22±0.00 | 0.00±0.00 | 0.00±0.00 | 0.00±0.00 | 0.00±0.00 | 0.70±4.87e-05 | 0.79±0.00 |

**Injective Function Design.**   To analyze the impact of different neighborhood aggregation functions on model performance, we conducted ablation experiments on commonly used aggregation functions, including sum, mean, and max, in Init Readout, Intermediate Readout, and Final Readout. Table D.11 shows that, while keeping the intermediate and final readout functions both as sum, a comparison of the initial readout functions reveals that different aggregation methods have similar overall performance across various datasets. For example, on the MUTAG dataset, mean and sum have similar performance (96.30 vs. 96.29), while max is slightly lower. On PTC and RDT-B, mean aggregation achieves better results of 65.69 and 78.67, respectively, indicating that mean aggregation is more effective in integrating local structural information.

*Table D.11.* Ablation study on init readout function. Readout: sum, Final readout: sum.

| | | MUTAG | PTC | PROTEINS | NCI1 | NCI109 | IMDB-B | IMDB-M | RDT-B |
|---|---|---|---|---|---|---|---|---|---|
| | sum | 96.29±5.26 | 61.76±6.35 | 74.17±2.78 | 81.51±0.91 | 79.45±2.48 | 76.67±2.05 | 53.33±6.79 | 76.50±3.89 |
| Init Method | max | 94.44±2.47 | 63.73±6.04 | 74.17±2.97 | 81.35±1.52 | 79.37±2.72 | 76.00±1.41 | 53.33±3.31 | 77.19±3.94 |
| | mean | 96.30±5.24 | 65.69±5.00 | 75.08±1.12 | 80.86±1.32 | 80.74±2.52 | 76.00±2.94 | 53.56±5.14 | 78.67±2.66 |

Table D.12 shows that the three aggregation functions show little difference in performance on most datasets, but exhibit different advantages on individual datasets. For example, on NCI1 and NCI109, sum and mean aggregations achieve similar performance, while on PTC, max aggregation achieves a relatively high accuracy (64.71).

*Table D.12.* Ablation study on intermediate layer readout function. Initial Method: mean, Final readout: sum.

|  |  | MUTAG | PTC | PROTEINS | NCI1 | NCI109 | IMDB-B | IMDB-M | RDT-B |
|---|---|---|---|---|---|---|---|---|---|
| | mean | 96.30±2.62 | 61.76±7.21 | 73.57±1.85 | 81.18±0.64 | 79.69±2.45 | 76.33±3.09 | 53.56±3.62 | 79.50±5.31 |
| Intermediate Readout | sum | 94.44±7.86 | 61.76±2.40 | 73.87±5.15 | 81.51±1.43 | 79.94±1.41 | 77.33±2.62 | 53.33±7.12 | 71.83±8.27 |
| | max | 94.44±7.86 | 64.71±2.40 | 74.47±3.06 | 80.70±1.20 | 80.02±2.35 | 76.00±5.72 | 52.44±5.37 | 78.67±2.09 |

Table D.13 observes that the final readout function has a more direct impact on model performance. For example, on the MUTAG dataset, using mean as the final readout function can improve the accuracy to 98.15, which is about 1.9% and 3.9% higher than sum and max, respectively. These results indicate that the choice of specific aggregation function is relatively stable for the initial and intermediate layers, while the final readout layer has a more significant impact on performance.

*Table D.13.* Ablation study on final readout function. Initial Method: mean, Readout: sum.

|  |  | MUTAG | PTC | PROTEINS | NCI1 | NCI109 | IMDB-B | IMDB-M | RDT-B |
|---|---|---|---|---|---|---|---|---|---|
| | mean | 98.15±2.63 | 62.75±4.99 | 74.17±2.25 | 81.83±0.11 | 81.07±1.55 | 77.32±3.74 | 53.11±4.23 | 79.83±1.84 |
| Final Readout | sum | 96.30±2.62 | 60.78±3.67 | 75.08±3.32 | 81.43±1.44 | 80.83±0.52 | 78.33±4.03 | 54.28±5.36 | 76.45±2.64 |
| | max | 94.44±7.86 | 61.34±5.27 | 75.68±2.65 | 80.32±4.39 | 80.66±2.10 | 75.67±0.47 | 54.00±6.28 | 77.27±1.69 |

# E. Time and Computational Complexity

We analyze the computational and memory complexity of the proposed combinatorial complex Weisfeiler-Lehman test and its neural instantiation CCIN. In a combinatorial complex, the degree of a cell is defined as follows: for any $k$-cell with $\sigma$, the boundary size $|\mathcal{B}(\sigma)|$ is the number of its neighboring $(k-1)$-cells, and the co-boundary size $|\mathcal{C}(\sigma)|$ is the number of its $(k+1)$-cells. The neighborhood degree $|\mathcal{N}_\downarrow(\sigma)|$ is the number of other k-cells sharing a given $(k-1)$-cell, and the set of k-cells of the coboundary corresponding to each $(k-1)$-cell $\tau$ is defined as $S_k(\tau)$. The neighborhood degree $|\mathcal{N}_\uparrow(\sigma)|$ is the number of other k-cells sharing a given $(k+1)$-cell, and the set of k-cells of the boundary to each $k+1$-cell $\tau$ is defined as $S_k(\eta)$. The computational complexity of the four neighborhood functions is shown in Table E.1.

*Table E.1.* Time complexity of four neighborhood function on combinatorial complexes

| Neighborhood Function | Boundary (✗) | Co-Boundary (✗) | Lower Adjacency (✓) | Upper Adjacency (✓) |
|---|---|---|---|---|
| Time complexity | $O(\sum_{\sigma \in \mathcal{X}_k} |\mathcal{B}(\sigma)|)$ | $O(\sum_{\sigma \in \mathcal{X}_k} |\mathcal{C}(\sigma)|)$ | $O(\sum_{\tau \in \mathcal{X}_{k-1}} |S_k(\tau)|^2)$ | $O(\sum_{\eta \in \mathcal{X}_{k+1}} |S_{k+1}(\eta)|^2)$ |

As before, Lemma 4.7 and Theorem 4.8 have simplified the update rules for combinatorial complexes, namely, retaining only the upper and lower neighborhoods , reducing the multisets of boundary and coboundary that need to be constructed and participate in the hash in each round without losing discriminative power. Specifically, The algorithmic complexity of CCIN is $O(\sum_{\tau \in \mathcal{X}_{k-1}} |S_k(\tau)|^2 + \sum_{\eta \in \mathcal{X}_{k+1}} |S_{k+1}(\eta)|^2)$, which is comparable to the expressive power of full higher-order message passing framework on combinatorial complexes. We compared the single-epoch training time (seconds/epoch) of CCIN with its full version CCIN-Full (with four neighborhood) in Table E.2. Specifically, training time is reduced by an average of approximately 6.6%, with the most speedup of 18.1% on NCI109 dataset, and a 24.8% speedup on ZINK-Small dataset. This indicates that by theoretically simplifying and removing redundant boundary, co-boundary calculations, runtime efficiency is improved while maintaining expressive power.

*Table E.2.* Train time (seconds per epoch)

|  | PROTEINS | NCI1 | NCI109 | ZINK-Small | ZINC-FULL | MOLHIV | PEPTIDES-FUNC | PEPTIDES-STRUCT | Avg. |
|---|---|---|---|---|---|---|---|---|---|
| CCIN-Full | 5.16 | 15.20 | 13.28 | 21.85 | 290.61 | 84.22 | 52.81 | 75.36 | 69.56 |
| CCIN | 4.24 (-17.83%) | 13.97 (-8.09%) | 10.87 (-18.15%) | 16.42 (-24.85%) | 280.85 (-3.36%) | 77.19 (-8.35%) | 46.90 (-11.19%) | 69.48 (-7.80%) | 64.99 (-6.57%) |

