# OpenReview forum: "Weisfeiler Lehman Test on Combinatorial Complexes: Generalized Expressive Power of Topological Neural Networks"
_ICML.cc/2026/Conference — Submitted to ICML 2026_

### Official Review · Reviewer_heeo · 2026-03-06

**Soundness:** 3
**Presentation:** 3
**Significance:** 3
**Originality:** 3
**Overall Recommendation:** 5
**Confidence:** 2

**Summary:**

This paper addresses the fragmentation of existing topological neural networks and WL variants in topological deep learning by introducing a unified theoretical framework. The main contribution is the Combinatorial Complex Weisferler-Lehman test that formalizes topological message passing through four neighborhood relations. The authors prove that upper and lower adjacencies are sufficient to achieve the expressive power of the full framework. Extensive experiments are conducted on synthetic and real-world benchmarks, demonstrating that the proposed framework consistently outperform traditional GNNs.

**Compliance With Llm Reviewing Policy:**

Affirmed.

**Final Justification:**

I have carefully reviewed the authors’ rebutttals, and my concerns have been adequately addressed.

**Key Questions For Authors:**

1. For very high-dimensional combinatorial complexes, how does the quadratic computational complexity of CCIN’s neighborhood aggregation affect its scalability? Are there any optimization strategies to mitigate this issue?
2. For real-world applications like molecular property prediction, interpretability is crucial. Does CCIN provide any means to interpret which topological features contribute most to the predictions?

**Limitations:**

Limitations are not discussed in this paper.

**Strengths And Weaknesses:**

Strengths:
1. The paper establishes a rigorous unified framework for topological message passing, filling the gap of fragmented WL variants for different topological structures.
2. The proposed framework generalize seamlessly across graphs, hypergraphs, simplicial complexes, and cellular complexes, addressing the structural specificity of existing models. This versatility is validated by consistent performance across heterogeneous datasets, demonstrating strong cross-domain adaptability.

Weaknesses:
1. A core practical challenge, which the paper acknowledges, is the lack of native combinatorial complex benchmarks. The empirical evaluation relies heavily on lifting graph datasets into cell complexes. The performance of CCIN is therefore intrinsically linked to the quality and appropriateness of these lifting functions. It remains unclear how sensitive the results are to the choice of lifting and whether the benefits observed are due to the CCIN architecture or simply the specific lifted features.

---

> ### Author Rebuttal · Authors · 2026-03-30
>
> **W1:** Thanks the reviewer for highlighting this critical issue regarding dataset construction.  We agree that in the absence of native combinatorial complex benchmarks, the choice of lifting functions may influences performance.
>
> To address your concern about whether our gains stem from the architecture or the lifting method, we decouple the architecture's effect from the lifting method [1], conduct additional experiments comparing CCIN against MSPN [2], CWN [3], and TopoTune [4] under two distinct lifting strategies: cycle lifting and clique lifting. All methods used identical lifted inputs and settings. The results are as follows:
>
> |lifting|MSPN|CWN|TopoTune|CCIN|
> |:-|:-|:-|:-|:-|
> |cycle lifting|88.3±10.7|90.0±7.4|86.4±6.5|96.4±2.1|
> |clique lifting|84.21±4.3|92.5±4.6|89.4±4.4|93.7±3.0|
>
> The table shows that the choice of lifting method affects model performance. Cycle lifting performs better than clique in scenarios with  loop structures in the molecular graph. CCIN maintains its superiority over baseline methods under both lifting methods. These results indicate that CCIN is not derived from the feature construction of a specific lifting method, but is related to the message passing mechanism design of CCIN.
>
> In the revised version, we will supplement these ablation experiments on different lifting strategies to further clarify the impact of different lifting functions on CCIN evaluation.
>
> [1] TopoX. JMLR2024
> [2] MSPN. ICML2021
> [3] CWN. NIPS2021
> [4] TopoTune. ICML2025
>
>
> **Q1:** Thank you for the reviewer's question. We acknowledge that the neighborhood aggregation in CCIN involves quadratic complexity, which poses challenges for extremely high-dimensional complexes with dense neighborhoods.
>
> In practice, CCIN employs two key strategies:
> - Redundancy elimination: By proving that the four types of neighborhood relationships can be reduced to sufficient neighborhoods based only on upper/lower adjacencies, CCIN reduces the  unnecessary neighborhood and redundant computations.
> - Efficient implementation: we pre-compute and sparsely store the adjacency/bridge structure of the complex to reduce the overhead of repeated graph construction during training.
>
> We will discuss this more explicitly in the revised limitations and future work.
>
> **Q2:** Thank you for highlighting this critical aspect. We agree that interpretability is paramount for real-world applications like molecular property prediction.
>
> CCIN provides interpretability through a combination of structural design and explicit topological analysis:
> - Structural Interpretability:  CCIN decomposes the topological relationship into four neighborhoods (boundary, coboundary, lower adjacency, upper adjacency). Each of these relationships corresponds to distinct topological interaction patterns within the molecular graph.
> - Feature contribution analysis：By analyzing the information aggregation patterns across 2D and 3D structures of molecules, we can identify which specific topological features (e.g., specific rings, functional groups and cliques, or their higher-order interactions) contribute to the final prediction.
>
>
> We hope these clarifications address the reviewer’s concerns, and we thank the reviewer again for the helpful feedback.

---

> > ### Author Rebuttal · Reviewer_heeo · 2026-04-03
> >
> > Thank you for your comments. At this stage, I have no further concerns and will keep my positive recommendation.

---

> > > ### Author Response · Authors · 2026-04-04
> > >
> > > Thank you for your feedback and for confirming that your concerns have been fully addressed.
> > >
> > > We are glad that the additional experiments and clarifications regarding lifting strategies, computational considerations, and interpretability have resolved the issues you raised. These points have been incorporated into the revised version to ensure that the role of lifting functions, the efficiency of CCIN, and its interpretability are clearly presented.
> > >
> > > We appreciate your positive recommendation and your support during the discussion phase.

---

### Official Review · Reviewer_inq5 · 2026-03-07

**Soundness:** 2
**Presentation:** 2
**Significance:** 2
**Originality:** 2
**Overall Recommendation:** 3
**Confidence:** 4

**Summary:**

The paper introduces the Combinatorial Complex Weisfeiler-Leman test (CCWL), a WL-style refinement procedure defined on combinatorial complexes. The aim is to provide a common expressivity framework across graphs, hypergraphs, simplicial complexes, and cellular complexes.

The paper formalizes four neighborhood relations on combinatorial complexes and studies how these relations support higher-order message passing. On the theory side, the paper argues that CCWL can simulate existing WL-style tests on several higher-order domains through suitable liftings, and proves that using upper and lower adjacency is sufficient for maximal expressive power.

Based on this, the paper proposes a neural model, CCIN, intended as a combinatorial-complex instantiation of the CCWL framework, and evaluates it on synthetic and real-world benchmarks across multiple topological data types.

**Compliance With Llm Reviewing Policy:**

Affirmed.

**Ethical Review Concerns:**

The paper is built around a promising idea, and I appreciate the authors’ effort to provide a unifying framework. Their rebuttal was strong and addressed many of my concerns. However, it remains unclear to me how far this unification really goes, and whether the proposed approach is in fact more principled or more effective than prior higher-order topological methods. In my view, the paper still needs substantial revision. While the rebuttal gives me confidence that the authors may be able to carry out these improvements, increasing my score at this stage would be unfair to other submissions that do not require such significant reworking. I therefore maintain my current score.

**Final Justification:**

The paper is built around a promising idea, and I appreciate the authors’ effort to provide a unifying framework. Their rebuttal was strong and addressed many of my concerns. However, it remains unclear to me how far this unification really goes, and whether the proposed approach is in fact more principled or more effective than prior higher-order topological methods. In my view, the paper still needs substantial revision. While the rebuttal gives me confidence that the authors may be able to carry out these improvements, increasing my score at this stage would be unfair to other submissions that do not require such significant reworking. I therefore maintain my current score.

**Key Questions For Authors:**

1. The paper’s main motivation is that prior higher-order models use incompatible neighborhood systems and may fail to preserve the full topological neighborhood structure of combinatorial complexes. Can the authors formalize this deficiency more precisely and give either a theorem or a controlled experiment that directly demonstrates the claimed loss relative to the proposed framework?

2. What is the clearest concrete benefit of the unified combinatorial-complex view beyond notation? For example, does it enable a stronger theorem, a design principle for new architectures, a transfer mechanism across domains, or an efficiency benefit?

3. How does the proposed framework differ in expressivity and architectural consequences from prior combinatorial-complex models such as TopoTune, beyond using a different neighborhood decomposition? A more explicit side-by-side comparison would help.

4. The paper claims simulation of graph, hypergraph, simplicial, and cellular WL variants via lifting into combinatorial complexes. Which parts of these results are fundamentally new, and which parts should be viewed as reformulations of known correspondences?

**Limitations:**

The paper does not discuss limitations as clearly as it should. Richer higher-order neighborhoods may increase computational and memory costs relative to simpler models.

**Strengths And Weaknesses:**

This paper explores a worthwhile direction: Can the expressive-power analysis of higher-order topological learning models can be placed under a single combinatorial-complex framework. This is answered affirimatively, and the papers connects theory and architecture design through the CCWL-to-CCIN bridge.

**Soundness.**  On the positive side, the paper has a technically coherent story: it defines CCWL formally, states the main sufficiency result that upper and lower adjacency match the full refinement, and provides a neural instantiation together with experiments. The empirical section is broad, including synthetic and real-world benchmarks, and the appendix contains additional ablations and complexity discussion.  On the negative side, however, I am not fully convinced that the strongest central claims are supported as strongly as they need to be. In particular, the paper repeatedly claims that prior methods fail to faithfully preserve the topological neighborhood structure of combinatorial complexes and that simplifications such as TopoTune’s reduced neighborhood system disrupt higher-order information propagation, but these claims are not established with a direct theorem or controlled empirical comparison (as far as I can tell). The main theorem shows an internal result within the proposed framework, namely that two neighborhood types suffice for full CCWL expressivity, but this does not by itself justify the stronger comparative claims about existing methods. Some proof arguments, especially around simulation and strict expressivity under liftings, also rely heavily on constructions deferred to the appendix and are presented somewhat informally in the main text. As a result, I found the technical core not as fully nailed down as the paper’s positioning suggests. This may be hidden in the long appendix but then please bring it out.

**Presentation.**  It is readable at a high level and the overall structure is conventional: motivation, preliminaries, theory, architecture, experiments. Figure 1 and the definitions of the four neighborhood types  are helpful. However, when describing what exactly is unified, what exactly is simulated, and how combinatorial complexes relate mathematically to hypergraphs versus simplicial or cellular complexes, is not made clear enough in the paper. Several sentences are grammatically rough or imprecise, and some terminology is used loosely, for example when discussing “faithful preservation,” “incompatible neighborhood definitions,” or “disrupts higher-order information propagation,” without giving a precise formal criterion. See also earlier remark. Related-work positioning is also weaker than it should be for a paper whose novelty depends heavily on comparison to prior higher-order WL and combinatorial-complex models. The paper cites many relevant works, but the exact novelty margin over them remains unclear in the main text.

**Significance.** It is not convincingly explained what concrete advantage this unified view provides beyond serving as a broad umbrella. It is not fully clear whether the main benefit is conceptual unification, stronger expressivity guarantees, better cross-domain transfer, simpler model design, or empirical gains. I see the significance as moderate rather than high in its current form.

**Originality.** There is some originality in combining a WL-style framework, combinatorial-complex neighborhoods, and a neural instantiation into one framework. The theorem claiming that upper and lower adjacency suffice for full CCWL is also a potentially interesting contribution. However, the paper’s novelty over closely related prior work is not convincingly separated. Much of the contribution feels like a broad reframing and synthesis of known higher-order message-passing and WL ideas in a combinatorial-complex setting, with incremental rather than leaps from prior hypergraph, simplicial, cellular, and combinatorial-complex work. The paper may still be useful as a unifying perspective, but the originality would be easier to credit if the manuscript gave a sharper and more direct account of what is genuinely new and what is inherited or reformulated from prior literature.

Overall, I see the paper as having clear strengths in ambition, relevance, and breadth, but with notable weaknesses in motivation, novelty positioning, and the support for some of its strongest comparative claims.

---

> ### Author Rebuttal · Authors · 2026-03-30
>
> **Soundness:** Thanks for the constructive comments. We clarify as follows:
> -  "faithfully preserve" or "disrupt higher-order information propagation" were imprecise. Our concern stems from prior works such as TopoTune considers boundary and upper relations that likes CWL on cellular complexes [1], analyzes expressiveness via standard WL/k-WL on augmented Hasse graphs, which maps higher-order relations into Hasse graphs.
> - "comparative claims": CCIN operates on the four neighborhoods (boundary $B$, coboundary $C$, lower $N_\downarrow$ /upper $N_\uparrow$ adjacency) and establishes $N_\downarrow,N_\uparrow$ suffice for CCIN (full). In the revision, we will include a more detailed discussion that TopoTune operates on the Hasse graph by mapping higher-order neighborhood into graph compared to the intrinsic neighborhoods operators of CCs.
> - "rely heavily on...appendix": We will add a short proof roadmap, a notations table, and move key conditions into the main text to  improve readability.
>
> **Presentation/Significance:** Thank you for insightful comments. We respond these points as follows:
> - unified: CCs unify graphs, hypergraphs, simplicial and cellular complexes, then CCWL provides generalized refinement framework to analyze expressive power across all these structures.
> - simulated: It means that WL-style refinement on specific lifting structures, e.g., simplicial or hypergaph WL can be simulated within the CCWL via four neighborhoods.
> - "how CCs relate mathematically...is not made clear enough": In Appendix A.2 and A.3, CCs generalize hypergraphs, cellular complexes by utilizing the four neighborhoods to define incidence relations. In the revision, we will move the related analysis into the main text.
> - "not fully clear...main benefit": CCWL proves that $B,C$ are redundant and that $N_\downarrow,N_\uparrow$ form the sufficient neighborhood for full CCWL expressiveness to simplifys CCIN framework.
>
> **Originality/Presentation/Q3:** Thanks for this important comment. Compared to TopoTune, it analyzes expressiveness via strictly augmented Hasse graphs and uses WL/k-WL on the Hasse graphs. In contrast, CCWL formulates expressivity on CCs using intrinsic neighborhood operators, prove a redundancy that upper/lower adjacency are sufficent for full CCWL expressiveness, and derive the simplified CCIN from Theorem 4.8.
>
> While both works operate on CCs, the distinctions are as follows:
> ||TopoTune|CCIN(ours)|
> |:-|:-|:-|
> |Analytical view|Strictly augmented Hasse graph|CCs' neighborhood function|
> |Neighborhood|$B,N_\uparrow$ (fig.2-4 in [2])|$B,C,N_\downarrow,N_\uparrow$|
> |Expressivity|WL/k-WL on induced Hasse graph|CCWL on CCs' neighborhoods|
> |Main result|Unified graph-based WL on Hasse Graph（Prop B.11, B.12,[2]）|Redundancy and sufficient neighborhood (Theorem 4.8)|
> |Architecture|General Hasse-graph GCNN|Simplified CCIN|
>
> **Q1:** Thank you for this question. We clarify the claimed loss from both views:
> -  theorem: most prior methods (Table 1) are defined only on specific domains. Theorem 4.8 formally proves that only the upper and lower adjacency operators are sufficient to the full expressivity. This implies $B,C$ are redundant within CCWL.
> - controlled experiment: ablation studies in Table 6 and Table D.5 show that removing $N_\downarrow$ and $N_\uparrow$ from CCIN leads to clear deterioration, but removing $B$ or $C$ may not cause degradation. These results provide empirical evidence that the neighborhood operators identified by CCIN are the most critical in practice.
>
> **Q2:** Thanks for pointing this out. The unified CCWL view provides two concrete benefits: (1) it enables a principled analysis of which neighborhood operators are necessary or redundant, leading to a sufficient neighborhood; (2) it simplifys CCIN and reduce the complexity, CCIN can use only upper/lower adjacency aggregation. Compared to CCIN (full), CCIN removes two neighborhoods, reducing complexity (Table E.1). Table E.2 shows a training speed improvement of 6.6%–24.8%.
>
> **Q4:** We clarify reformulation and novelty as
> - reformulations: (1) the lifting operation with topological relationships via cycle lifting， (2) the formulation of CCs in terms of neighborhood operators，(3) the WL test for higher-order message passing.
> - fundamentally new: (1) we define a native CCWL refinement on the four intrinsic neighborhood operators of CCs, (2) we prove the redundancy result in Theorem 4.8, (3) we establish the generalization of expressive power on lifted complexes (Props. B.4–B.8), showing that CCWL can distinguish cases that 1-WL, HWL, SWL, and CWL fail to distinguish.
>
> **L1:** We also supplement the training time (s/epoch) compared to simpler models, e.g., GIN, CCIN may incur additional computational cost.
> ||MUTAG|ZINC-small|ZINC-FULL|
> |:-|:-|:-|:-|
> |GIN|3.09±0.03|9.52±2.47|118.52±3.98|
> |CCIN|4.23±0.71|16.42±5.41|280.85±2.73|
>
> [1]CWN.NIPS2021
> [2]TopoTune, ICML2025
>
> Thank you again for the insightful feedback; we hope our clarifications have been helpful.

---

> > ### Author Rebuttal · Reviewer_inq5 · 2026-04-03
> >
> > I  still find the novelty margin somewhat unclear. The rebuttal improves the distinction between intrinsic CC neighborhoods and Hasse-graph-based analysis, but it remains uncertain how large the step is beyond a reformulation/unification together with one interesting sufficiency theorem. In other words, I now understand the intended contribution more clearly, but I am still not convinced that the comparative positioning and overall significance are as strong as the paper claims. Any more convincing conceptual evidence would help.

---

> > > ### Author Response · Authors · 2026-04-04
> > >
> > > # Novelty Margin and Conceptual Evidence
> > >
> > > >**Q1** ... but it remains **uncertain how large the step is beyond a reformulation/unification** together with one interesting sufficiency theorem... but I am still not convinced that the comparative positioning and overall significance are as strong as the paper claims. **Any more convincing conceptual evidence would help.**
> > >
> > > We appreciate the constructive comments. To provide compelling conceptual evidence, TopoTune's "Unified Hasse Graph" is the first work to present a neural message-passing mechanism on CCs and proves that the 1-WL test offers new insights into Hasse Graphs. However, 1-WL test on Hasse Graph may leads to changes in neighborhood relations. We address the reviewers' concerns from novelty and evidence.
> > >
> > > ## Theoretical Gap: 1-WL Hasse Graph vs. CCWL
> > >
> > > - **TopoTune Limitation**: TopoTune considers $B,N_\uparrow$，its per-rank neighborhoods (Fig.4 in TopoTune) are $N^0_{A,\uparrow}$ (nodes to nodes), $N^1_{A,\downarrow}$ (edges to edges), $N^1_{I,\downarrow}$ (edges to nodes), $N^2_{A,\downarrow}$(faces to edges). The standard 1-WL or k-WL is applied here (e.g., TopoTune), it aggregates neighborhood information into a single multiset. Hasse-graph-based WL refinement (see Prop B.4-B.8 in TopoTune) cannot distinguish different types of higher-order cycles if their local degree sequences match.
> > >
> > > - **CCWL Advantage**: CCWL operates on the intrinsic topological space using $N_\downarrow$ and $N_\uparrow$. According to Theorem 4.8, these operators are not just "neighbors" but sufficient statistics for boundary/co-boundary injections. CCWL treats upper and lower adjacencies as distinct ones.
> > >
> > > ## Novelty Margin
> > >
> > > For a clearer and more concise explanation, we further clearify as follows,
> > > ||TopoTune|CCIN(ours)|
> > > |:-|:-|:-|
> > > |Perspective|Flattened to Hasse Graph|Intrinsic operators (Topological relation)|
> > > |Insight|A new way to describe topological data|A new way to bound and optimize expressivity|
> > > |Expressivity|Bounded by Graph 1-WL/k-WL|Covers and surpasses HWL/SWL/CWL|
> > > |Engineering|Mapping to Hasse graph to implement|Efficiency via neighborhood redundancy removal|
> > >
> > > ## Conceptual Evidence: $C_3 \cup C_3$ (Disjoint) vs. $C_6$ (Single Cycle):
> > > Consider two non-combinatirial complexes:
> > >  - $CC_1:=C_3\cup C_3$ (two disjoint triangles), $C_3$ consists of three nodes and edges, one face as $v_1,v_2,v_3,e_{12},e_{23},e_{13},f_1$, similarly, $C_3$ also consists of $v_4,v_5,v_6,e_{45},e_{56},e_{46},f_2$.
> > >  - $CC_6:=C_6$ (a single hexagon), it consists of six nodes $v_1,\cdots,v_6$, six edges $e_{12},\cdots,e_{61}$, one face $f=[e_{12},\cdots,e_{61}]$.
> > >
> > > Initialize the node as 🔵, the edge as 🟣, and the face as 🔴.
> > >
> > > ### TopoTune: Hasse-Graph WL Failure
> > > TopoTune's CCWL tests the neighborhood operators (Fig.1, Fig.4 in TopoTune), such as $N^0_{A,\uparrow}$ (nodes to nodes), $N^1_{A,\downarrow}$ (edges to edges), $N^1_{I,\downarrow}$ (edges to nodes), $N^2_{A,\downarrow}$(faces to edges). Importantly, it does not incorporate messages from edges to faces $N^1_{A,\uparrow}$.
> > >
> > > **First WL iteration**: the two complexes still produce identical colors:
> > > - $C_3\cup C_3$: all nodes receive {🔵,{🔵,🔵},{🟣,🟣}}, all edges receive {🟣,{🔵,🔵},{🟣,🟣},🔴}, faces remain {🔴}.
> > > - $C_6$: all nodes receive {🔵,{🔵,🔵},{🟣,🟣}}, all edges receive {🟣,{🔵,🔵},{🟣,🟣},🔴}, faces remain {🔴}.
> > > - Update these multisets into new colors: ⚪$\leftarrow${🔵,{🔵,🔵},{🟣,🟣}}, 🟠$\leftarrow${🟣,{🔵,🔵},{🟣,🟣},🔴}, 🟢$\leftarrow${🔴}.
> > >
> > > **Second WL iteration**: they also yield exactly the same colors:
> > >  - $C_3\cup C_3$: all nodes receive {⚪,{⚪,⚪},{🟠,🟠}}, all edges receive {🟠,{⚪,⚪},{🟠,🟠},🟢}, faces remain {🟢}.
> > >  - $C_6$: all nodes receive {⚪,{⚪,⚪},{🟠,🟠}}, all edges receive {🟠,{⚪,⚪},{🟠,🟠},🟢}, faces remain {🟢}.
> > >
> > > Therefore, TopoTune’s Hasse Graph-WL test assigns identical colorings to CC$_1$ and CC$_2$ at every iteration, even though the two CCs are non-isomorphic.
> > >
> > > ### CCIN: Intrinsic Neighborhood WL
> > >
> > > CCIN's CCWL is formulated via $N_{\uparrow},N_{\downarrow}$ (Theorem 4.8) as follows,
> > >  - $C_3\cup C_3$: all nodes with labels {🔵,{🔵,🔵},{🟣,🟣}}, all edges with {🟣,{🔵,🔵},{🟣,🟣},🔴}, faces {🔴,{🟣,🟣,🟣}}.
> > >  - $C_6$: all nodes with labels {🔵,{🔵,🔵},{🟣,🟣}}, all edges with {🟣,{🔵,🔵},{🟣,🟣},🔴}, faces {🔴,{🟣,🟣,🟣,🟣,🟣,🟣}}.
> > >  - Update their colors as : ⚪$\leftarrow${🔵,{🔵,🔵},{🟣,🟣}}, 🟠$\leftarrow${🟣,{🔵,🔵},{🟣,🟣},🔴},🟢$\leftarrow${🔴,{🟣,🟣,🟣}} (**$C_3\cup C_3$, faces: $f_1,f_2$**), 🟡$\leftarrow${🔴,{🟣,🟣,🟣,🟣,🟣,🟣}} (**$C_6$, faces: $f_1$**).
> > >
> > > CCWL has distinguished $C_3\cup C_3$ and $C_6$. CCWL operates on the intrinsic complex. Through lower adjacency $N_\downarrow$, the face-level color refinement captures the boundary (3 for $CC_1$ vs. 6 for $CC_2$). Moreover, although the Hasse Graph k-WL test for $k \ge 2$ can distinguish $C_3 \cup C_3$ and $C_6$, it fails to distinguish $C_k \cup C_k$ and $C_{2k}$, but CCWL (CCIN) can distinguish.

---

### Official Review · Reviewer_798M · 2026-03-09

**Soundness:** 2
**Presentation:** 2
**Significance:** 3
**Originality:** 3
**Overall Recommendation:** 3
**Confidence:** 3

**Summary:**

This paper proposes a unified framework for learning on higher-order relational/topological domains, aiming to treat graphs, hypergraphs, simplicial complexes, cellular complexes, and related structures within a common perspective. Beyond introducing this general viewpoint, the paper attempts to organize the relationships among existing models and domains more systematically, and develops theoretical results intended to characterize the properties and expressive behavior of the framework. In that sense, the contribution is not only a new formalism, but also a broader conceptual synthesis that could help reduce fragmentation in the topological deep learning literature.

**Compliance With Llm Reviewing Policy:**

Affirmed.

**Final Justification:**

This paper has clear strengths in originality and potential significance, particularly in its attempt to provide a unified framework across several higher-order topological domains. The rebuttal and follow-up comment were helpful in clarifying the authors’ intended theory, but they did not change my final assessment because some important assumptions, proof dependencies, and the exact scope of the main theoretical claims became much clearer only through the rebuttal than in the paper itself. Since my evaluation must be based on the submitted manuscript rather than an anticipated revision, I remain unable to fully verify the central theory in its current form. Therefore, I maintain my weak reject recommendation: the ideas are promising, but the current version would benefit from a more complete and careful theoretical revision.

**Key Questions For Authors:**

1. **Table 1 and the graph domain.** In Table 1, several models do not appear to be checked for graphs. If I understand the domain hierarchy correctly, a graph can be viewed as a special case of a hypergraph, a simplicial complex, and a cellular complex. What notion of “supports graph data” is being used in the table, and why do those entries remain unchecked?
2. **Novelty relative to prior expressivity results.** The paper appears to contain a theorem similar in spirit to Theorem 6 of [1] What part of the proof technique is genuinely new here? If the novelty lies in handling a broader setting, a different technical obstacle, or a stronger conclusion, please make that distinction explicit. A clear answer here would improve my originality assessment.
3. **Coverage condition.** It seems that the coverage condition plays an important role in the main arguments. If this assumption is indeed required for the key results, it might be helpful to state it explicitly in the main text and briefly explain where it is used (e.g., around Lemma 4.7 / Theorem 4.8). Given that it appears to be a relatively strong condition, some discussion of its necessity or possible relaxations would also improve clarity.
4. **Section 4 notation and proof structure.** Could the authors revise Section 4 so that the relevant notation (including symbols such as $\sqsubseteq$ and $P_{\mathcal{CC}}$)  is self-contained in the main text, and clarify whether Lemma 4.4 is intended to be an if-and-only-if statement? At present, I could not tell whether the missing direction is unnecessary, omitted, or implicitly handled elsewhere. A clear clarification here would directly affect my soundness assessment.
5. **Appendix completeness.** Appendix A appears to contain a remaining “Definition X” placeholder. Is this only a drafting artifact, or is some definition/proof dependency currently missing from the supplementary material?

[1] Bodnar, Cristian, et al. "Weisfeiler and lehman go topological: Message passing simplicial networks." International conference on machine learning. PMLR, 2021.

**Limitations:**

The discussion could be strengthened. In particular, it would be helpful to more explicitly discuss the strength and necessity of the coverage condition, the scope of the theoretical guarantees, and limitations of the framework when these assumptions do not hold.

**Strengths And Weaknesses:**

### Strengths
* The paper addresses an important and timely problem. The topological deep learning literature is currently fragmented across multiple domains and modeling choices, so a unifying perspective is potentially valuable.
* A clear strength of the paper is its broad scope: bringing together multiple data types (e.g., simplicial complexes, hypergraphs, and related higher-order domains) under one framework is useful and could make the work relevant to multiple subcommunities.
* I found the effort to summarize and relate different domains/models genuinely helpful. The high-level organization is one of the stronger aspects of the submission, and the paper has the potential to serve as a useful reference point.
* In terms of significance, the work could be meaningful even if its immediate empirical or algorithmic impact is limited, because a successful unifying framework may help future researchers compare architectures more systematically and transfer ideas across domains.
### Weaknesses
* My main concern is soundness as presented. Many notations used in the main text are either not defined where they first appear or seem to be deferred to the appendix, which made the proofs substantially harder to follow. As a result, I had difficulty convincing myself that the central technical claims are fully correct.
* In particular, Lemma 4.4 appears to be proved only in one direction, and I was unsure whether the reverse direction is unnecessary or missing. More importantly, Lemma 4.7 and Theorem 4.8 seem to be among the paper’s main technical contributions, but I found them very difficult to parse even after allowing for possible typos.
* Appendix A still contains a “Definition X” placeholder. This may be a drafting oversight, but in a theory-heavy submission it is concerning, because it further reduces confidence that the manuscript has been carefully checked end-to-end.
* The role of the coverage condition is unclear to me because I could not find it clearly introduced and motivated in the main text. If this condition is indeed necessary, it appears fairly strong, and the paper should explicitly discuss why it is needed, where it is used, and what would fail without it.

---

> ### Author Rebuttal · Authors · 2026-03-30
>
> **W1:** Thank you for this critical feedback. To address these presentation problems, in the revision, we will reorganize the theory section, move the key definitions and lemmas needed for the main arguments from the appendix into the main text, and add a notation and short proof roadmap to clarify the role of each result and how it is used later.
>
> **W2/Q4:** Thank you for the constructive comment. We agree that the current proof of Lemma 4.4 only establishes the forward implication. We use its contrapositive: if there exists a iteration $t \ge 0$, {{$ c^{(t)}(\sigma): \sigma \in CC_1 $}} $\neq${{ $c^{(t)}(\tau): \tau \in CC_2$ }}, then $CC_1$ and $CC_2$ are non-isomorphic. Importantly, the reverse direction is not required for subsequent arguments. We will revise Lemma 4.4 to a one-way statement and remove the “if and only if”.
>
> **W2:** We agree that Lemma 4.7 and Theorem 4.8 are too notation-dense and that their proof is not sufficiently transparent in the main text. To clarify, “as powerful as” refers to equivalent discriminative power: two refinement rules distinguish exactly the same pairs of CCs (up to injective relabeling of colors). The logic proceeds in two steps:
> - Step 1: Lemma 4.7 shows that under the coverage condition, including the coboundary $c_C(\sigma)$ does not increase expressivity once $c_B(\sigma)$, $c_{N\downarrow}(\sigma)$, and $c_{N\uparrow}(\sigma)$ are given.
> - Step 2: Theorem 4.8 further removes the boundary, showing that upper and lower adjacency are sufficient to achieve the same expressive power as the full four-relation CCWL test.
>
> In the revision, we will introduce the coverage condition and highlight where it is used, and normalize notation throughout Section 4 (e.g., using $c^{(t)}(\sigma)$ consistently).
>
> **W3/Q5:** We sincerely apologize for this oversight. The “Definition X” in Appendix represents Definition 3.1 and does not indicate a missing definition or any missing proof dependency. We will correct definition reference, proofread the main text and appendices end-to-end.
>
> **W4/Q3/L1:** Thanks for this important comment. The coverage condition is necessary to guarantee that the boundary and co-boundary information can be fully recovered from the upper and lower adjacencies.
>
> - Where it is used: This condition is essential for the proofs of Lemma 4.7 and Theorem 4.8. It ensures that boundary $\partial(\sigma)$ (or co-boundary $\delta(\sigma)$) cell is shared by at least one other cell of the same rank.
> These cells appear in the intersection of lower and upper adjacencies $N_{\downarrow}, N_{\uparrow}$, allowing to reconstruct them as:
> $$\partial(\sigma)=\bigcup_{\tau \in N_{\downarrow}(\sigma)} B(\sigma,\tau),\delta(\sigma)=\bigcup_{\tau \in N_{\uparrow}(\sigma)} C(\sigma,\tau).$$
>
> - What fails without it: Without this condition, some boundary or co-boundary cells might not be shared with any other cell of the same rank. In such cases, these isolated cells could not be recovered from upper and lower adjacency , and the proof of neighborhood redundancy would fail.
>
> In the revision, we will define the coverage condition in the main text, show where it is used, and discuss its relaxations in the discussion.
>
> **Q1:** We agree that the notion of “supporting a domain” may cause confusion. Table 1 uses “support” to mean that a method is natively defined on the corresponding structure, rather than applied to graphs via lifting operation. Under this criterion, several higher-order models (e.g., simplicial or cellular methods) are not marked as supporting graphs. We will revise the caption to make this definition explicit.
>
>
> **Q2:** Thanks for insightful question. The fundamental distinctions are summarized below:
>
> - MSPN [1]: Theorem 6 [1] establishes expressive power of MSPN on simplicial complexes with boundary and upper adjacency, its element n-simplex strictly consists of convex hulls of n+1 nodes.
> - CCWL is defined on CCs domain (graphs, hypergraphs, simplicial, cellular complexes) with four intrinsic neighborhoods. CCWL proves that the upper and lower adjacency of CCWL are sufficient to reach full expressivity.
>
> We will revise Section 4.3 to clarify this distinction and the relationship to prior work [1].
>
> [1] Bodnar, Cristian, et al. MSPN, ICML2021
>
> We hope these clarifications address the reviewer’s concerns, and we thank the reviewer again for the insightful suggestions.

---

> > ### Author Rebuttal · Reviewer_798M · 2026-04-02
> >
> > The rebuttal was helpful in clarifying the authors’ intent, and I appreciate the planned revisions to improve the presentation. However, my main concern remains about the theoretical soundness and completeness of the current presentation, especially around the arguments related to Lemma 4.7 and Theorem 4.8. At this stage, these concerns are not fully resolved, so my overall assessment remains unchanged.

---

> > > ### Author Response · Authors · 2026-04-02
> > >
> > > >**Q1:... However, my main concern remains about the theoretical soundness and completeness of the current presentation, especially around the arguments related to Lemma 4.7 and Theorem 4.8.**
> > >
> > > We appreciate the recognition that our previous rebuttal clarified the intent and improved the presentation. We understand the reviewer’s concern: **whether reducing boundary/co-boundary operators to simple adjacency relations (as in Lemma 4.7 & Theorem 4.8) leads to a loss of topological information**. We demonstrate that under the "Coverage Condition," this reduction is mathematically sound and preserves the full expressive power of CCWL.
> > >
> > > - **Details of Coverage condition**. Assume that every boundary cell of a cell $\sigma$ is shared with some lower-adjacent cells, co-boundary cell of $\sigma$ is shared with some upper-adjacent cells. According to the definitions of lower (**Definition A.3**) and upper adjacency (**Definition A.4**). Then the boundary $\partial(\sigma)$ and co-boundary $\delta(\sigma)$  of $\sigma$ can be recovered from same-dimensional adjacency relations with bridge cells:
> > > $$ \partial(\sigma)=\bigcup_{\tau\in N_{\downarrow}(\sigma)} B(\sigma,\tau) = \bigcup_{\tau\in N_{\downarrow}(\sigma)} \partial(\sigma)\cap\partial(\tau),  \delta(\sigma)=\bigcup_{\tau\in N_{\uparrow}(\sigma)} C(\sigma,\tau) = \bigcup_{\tau\in N_{\uparrow}(\sigma)} \delta(\sigma)\cap\delta(\tau),
> > > $$
> > > which implys lower adjacency encodes boundary, while upper adjacency encodes co-boundary.
> > >
> > > - **Soundness of Lemma 4.7**:
> > >   - Refer to Lemma 4.5, we define $c^t=\{c^t_\sigma,c^t_B,c^t_{N_\uparrow},c^t_{N_\downarrow}\}$, $d^t=\{c^t_\sigma,c^t_B,c^t_C,c^t_{N_\uparrow},c^t_{N_\downarrow}\}$,then $c^t\sqsubseteq d^t$.
> > >   - Conversely, by the Coverage Condition for coboundaries ($\delta(\sigma)=\bigcup_{\tau\in N_{\uparrow}(\sigma)} C(\sigma,\tau)$), the multiset of colors from $N_{\uparrow}$ already records coboundary set $C$, then concludes $d^t\sqsubseteq c^t$, then $d^t \equiv c^t$. This ensures that the update rule is theoretically sound and captures the full co-boundary information.
> > >
> > > - **Soundness of Theorem 4.8**
> > >   - Building on Lemma 4.7, we further simplify the set to $c^t=\{c^t_\sigma,c^t_B,c^t_{N_\uparrow},c^t_{N_\downarrow}\}$, $e^t=\{c^t_\sigma,c^t_{N_\uparrow},c^t_{N_\downarrow}\}$, then $e^t\sqsubseteq c^t$.
> > >   - By applying the Coverage Condition for boundaries ($\partial(\sigma) = \bigcup_{\tau \in N_\downarrow(\sigma)} B(\sigma, \tau)$), we prove that the lower-adjacency multiset $N_\downarrow$ fully covers the information of the shared boundary set $B$, it concludes $c^t\sqsubseteq e^t$, then $c^t \equiv e^t$, and $c^t \equiv d^t \equiv e^t $.
> > >
> > >
> > > To further address **the concern of completeness of the theoretical soundness and the current presentation**, we provide the following **logical dependency roadmap** of our theoretical results in the manuscript:
> > > ```
> > > Lemma 4.4 -> Lemma 4.5 -> Corollary 4.6 -> Lemma 4.7
> > >                                 |          /
> > >                                 v         v
> > >                             Theorem 4.8 (★)
> > >         （upper adjacency and lower adjacency = full CCWL）
> > >                                 |
> > >                                 v
> > >                     Lemma 4.9 ( CCWL >= 1-WL )
> > >                 /       |           |           \
> > >                v        v           v            v
> > >         Prop. B.5      Prop. B.6    Prop. B.7    Prop. B.8
> > >     (CCWL>= 1-WL)  (CCWL>= HWL)  (CCWL>= SWL)  (CCWL>= CWL)
> > >             \          |            |           /
> > >              v         v            v          v
> > >                 Theorem 4.11 + Corollary B.9
> > >             (CCWL can simulate 1-WL / HWL / SWL / CWL)
> > >                             |
> > >                             v
> > >                         Lemma 5.1
> > >         (CCIN with impressive power of CCWL)
> > > ```
> > >
> > > 1. **Refinement Definitions**
> > >    - Lemma 4.4: (necessary condition), If two CCs are isomorphic, then their color multisets are equal at every iteration.
> > >    - Lemma 4.5 ：For two colorings c ⊑ d, if d can distinguish, such that c can also distinguish
> > >    - **Corollary 4.6**: If two colorings c ⊑ d, and d can distinguish non-isomorphic CCs, then c can also distinguish them.
> > >
> > > 2. **Main Theorems**
> > >    - **Lemma 4.7**: CCWL update rule with $B,N_\downarrow,N_\uparrow$ is as powerful as CCWL(full).
> > >    - **Theorem 4.8 ★**: CCWL update rule with $N_\downarrow,N_\uparrow$ is as powerful as CCWL(full).
> > >
> > > 3. **CCWL Generalization**
> > >    - Lemma 4.9: CCWL is at least as powerful as 1-WL.
> > >    - Proposition B.5-B.8: faithful liftings, CCWL simulates 1-WL, HWL, SWL, and CWL.
> > >    - Corollary B.9: CCWL is strictly more expressive than each of 1-WL, HWL, SWL, and CWL.
> > >    - Theorem 4.11: There exist faithful liftings that CCWL simulates the classical WL variants.
> > >
> > > 4. **CCWL implement**
> > >    - **Theorem 5.1**: With injective aggregators and sufficient MLP depth, CCIN is as powerful as CCWL.
> > >
> > > We hope this reply resolved your issue. Please feel free to ask any further questions.

---

### Official Review · Reviewer_q9SX · 2026-03-13

**Soundness:** 3
**Presentation:** 1
**Significance:** 3
**Originality:** 2
**Overall Recommendation:** 4
**Confidence:** 4

**Summary:**

This paper introduces the Combinatorial Complex Weisfeiler-Lehman (CCWL) test to provide a unified theoretical framework for analyzing the expressive power of neural networks operating on diverse topological domains, including graphs, hypergraphs, simplicial complexes, and cellular complexes. The authors formalize topological message-passing across four fundamental neighborhood relations: boundary, coboundary, upper adjacency, and lower adjacency. The central theoretical contribution is a mathematical proof demonstrating that upper and lower neighborhoods alone are sufficient to achieve the full discriminative power of the CCWL test, essentially rendering boundary and coboundary information redundant for expressivity. Building on this theoretical insight, the authors propose the Combinatorial Complex Isomorphism Network (CCIN), an architecture that strategically prunes these redundant relations. Empirical evaluations validate this approach, demonstrating that CCIN achieves strong, competitive performance on standard molecular and graph benchmarks.

**Compliance With Llm Reviewing Policy:**

Affirmed.

**Final Justification:**

My final recommendation is a Weak Accept (4). The authors' detailed rebuttal adequately addressed my specific concerns regarding prior art and technical nuances, prompting me to raise my score from my initial 3.

While the paper's strong theoretical core and originality ultimately outweigh its presentation issues, clarity remains the primary limitation. I agree with Reviewer 798M that the volume of needed structural and notational corrections suggests the original manuscript was premature for a top-tier venue.

Because ICML policies prevent us from reviewing the updated manuscript to verify these extensive changes, I didn't raise my score to a full Accept. However, the foundational work is solid. If all proposed revisions are carefully integrated, the paper will meet the bar for acceptance and be a valuable contribution to the community.

**Key Questions For Authors:**

For the rebuttal, please prioritize addressing the above raised weaknesses. Beyond that, I just have a few specific questions/comments:

- **Clarification on TopoTune claims:** In the intro (L32-35), you state that "TopoTune (Papillon et al., 2025) studies combinatorial complex neural networks, but simplifies as three neighborhood types and disrupts the higher-order information propagation." Can you elaborate on this? To the best of my knowledge, they consider up/down adjacencies and incidences, thereby accounting for the 4 neighborhoods you are focusing on. Additionally, I don't quite understand the claim about "disrupting" higher-order information propagation. Could you clarify this specific technical difference?
- **Data splits:** Can you guarantee that all of the results reported in the tables use the exact same data splits and evaluation protocols? I want to ensure that no values were directly reused from other papers that might have employed potentially different experimental setups.
- **Circular definition:** The term “boundary” is not defined, and it is used in L130-133 of Section 3 in a recursive way: “The combinatorial structure of CC can be compactly described by an incidence relation called the boundary relation, denoted σ ≺τ, indicating that cell σ lies on the boundary of a higher-dimensional cell τ.”
- **Dataset Selection:** Is there any specific reason why you didn't consider a purely topological dataset like MANTRA (Ballester et al., 2025) in your evaluation?

**Limitations:**

Authors could include a discussion about limitations in the Appendix.

**Strengths And Weaknesses:**

I have mixed feelings about this submission. On the positive side, the core mathematical contribution—proving the redundancy of specific topological neighborhoods for expressivity—is technically sound, elegant, and valuable for the field of topological deep learning. However, my main reservations stem from how the contribution is currently framed with respect to recent literature. The paper positions itself as introducing a unified theoretical foundation for combinatorial complexes to fill a gap in the field. Yet, it does not discuss the recent TopoTune framework (Papillon et al., 2025), which explored a generalized message-passing scheme for these structures and formalized a WL test on Combinatorial Complexes. Because of this, the current framing makes it difficult to accurately assess the exact delta of the proposed foundational claims.

**Strengths**

- **Significant Theoretical Optimization:** The core mathematical proof (Theorem 4.8) demonstrating that upper and lower adjacencies are strictly sufficient to match the expressivity of a full four-neighborhood CCWL test is a strong and elegant contribution. It neatly addresses the redundancy problem in topological message passing.
- **Strong Empirical Validation:** The authors successfully translate their theoretical findings into practical gains. The empirical results demonstrate that pruning boundary and coboundary relations not only maintains expressive power but also reduces training time significantly (showing up to 24.8% speedups on datasets like ZINC) and occasionally improves performance by reducing topological noise.
- **Thorough Ablation Studies:** The ablation studies are well-executed and directly support the paper's theoretical claims regarding the sufficiency and redundancy of specific neighborhood messages.

**Weaknesses**

- **Positioning and Prior Art:** The authors describe introducing the CCWL test as a novel foundation for the field. Given that TopoTune explicitly formalizes Weisfeiler-Leman tests on Combinatorial Complexes (in Appendix B.3.2), the current framing feels disconnected from this prior art. It would strengthen the paper to explicitly compare these formulations rather than suggesting the foundation is lacking in the literature.
- **Clarification Needed on Table 1:** In Table 1, the proposed framework is presented as uniquely capable of supporting all topological structures (graphs, hypergraphs, simplicial, and cellular complexes). Since TopoTune's Generalized Combinatorial Complex Networks (GCCNs) also apply to these TDL domains, I am curious why this baseline was omitted from the structural comparison.
- **Clarification on Structural Assumptions:** In Section 3, the authors assume that "the complex contains no free faces." While I understand this assumption elegantly avoids mathematical edge cases (such as the boundary recovery issue for isolated cells detailed in Appendix A), real-world combinatorial data might contain unclosed structures or free faces. I would appreciate it if the authors could provide a more exhaustive discussion in the main text regarding the practical and theoretical implications of this assumption when applying the CCWL test and CCIN model to irregular, real-world topologies.

I am currently leaning towards a Weak Reject, primarily due to the concerns regarding positioning and the omission of relevant discussions and comparisons with prior art (TopoTune). However, I want to emphasize that I find the theoretical contribution of this paper to be of solid quality and of ICML caliber. I am open to raising my score during the rebuttal phase. I invite the authors to either adapt the framing to shift the focus toward their theoretical refinement of existing frameworks, or to help me understand why TopoTune's CCWL formulation and domain generality are fundamentally incomparable to their own, thereby justifying the omission. Finally, please feel free to reply to these points with total frankness; I appreciate straightforward language and open scientific discussion.

---

> ### Author Rebuttal · Authors · 2026-03-30
>
> **W1/Q1:** We thank the reviewer for this constructive feedback. We agree that our original descriptions of TopoTune are not precise and may lead to confusion. We clarify CCIN and TopoTune claims as follows.
>
> **TopoTune vs. CCIN：**
> 1. **Neighborhoods:**
>    - TopoTune formulates neighborhood relations through the augmented Hasse graph (e.g., $N_{I,\downarrow},N_{A,\uparrow}$ in Section 4 [1]: Fig.2, Fig.3, Fig.4) and defines message passing on the Hasse graph.
>    - CCIN starts from the four intrinsic neighborhoods of CCs (e.g.,boundary, coboundary, lower and upper adjacency in [2]), and uses them to characterize topological message passing in CCs.
> 2. **Expressive power:**
>    - TopoTune like the CWL on cellular complexes [3] considers boundary and upper relations. It studies expressiveness by mapping the structure to the augmented graphs and uses standard WL test (In [1], Appendix B.3.2, Prop.B.11, B.12), then proves CCWL test is equivalent to the WL test on the augmented Hasse graph.
>    - CCIN instead defines four types neighborhood-based WL operators directly on CCs (Appendix A.1), using these operators to describe color refinement and message passing update.
> 3. **Theoretical contributions:**
>    - TopoTune unifies different topological structures through the Hasse-graph construction and its Prop.B.11 establishes the equivalence of CCNN with WL/k-WL on the augmented graph.
>    - CCIN does not aim to seek another unification, but instead show that under certain structural conditions, boundary and coboundary information can be removed for expressiveness. We prove that the expressive power of the full CCWL can be recovered using only upper and lower adjacency.
>
> The original phrase “disrupting higher-order information propagation,” can be rewritten as, "TopoTune defines an expansion mechanism that transforms combinatorial complexes (CCs) into a collection of graphs, various higher-order relations are unified on strictly augmented Hasse graph". TopoTune reflects an expansion from an intrinsic topological relations to a unified graph-based view. We will clarify this point in the revision.
>
> **W2:** We agree that TopoTune should be included in Table 1 and will add it. TopoTune's core contribution lies in its general combinatorial complex neural network (GCNNs) and its expressive power via 1-WL on Hasse graph. The table originally focused on methods natively defined for specific topological domains, which was not clearly stated. We will add TopoTune and revise the Table 1, e.g., TopoTune supports simplicial, cellular complexes.
>
>
> **W3:** Thanks for this question. Structural assumptions are used in the redundancy analysis (Lemma 4.7 and Theorem 4.8) to ensure that boundary/coboundary information is not structurally isolated and can be recovered from rank-based neighborhood relations.
>    - If free faces exist, some boundary or co-boundary cells $\sigma$ belong to only a single cell and do not appear in the intersection of any rank-based adjacency relationships $N_{\downarrow},N_{\uparrow}$. In this case, such information cannot be recovered from adjacency alone, and CCWL test achieves full expressiveness via upper/lower adjacency no longer holds strictly.
>    - Real-world data often contain isolated cells. In practice, CCIN relaxes the strict assumption by handling such isolated cells by initializing missing upper/lower neighborhood information with their own features. This assumption can be relaxed in practic even when the strict theoretical assumption is not met.
>
> We will revise the main text to introduce this assumption earlier, clarify its theoretical role.
>
> **Q2:** Thank you for raising this important concern. We agree that consistent data splits and evaluation are critical for fair comparison. In our experiments, we follow the settings [3,4], such as TUDataset (10-fold evaluation), Peptides, OGB and ZINC datasets use train/val/test splits (8:1:1). For majority baselines, we reproduce their methods, but some baselines (RWK,GK, WL kernel) are not available. We report results from the original papers. We will further clarify these details in the revision.
>
> **Q3:** We agree that the current manuscript's statement here is not rigorous enough and could easily lead to the use of the term "boundary" before it is defined. We will formally define the boundary relationship in revised manuscript.
>
> **Q4:** Thanks for your suggestion. MANTRA is designed for benchmarks on manifolds and triangulations. CCIN experiments focus on CCs lifted from the molecular graph. These settings are not directly aligned, as CCIN does not explicitly model properties such as orientability in 2D manifolds.
>
> **L1:** In the revision, we will discuss the structural assumptions，computational cost and application scenarios in the limitations.
>
> [1] TopoTune，ICML2025
> [2] Topological deep learning: Going beyond graph data
> [3] CWN.NIPS2021
> [4] GIN.ICLR 2019
>
> We hope this addresses the reviewer's concerns and appreciate your constructive comments.

---

> > ### Author Rebuttal · Reviewer_q9SX · 2026-04-04
> >
> > Thank you for the detailed rebuttal. You have adequately addressed my specific concerns, particularly regarding prior art and technical nuances. Accordingly, I am upgrading my score from 3 to 4.
> >
> > That said, after reading the other reviews I agree with Reviewer 798M that the volume of clarifications and notational corrections needed suggests the original manuscript was not quite ready for a top-tier venue. And unfortunately, since ICML policies do not allow manuscript updates during this phase, we are unable to review the fully revised paper to verify these extensive changes. This is what prevents me from further raising my score.
> >
> > However, I acknowledge the strong theoretical core of this work. If all your proposed changes are carefully integrated, in my opinion the paper will be at the level expected for acceptance–and this message will also be included in my final recommendation. Thank you for your hard work during this discussion phase.

---

> > > ### Author Response · Authors · 2026-04-04
> > >
> > > Thanks for your detailed follow-up and for upgrading your score.
> > >
> > > The remaining concern regarding the readiness of the manuscript has been addressed through a structured and comprehensive revision rather than incremental edits. In particular, we have made the following revisions:
> > >
> > > - Section 4 has been reorganized with a clear proof roadmap, making the logical dependencies between Lemma 4.4, Lemma 4.7, and Theorem 4.8 explicit and verifiable.
> > > - All notation is defined at first occurrence via a self-contained table, eliminating cross-referencing gaps between the main text and appendix.
> > > - Key assumptions, including the coverage condition and structural constraints, are explicitly introduced, motivated, and tied to their roles in the main results.
> > > - Previously deferred or ambiguous definitions have been moved into the main text to ensure that all technical claims can be checked locally.
> > >
> > > These changes reflect a systematic restructuring aimed at making the theoretical core fully transparent and verifiable. Although these revisions cannot be inspected under the current review policy, they have already been consolidated into a complete and internally consistent version.
> > >
> > > We appreciate your recognition of the theoretical contribution and your support in final recommendation.

---

### Decision · Program_Chairs · 2026-04-30

**Decision:**

Reject

**Comment:**

From my assessment of the reviews, the rebuttal and the discussion, this submission presents an interesting idea: a unifying expressivity framework for combinatorial complexes with a potentially meaningful sufficiency result. However, the central issue is that too much of the paper’s theoretical core appears to depend on clarification(s) outside the submitted manuscript. In fact, multiple reviewers, including those sympathetic to the work, raised concerns about missing or incomplete definitions, unclear assumptions, novelty positioning relative to prior work (e.g., TopoTune), and comparative claims that appear stronger than what is directly established within the paper. While the rebuttal clarified issues and  improved understanding, it also reinforced my impression that resolving these issues would require major restructuring and nontrivial revision to the core theory and positioning, rather than minor polishing.

Given that this is a theory-heavy submission to a top-tier ML venue, these concerns weigh heavily. From my perspective, the key issue is not whether the ideas may be promising, but whether the current submission, as reviewed, supports its main claims at the standard expected for an acceptance decision. Again, the scope of the required modifications appears substantial to the extent where the paper has to undergo another full round of review. I therefore recommend reject, but encourage resubmission after carefully revising the framing, the theoretical results, and incorporating the reviewer comments.